# 't Hooft lines of ADE-type and Topological Quivers

Y. Boujakhrout, E.H Saidi, R. Ahl Laamara, L.B Drissi

1. LPHE-MS, Science Faculty, Mohammed V University in Rabat, Morocco

2. Centre of Physics and Mathematics, CPM- Morocco, Rabat

December 30, 2022

### Abstract

We investigate 4D Chern-Simons theory with ADE gauge symmetries in the presence of interacting Wilson and 't Hooft line defects. We analyse the intrinsic properties of these lines' coupling and explicate the building of oscillator-type Lax matrices verifying the RLL integrability equation. We propose gauge quiver diagrams $Q_G^\mu$ encoding the topological data carried by the Lax operators and give several examples where Darboux coordinates are interpreted in terms of topological bi-fundamental matter. We exploit this graphical description $(i)$ to give new results regarding solutions in representations beyond the fundamentals of $sl_N$, $so_{2N}$ and $e_{6,7}$, and $(ii)$ to classify the Lax operators for simply laced symmetries in a unified $E_7$ CS theory. For quick access, a summary list of the leading topological quivers $Q_{ADE}^\mu$ is given in the conclusion section [Figures **29**.(a-e), **30**.(a-d) and **31**.(a-d)].

**Keywords**: 4D Chern-Simons theory, Wilson /'t Hooft lines, Lax operators, Oscillator realisation, Gauge quiver diagrams, Topological matter, $E_7$ Unified Theory.

## 1  Introduction

Integrable two-dimensional field theories and spin models represent a significant area in classical and quantum physics that still bear several open questions intending to explicitly describe the interactions between fundamental particles [1]- [9]. The investigation of special features of these low dimensional theories has aroused much interest since the integrable spin chains advent [10] and the factorisation of many body scattering amplitudes of relativistic QFT [11, 12]. In these regards, tremendous efforts have been deployed to deal with the basic equations underlying these systems by following various approaches such as the Bethe Ansatz [13]- [15], quantum groups [16] and algebraic methods involving Yangian and graded-Yangian representations [17]- [20].

Recently, these efforts gained a big impulse after the setup by Costello, Witten and Yamazaki of a Chern-Simons -like theory living on four-dimensional $M_4$ with the typical (rational) fibration $\mathbb{R}^2 \times C$, and having a complexified gauge symmetry $G$ [21]- [23]. This topological gauge theory represents a higher dimensional field framework to approach quantum integrability and offers a new form of the gauge/Integrability correspondence [24]- [31]. On another side, it bridges to $N = (1, 1)$ supersymmetric Yang Mills theory in six and lower dimensions [32]- [35] and to supersymmetric quiver gauge theories [36]- [39]. It also allows for an interesting realisation of solvable systems in terms of intersecting M-branes of the 11d M-theory and, via dualities, in terms of intersecting branes in type II strings with NS5- and D-branes as the main background [40]- [43].

The main ingredients of the 4D Chern-Simons theory are line and surface defects [44]-[49]; these topological quantities play a fundamental role in the study of this theory and the realisation of lower dimensional solvable systems. In particular, we distinguish two basic line operators: $(i)$ Electrically charged Wilson lines which, roughly speaking, are asimilated to worldlines of particles in 2D space-time with a spectral parameter $z$ related to rapidity; they are characterised by highest weights $\lambda$ of representations $R$ of the gauge symmetry $G$. $(ii)$ Magnetically charged 't Hooft lines characterised by coweights $\mu$ of $G$ and acting like Dirac monopoles. The coupling of these lines in the 4D gauge theory is behind important results of quantum integrability. In these regards, recall that inersecting Wilson lines yield a nice realisation of the famous R-matrix and the Yang-Baxter equation of integrable two-dimenional QFTs [31].

Regarding the magnetically charged line defects to be further explored in this paper, they have recently been subject to particular interest where they were interpreted in terms of the monodromy matrix for non compact spin chains, the transfer matrix for compact spin chains [50, 51] and more specifically as the Baxter Q-operator [52]. They have also been implemented in various contexts as boundaries of surface defects [53], or as type II branes intersecting along distinguished directions [54]. Moreover, these brane realisations open windows to links between integrable spin and superspin chains and supersymmetric gauge quiver theories via correspondences like the so-called Bethe/gauge correspondence [55]-[58].

In what concerns us here, a quantum integrable XXX spin chain with $N$ nodes can be generated in the framework of the 4D CS by taking $N$ parallel Wilson lines perpendicularly crossed by a 't Hooft line standing for the magnetic field created by the system of the spin chain particles [52]. In this spirit, one can calculate the Lax operator for each node of the chain as a coupling of Wilson and 't Hooft lines in the gauge theory. The power of this construction with interacting lines in 4D comes from: $(i)$ the topological invariance on the real plane $\mathbb{R}^2$ that translates into the RLL integrability equation, $(ii)$ the Dirac -like singularity of the topological gauge configuration in the presence of 't Hooft line yielding the oscillator realisation of the Lax operator, $(iii)$ the holomorphy of observables on the Rieman surface $C$ where the complex parameter $z$ allows for realisations in the Yangian representation. These features constitute the common thread of the fascinating results derived from this Gauge/Integrability correspondence. In particular, it was shown in [52] that for the special case where the magnetic charge of the 't Hooft line is given by a minuscule coweight $\mu$ of the

gauge group $G$, the oscillator realisation of the Lax operator for a spin chain with internal symmetry given by $g$, the lie algebra of $G$, is easily constructed in 4D CS as the parallel transport of the topological field connexion through the singular 't Hooft line. This yields a general formula permitting to explicitly realise the Lax or the L-operator in the fundamental representation of any lie algebra $g$ having at least one minuscule representation, in terms of harmonic oscillators.

The main goal of this paper is to deeply analyse the data carried by the Lax operator and encode it into a simple gauge quiver description unveiling interesting common features of this quantity. These properties are relevant for both the study of integrable spin chains and of the gauge fields behavior in the presence of disorder operators. To this end, we investigate 4D Chern-Simons theories on $\mathbb{R}^2 \times \mathbb{CP}^1$ with complex gauge symmetries $G = A_n$, $D_n$, $E_{6,7}$ by implementing Wilson and 't Hooft line defects and studying intrinsic topological features of their coupling. In these regards, notice that the oscillator realisation of Lax matrices for minuscule nodes of $sl_N$ and $so_{2N}$ was firstly recovered from 4D CS in [52]; the exceptional $E_6$ and $E_7$ minuscule Lax operators were constructed in details in [59], while a full list of ABCDE minuscule Lax matrices is collected in [60] where the absence of a Lax matrix for the $E_8$ symmetry is because this group has no minuscule coweight. Here, in order to graphically visualise the effect of the Dirac-like singularity induced by a 't Hooft line on a deep level of the gauge configuration, we treat each case separately by demystifying the Lie algebra components appearing in the construction of the L-operator and derive its action on the internal quantum states by using a projector basis in the electric representation. Eventually, we can build the corresponding topological quivers $Q_G^\mu$ where we translate the topological data of the lines' coupling into quiver-like diagrams with nodes and edges as inspired from supersymmetric quiver gauge theories (see subsection 3.1 for motivation). This graphical representation allows to $(i)$ interpret sub-blocks of the L-matrices in terms of topological adjoint and bi-fundamental matter, $(ii)$ forecast the form of cumbersome Lax matrices without explicit calculation, $(iii)$ link Levi decompositions of ADE Lie algebras to exceptional symmetry breaking chains of a unified $E_7$ Chern-Simons theory. These results are summarized in the conclusion section, see Figures 29.(a-e), 30.(a-d), and 31.(a-d).

The presentation is as follows: In section 2, we begin by considering the 4D CS theory with $SL_N$ gauge symmetry as a reference model where we describe in details the implementation of the electrically and magnetically charged line defects and the calculation of their coupling in the topological theory. We revisit the oscillator realisation of the A-type minuscule Lax operators in the fundamental representation and then extend the construction by discussing other cases where electric charges of the Wilson lines correspond to representations of $sl_N$ beyond the fundamental. In section 3, we derive the topological gauge quiver diagrams corresponding to the A-type L-operators calculated in section 2, and give an interpretation of their nodes and links in terms of topological matter. Moreover, we yield quiver diagrams describing the form of L-operators for the symmetric $\boldsymbol{N} \vee \boldsymbol{N} \vee \boldsymbol{N}$ and adjoint representations of $sl_N$. In section 4, we study the minuscule D-type line defects in 4D CS theory with $SO_{2N}$ gauge invariance. Here, we distinguish two sub-families given by the vector-like minuscule coweight, and the two spinorial ones. Focussing on the vector-like family, we calculate the corresponding L-operator and construct the associated topological gauge quiver. In section

5, we move on to the minuscule spinor-like D-type L-operators where we also build the associated topological quiver. Other aspects concerning fermionic lines and the link with the $sl_N$ family are also discussed. In section 6 and 7, we similarly treat the 4D CS theories with exceptional $E_6$ and $E_7$ gauge symmetries in order, we focus on the minuscule topological lines and their associated topological quivers. The conclusion section is devoted to a summary of the results.

# 2   Wilson and 't Hooft lines of A- type

In this section, we begin by focusing on the 4D Chern-Simons theory of [23] with $sl_N$ gauge symmetry where we introduce the basics of this theory and the implementation of topological line defects. We consider the various types of minuscule 't Hooft lines for the $sl_N$- family with $N \geq 2$ and investigate their interaction with electric Wilson lines. We show how the symplectic oscillators of the phase space of 't Hooft lines allow for an explicit realisation of the Lax operators. We moreover extend the results by considering Wilson lines for different representations of $sl_N$ and investigating their properties according to the nature of their electric charges.

## 2.1   Electric/Magnetic lines in $sl_N$ Chern-Simons theory

In order to study the A- type electric Wilson lines and magnetic 't Hooft line defects as well as their interpretation in quantum integrable systems, we begin by briefly recalling some useful aspects of the 4D Chern-Simons theory with $SL_N$ gauge symmetry. This is an unconventional topological field theory living on a 4D space $\boldsymbol{M}_4$ that we take as $\mathbb{R}^2 \times \mathbb{CP}^1$ parameterised by $(x, y; z)$ with real $(x, y)$ for $\mathbb{R}^2$ and local complex $z = Z_1/Z_2$ for $C = \mathbb{CP}^1$. The field action of the topological theory was first constructed in [26] and reads as follows

$$S_{4dCS} = \int_{\mathbb{R}^2 \times \mathbb{CP}^1} dz \wedge tr\Omega_3, \tag{2.1}$$

where $\Omega_3$ is the CS 3-form

$$\Omega_3 = \mathcal{A} \wedge d\mathcal{A} + \frac{2}{3}\mathcal{A} \wedge \mathcal{A} \wedge \mathcal{A}, \tag{2.2}$$

with 1-form gauge potential $\mathcal{A} = t_a\mathcal{A}^a$ where $t_a$ stand for the generators of $sl_N$ and $\mathcal{A}^a$ is a partial gauge connection as follows [26]

$$\mathcal{A}^a = dx\mathcal{A}_x^a + dy\mathcal{A}_y^a + d\bar{z}\mathcal{A}_{\bar{z}}^a \tag{2.3}$$

The contribution of the missing component $dz\mathcal{A}_z^a$ is killed by the the factor $dz\wedge$ in the measure of the field action $S_{4dCS}$; and can be treated in terms of an extra symmetry in the QFT formulation. The equation of motion of the potential field $\mathcal{A}$ is given by the vanishing gauge curvature

$$\mathcal{F}_2 = d\mathcal{A} + \mathcal{A} \wedge \mathcal{A} = 0 \tag{2.4}$$

This flat curvature property agrees with the topological nature of the CS theory indicating that the system is in the ground state with zero energy. To deform this state, we consider observables given by line or surface defects such as the Wilson $W_{\xi_z}^{\boldsymbol{R}}$ and 't Hooft $tH_{\gamma_0}^{\mu}$ lines that we are interested in here. These are represented by curves in the topological plane $\mathbb{R}^2$ and located at positions z in $\mathbb{CP}^1$; they can be represented as in the Figure 1.

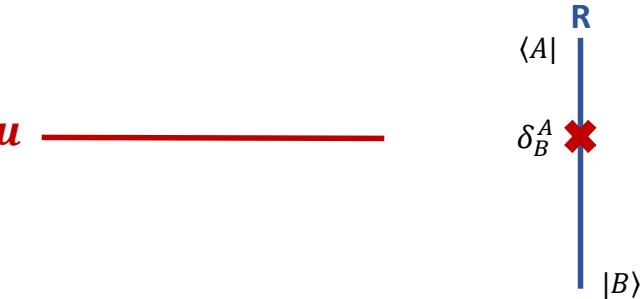

Figure 1: Line defects in the real plane $\mathbb{R}^2$. On the left, a horizontal 't Hooft line with magnetic charge $\mu$ expanding along the x-axis ($y = 0$) at $z = 0$. On the right, a vertical Wilson line expanding along the y-axis ($x = 0$) at $z \neq 0$ with electric charge in some representation $\boldsymbol{R}$. Notice that the 't Hooft line is in fact paired to a similar one located at $z = \infty$ with magnetic charge $-\mu$ [52].

Regarding the Wilson lines expanding along $\xi_z \subset \mathbb{R}^2$ with $z \in \mathbb{CP}^1$, they are electrically charged and can be naively thought of as

$$W_{\xi_z}^{\boldsymbol{R}} = Tr_{\boldsymbol{R}} \left[ \mathrm{P} \exp \left( \int_{\xi_z} \mathcal{A} \right) \right] \tag{2.5}$$

This shows that they are functions of $\xi_z$ and $\boldsymbol{R}$ which is here a representation of $sl_N$ characterised by a highest weight state $|\omega_{\boldsymbol{R}}\rangle$ with $\omega_{\boldsymbol{R}} = \sum_{i=1}^{N-1} n_i^{\boldsymbol{R}} \omega_i$. To perform explicit calculations, $\boldsymbol{R}$ is often taken as the (anti-) fundamental $\boldsymbol{N}$ representation of $sl_N$ with fundamental weight $\omega_1$; however this construction can be extended to the other $sl_N$ representations $n_i^{\boldsymbol{R}} \omega_i$ such as the family of completely antisymmetric representations $\boldsymbol{N}^{\wedge k} \sim \omega_k$, the family of completely symmetric $\boldsymbol{N}^{\vee n} \sim n\omega_1$ and the adjoint representation $\boldsymbol{N} \times \bar{\boldsymbol{N}}$. As examples, Wilson lines with electric weight charges in the representations $\boldsymbol{N} \wedge \boldsymbol{N}$ and $\boldsymbol{N} \vee \boldsymbol{N}$ as well as in the adjoint are depicted in the Figure 2. The interest into Wilson lines $W_{\xi_z}^{\boldsymbol{R}}$ with generic $\boldsymbol{R}$ can be motivated by the two following:
($\boldsymbol{1}$) The special $sl_N$ representation theory where from fundamental objects like $\boldsymbol{R} = \boldsymbol{N}$ and/or $\bar{\boldsymbol{N}}$ with weight $\omega_{N-1}$, one can construct many composites carrying higher weight charges and describing higher conserved quantities. For example, the particles' current running along $W_{\xi_z}^{\boldsymbol{R}}$ is given by quadratic composites transforming like $\boldsymbol{N} \otimes \bar{\boldsymbol{N}} = \boldsymbol{1} + \boldsymbol{adj}$. In this regard, notice that for the fundamental $W_{\xi_z}^{\boldsymbol{R}=\boldsymbol{N}}$, we have N quantum states $|A\rangle$ traveling along the

vertical blue line of the Figure **1**. They couple to the CS gauge field like $\mathcal{J}_a \mathcal{A}^a$ with $\mathcal{J}_a \sim \langle A | t_a | B \rangle$.

(**2**) Knowing the action of the minuscule coweight $\mu$ on the fundamental representation of $sl_N$, we can deduce its action on higher dimensional representations by help of the tensor product properties. To fix ideas, see eq(2.26).

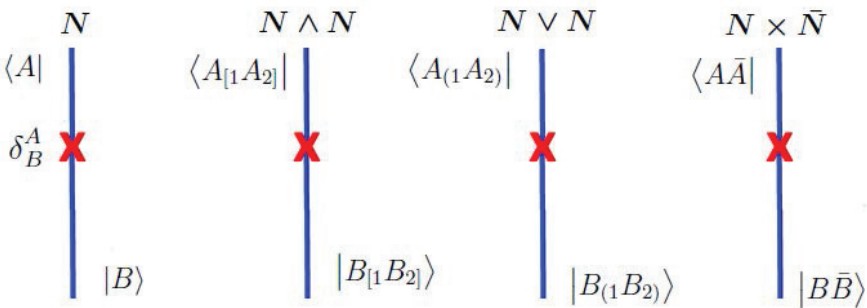

Figure 2: Four examples of Wilson lines in different representations of $sl_N$ occupying vertical lines in $\mathbb{R}^2$ (x=cte); they carry different electric charges. The representation **R** and the type of incoming quantum states are indicated above each line, while the outgoing states are given at the bottom of the line. The red cross indicates a local interaction point.

Concerning the 't Hooft lines that we denote like $tH_{\gamma_0}^{\mu}$, they are magnetically charged line defects with magnetic charge given by a minuscule coweight $\mu$ of the complex Lie algebra $sl_N$. The curve $\gamma_0$ belongs to $\mathbb{R}^2$ and sits at a point $z_0$ in the holomorphic plane that we take at the origin. It is imagined in the 4D CS theory as the intersection $\mathcal{U}_1 \cap \mathcal{U}_2$ of two patches $\mathcal{U}_1$ and $\mathcal{U}_2$ of the topological plane $\mathbb{R}^2$. Following [52], the topological field $A^{[\mu]}$ sourced by the magnetic 't Hooft line defect $\gamma_0$ is generated by a *singular* gauge transformation $g = g(z)$ from the patch $\mathcal{U}_1$ to the patch $\mathcal{U}_2$. By thinking of $\gamma_0$ as coinciding with the x-axis in the topological plane, meaning that

$$\gamma_0 = \mathbb{R}^2_{y\leq 0} \cap \mathbb{R}^2_{y\geq 0} \qquad , \qquad \mathbb{R}^2 = \mathbb{R}^2_{y\leq 0} \cup \mathbb{R}^2_{y\geq 0}, \tag{2.6}$$

the gauge configuration $A^{[\mu]}$ in the presence of singularity $\mu$ is generated by a parallel transport of the gauge field bundles from $\mathbb{R}^2_{y\leq 0}$ towards $\mathbb{R}^2_{y\geq 0}$. In this case, the transport path is then given by the y-axis and the topological gauge configuration is given by

$$\mathcal{A}^{[\mu]} = g_1 z^{\mu} g_2 \tag{2.7}$$

with gauge transformations $g_1(z)$ and $g_2(z)$ singular near $z = 0$ but regular in the neighbourhood of $z = \infty$ with the limit $g_1(\infty) = g_2(\infty) = I_{id}$. Notice that $z^{\mu}$ is the operator $exp(log(z)\mu)$ with $\mu$ refering to the adjoint action of the coweight operating as in eqs(2.10,2.14). Using this configuration, one can associate to the $tH_{\gamma_0}^{\mu}$ the following gauge

invariant observable measuring the parallel transport from $y \leq 0$ to $y \geq 0$ as follows

$$L^{[\mu]}(z) = \mathrm{P} \exp \left( \int_y dy \mathcal{A}_y^{[\mu]} \right) \tag{2.8}$$

This $L^{[\mu]}$ is a holomorphic function of z valued in the $SL_N$ gauge group; it may have poles and zeros at z= 0 and z=$\infty$ arising from the $tH_{\gamma_0}^\mu$ at $z = 0$ and the mirror $tH_{\gamma_\infty}^{-\mu}$ line at $z = \infty$ [52]. The gauge singularity is implemented in this construction by thinking of $A_y^{[\mu]}$ as valued in the Levi decomposition of $sl_N$ with respect to the minuscule coweight $\mu$, namely [61]

$$\begin{array}{rcl} sl_N & \to & \mathbf{n}_- \oplus \boldsymbol{l}_\mu \oplus \mathbf{n}_+ \\ \mathcal{A}^{[\mu]} & \sim & \mathcal{A}_{\mathbf{n}_-} + \mathcal{A}_{\boldsymbol{l}_\mu} + \mathcal{A}_{\mathbf{n}_+} \end{array} \tag{2.9}$$

Notice that this decomposition is due to the fact that the minuscule coweight $\mu$ acts on the Lie algebra elements with only three eigenvalues $0; \pm 1$. Therefore, a Lie algebra is decomposed to three subspaces; the $\boldsymbol{l}_\mu$ is a Levi subalgebra, and $\mathbf{n}_\pm$ are nilpotent subalgebras constrained as follows, with Levi charge $q = \pm 1$:

$$[\mu, \boldsymbol{l}_\mu] = 0 \quad , \quad [\mu, \mathbf{n}_q] = q\mathbf{n}_q \quad , \quad [\mathbf{n}_q, \mathbf{n}_q] = 0 \tag{2.10}$$

In these regards, notice that for the case of the topological $sl_N$ gauge theory, we can define $N - 1$ minuscule 't Hooft lines carrying different magnetic charges :

$$tH_{\gamma_{z_1}}^{\mu_1} \quad , \quad ... \quad , \quad tH_{\gamma_{z_{N-1}}}^{\mu_{N-1}} \tag{2.11}$$

They are in 1:1 correspondence with the $N - 1$ minuscule coweights $\mu_1, ..., \mu_{N-1}$ of the $sl_N$ Lie algebra of the gauge symmetry (as listed in (2.16)); and eventually with the $N - 1$ simple roots $\alpha_1, ..., \alpha_{N-1}$ of the Dynkin diagram of $sl_N$ as depicted in the Figure **3**.

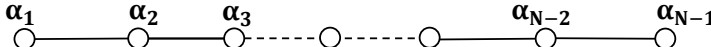

Figure 3: The Dynkin diagram for the $sl_N$ family, it has $N-1$ simple roots, all corresponding to minuscule coweights

In what follows, we focus our attention on the XXX spin chain construction in the framework of the 4D CS theory. As described in the figure 4, we need to take N vertical (parallel) Wilson lines $W_{\xi_z^i}^{\boldsymbol{R}}$ in the topological plane $\mathbb{R}^2$ travesrsed by a horizontal 't Hooft line $tH_{\gamma_0}^\mu$ (in red color). The $W_{\xi_z^i}^{\boldsymbol{R}}$s sit at the position $z \neq 0$ in the holomorphic plane while the $tH_{\gamma_0}^\mu$ is in $z = 0$. From the integrable spin chain point of view, every Wilson line presents a node of the chain and the 't Hooft line is interpreted as the Baxter Q-operator [52]. This way, we have a 't

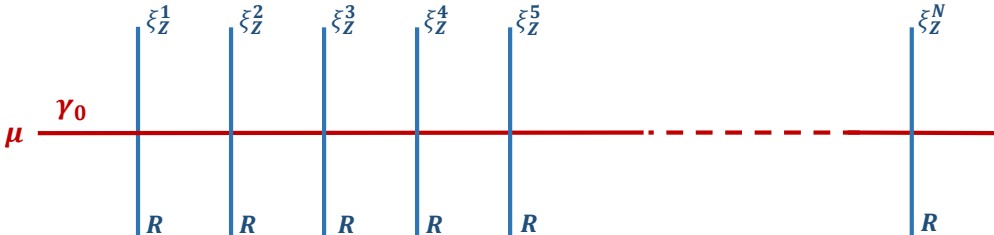

Figure 4: The spin chain configuration in the Chern-Simons theory: N Wilson lines represented by the blue vertical lines crossed by a 'tH$_\gamma^\mu$ represented by the red horizontal line.

Hooft-Wilson coupling in the topological plane at every node as depicted by the Figure. The interaction by line-crossing is interesting as it allows to define the Lax operator in every node of the spin chain which plays an important role in the study of integrable systems. Because $W_{\xi_z}^{\boldsymbol{R}}$ is characterised by $(\xi_z; \boldsymbol{R})$ and tH$_{\gamma_0}^\mu$ by $(\gamma_0; \mu)$, the coupling between them should carry all this data and can be defined as follows

$$L_{\boldsymbol{R}}^\mu(\gamma_0, \xi_z) = \left\langle tH_{\gamma_0}^\mu, W_{\xi_z}^{\boldsymbol{R}} \right\rangle \tag{2.12}$$

Following [52], this L-operator, denoted from now on like $\mathcal{L}_{\boldsymbol{R}}^\mu$, is precisely given by (2.8) such that the transport path is identified with the Wilson line. Moreover, it can be put into a simpler form using the Levi-like factorisation

$$\mathcal{L}_{\boldsymbol{R}}^\mu(z) = e^{X_{\boldsymbol{R}}} z^{\mu_R} e^{Y_{\boldsymbol{R}}} \tag{2.13}$$

where $X_{\boldsymbol{R}}$ is a nilpotent matrix valued in the nilpotent algebra $\mathbf{n}_+$, and $Y_{\boldsymbol{R}}$ is also a nilpotent matrix but valued in the nilpotent algebra $\mathbf{n}_-$. These matrices are constrained by the Levi decomposition requiring

$$[z^{\mu_R}, X_{\boldsymbol{R}}] = +X_{\boldsymbol{R}} \quad , \quad [z^{\mu_R}, Y_{\boldsymbol{R}}] = -Y_{\boldsymbol{R}} \tag{2.14}$$

## 2.2 Interacting tH$_{\gamma_0}^{\mu_i}$-$W_{\xi_z}^{\boldsymbol{R}}$ lines in CS theory

For the next step in the study of minuscule tH$_{\gamma_0}^{\mu_i}$ lines interacting with $W_{\xi_z}^{\boldsymbol{R}}$ in 4D CS theory, it is interesting to explore the algebraic structure of the magnetic charges $\mu_i$ of the tH$_{\gamma_0}^{\mu_i}$'s. As these charges are given by the minuscule coweights of the $sl_N$ Lie algebra, we give below some useful tools regarding their properties and then turn to study their coupling with $W_{\xi_z}^{\boldsymbol{R}}$.

### 2.2.1 Minuscule coweights of $sl_N$

First, we recall that there are $N - 1$ fundamental coweights $\omega_i$ for the $sl_N$ Lie algebra, they are defined as the algebraic dual of the $N - 1$ simple roots $\alpha_i$; which means that $\omega_i.\alpha_j = \delta_{ij}$.

These simple roots of $sl_N$ are realised in terms of a weight basis vectors $\langle e_i \rangle$ like $\alpha_i = e_i - e_{i+1}$. So, the fundamental coweights solving $\omega_i . \alpha_j = \delta_{ij}$ read in terms of the $e_i$'s as follows

$$\omega_i = \frac{N-i}{N}(e_1 + ... + e_i) - \frac{i}{N}(e_{i+1} + ... + e_N) \tag{2.15}$$

It turns out that in the case of the $sl_N$ Lie algebra, all the fundamental coweights are minuscule [61]. So, the magnetic charges of the $(N-1)$ lines $tH_{\gamma_0}^{\mu_i}$ of the $A_{N-1}$- CS theory are given by

$$\begin{array}{rcl}
\mu_1 & = & \frac{N-1}{N}e_1 - \frac{1}{N}(e_2 + ... + e_N) \\
\mu_l & = & \frac{N-l}{N}(e_1 + ... + e_l) - \frac{l}{N}(e_{l+1} + ... + e_N) \\
\mu_{N-1} & = & \frac{1}{N}(e_1 + ... + e_{N-1}) - \frac{N-1}{N}e_N
\end{array} \tag{2.16}$$

with $2 \leq l \leq N-2$. Obviously one can treat all these coweights collectively; but it is interesting to cast them as we have done.

As illustrating examples, we have for the $sl_2$ model, one minuscule charge $\mu = \frac{1}{2}(e_1 - e_2)$. For the $sl_3$ theory, we have two minuscule coweights $\mu_1 = \frac{2}{3}e_1 - \frac{1}{3}(e_2 + e_3)$ and $\mu_2 = \frac{1}{3}(e_1 + e_2) - \frac{2}{3}e_3$; and for the $sl_4$ CS theory, we have three minuscule charges given by

$$\begin{array}{rcl}
\mu_1 & = & \frac{3}{4}e_1 - \frac{1}{4}(e_2 + e_3 + e_4) \\
\mu_2 & = & \frac{1}{2}(e_1 + e_2) - \frac{1}{2}(e_3 + e_4) \\
\mu_3 & = & \frac{1}{4}(e_1 + e_2 + e_3) - \frac{3}{4}e_4
\end{array} \tag{2.17}$$

As far as the $sl_4$ example is concerned, notice that using the isomorphism $sl_4 \sim so_6$, these fundamental weights can be also viewed as the fundamental of $so_6$. Here, the $\mu_2$ corresponds to the vector of $so_6$ while the $\mu_1$ and $\mu_3$ correspond to the two Weyl spinors of orthogonal groups in even dimensions, they will be encountered later when we study the L-operators of D-type.

Notice also that given a minuscule coweight $\mu$ of $sl_N$, one defines its adjoint form by help of the $e_j^*$'s obeying $e_j^*(e_i) = \delta_i^j$. We denote the adjoint form of the coweight $\mu_l$ by the bold symbol $\boldsymbol{\mu}_l$ and express it as follows

$$\boldsymbol{\mu}_l = \frac{N-l}{N}\Pi_l - \frac{l}{N}\bar{\Pi}_l \tag{2.18}$$

with projector $\Pi_l$ and co-projector $\bar{\Pi}_l = I_{id} - \Pi_l$ as follows

$$\Pi_l = \sum_{i=1}^{l} e_i e_i^* \quad , \quad \bar{\Pi}_l = \sum_{i=l+1}^{N} e_i e_i^* \tag{2.19}$$

The use of this projector in the above decomposition is crucial in our modeling; it is at the basis of our way to approach the coupling between the $tH_{\gamma_0}^\mu$ and $W_{\xi_z}^{\boldsymbol{R}}$ as well as in the construction of the topological gauge quivers $Q_{\boldsymbol{R}}^\mu$ describing the A-type L-operators.

## 2.2.2 the $tH_{\gamma_0}^{\mu_i}$ - $W_{\xi_z}^R$ coupling

To properly define the coupling between $W_{\xi_z}^R$ and a given minuscule 't Hooft line $tH_{\gamma_0}^{\mu_k}$ with a magnetic charge $\mu_k$ in the 4D Chern-Simons theory living in $\mathbb{R}^2 \times \mathbb{CP}^1$, we follow [52] and proceed as summarised below:

(**i**) $tH_{\gamma_0}^{\mu_k}$ *as a horizontal magnetic defect in* $\mathbb{R}^2$

We think of the 't Hooft $tH_{\gamma_0}^{\mu_k}$ as the curve $\gamma_0$ extending in the topological plane $\mathbb{R}^2$ of the 4D space. The defect $\gamma_0$ is located at a given point $z$ in $\mathbb{CP}^1$ that we take as $z = 0$; say the south pole of $\mathbb{S}^2 \sim \mathbb{CP}^1$. For convenience, we think of $\gamma_0$ as the horizontal line given by the x-axis of the plane $\mathbb{R}^2$ with $(x, y)$ coordinates; see the red line in the Figure **4**. Topologically speaking, this $\gamma_0$ can be also imagined as the intersection of two patches like $\gamma_0 = \mathbb{R}_{y \leq 0}^2 \cap \mathbb{R}_{y \geq 0}^2$. Along with this $tH_{\gamma_0}^{\mu_k}$, we also have a $tH_{\gamma_\infty}^{-\mu_k}$ sitting at $z = \infty$ corresponding to the north pole of $\mathbb{S}^2$.

(**ii**) $tH_{\gamma_0}^{\mu_k}$ *crosses a vertical Wilson line*

The horizontal $tH_{\gamma_0}^{\mu_k}$ crosses a vertical Wilson line $W_{\xi_z}^R$ with $\xi_z$ located at a generic point $z$ of $\mathbb{CP}^1$. We imagine $\xi_z$ as coinciding with the y-axis in $\mathbb{R}^2$, i.e. $\xi_z = \{(x, y) \, | = x = 0, y \in \mathbb{R}\}$. Recall that the quantum states $|A\rangle$ propagating in the electrically charged line $W_{\xi_z}^R$ are in the fundamental $\mathbf{N}$ representation of $sl_N$. The incoming particle states are denoted by the bra $\langle A|$ and the outgoing states by the ket $|B\rangle$ with

$$\langle A|B\rangle = \delta_A^B \tag{2.20}$$

in the case of free propagation. In the presence of interaction, the above $\delta_A^B$ is replaced by a multi-label vertex object.

(**iii**) $\mathcal{L}$*-operator and phase space*

The crossing of the horizontal $tH_{\gamma_0}^{\mu_k}$ and the vertical $W_{\xi_z}^R$ lines is thought of in terms of lines' coupling described by the $\mathcal{L}$-operator (2.12) represented by the typical matrix operator

$$\left\langle A|\mathcal{L}_R^{(\mu)}|B\right\rangle = L_{AB}^{(\mu)} \tag{2.21}$$

This operator is equivalent to the usual Lax operator of integrable spin chain systems [18,66]. It is a *holomorphic* function of $z$ and its representative matrix $L_{AB}^{(\mu)}$ is valued in the algebra $\mathfrak{A}$ of functions on the phase space of $tH_{\gamma_z}^\mu$. Formally, we have

$$\mathcal{L}_R^\mu \in \mathfrak{A} \otimes End(\mathbf{N}) \tag{2.22}$$

with $\mathfrak{A}$ generated by Darboux coordinates $(b, c)$ to be commented later on; see eq(2.32). The phase space of the $L_j^i(z)$ operator is obtained by considering two coupled vertical Wilson lines $W_{\xi_z}^R$ and $W_{\xi_{z'}}^R$ crossed by a horizontal $tH_{\gamma_0}^{\mu_k}$ as depicted by the Figure **5**.

This topological invariant crossing describes integrability as encoded in the following RLL relations

$$L_j^r(z) R_{rs}^{ik}(z - z') L_l^s(z') = L_r^i(z) R_{jl}^{rs}(z - z') L_s^k(z') \tag{2.23}$$

In this equation, $R_{rs}^{ik}(z - z')$ is the well known R-operator appearing in the Yang-Baxter equation, it is proportional to the second Casimir $C_{rs}^{ik}$ of $sl_N$ having the value $\delta_r^i \delta_s^k$. For the trigonometric case corresponding to the holomorphic line $\mathbb{CP}^1$, the structure of this R-matrix as a series of $\hbar$ has leading terms like $R_{rs}^{ik}(z) = \delta_r^i \delta_s^k + \frac{\hbar}{z} C_{rs}^{ik} + O(\hbar^2)$.

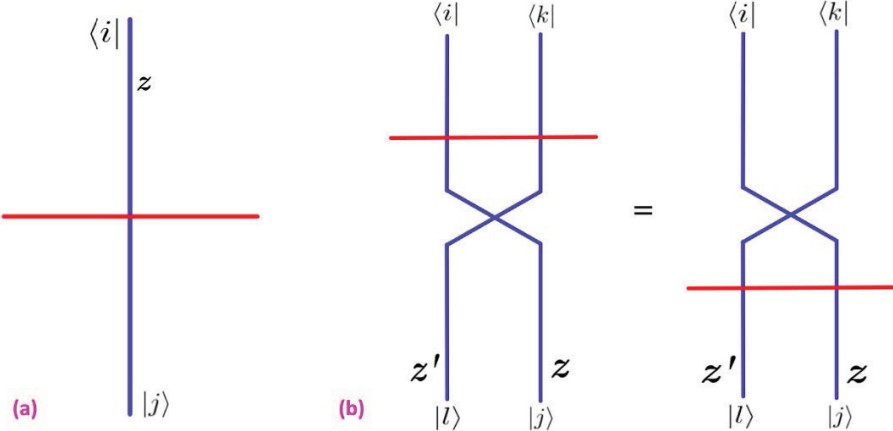

Figure 5: (**a**) The operator $\mathcal{L}(z)$ encoding the coupling between a 't Hooft line at z=0 (in red) and a Wilson line at z (in blue) with incoming $\langle i|$ and out going $|j\rangle$ states. (**b**) RLL relations encoding the commutation relations between two L-operators at z and z'.

### 2.2.3 Levi decomposition of $sl_N$

The RLL relations of eq(2.23) can be shown to be equivalent to the usual Poisson bracket $\{b^\alpha, c_\beta\}_{PB} = \delta^\alpha_\beta$ of symplectic geometry with $b^\alpha$ and $c_\beta$ as phase space coordinates (Darboux coordinates). This equivalence between the $L^i_j$ bracket (eq(2.23)) and the $\{b^\alpha, c_\beta\}_{PB}$ follows from the Levi decompositions of $sl_N$ that we describe here for different coweights of eq(2.17).

    **1)** *Minuscule coweight $\mu_1$*

The Levi decomposition of $sl_N$ and its fundamental representation $\boldsymbol{N}$ with respect to the minuscule coweight $\mu_1$ reads as follows

$$
\begin{aligned}
\mu_1 \;\; : \;\; sl_N \;\; &\rightarrow \;\; sl_1 \oplus sl_{N-1} \oplus \mathbf{n}_+ \oplus \mathbf{n}_- \\
\boldsymbol{N} \;\; &\rightarrow \;\; \mathbf{1}_{\frac{N-1}{N}} \oplus (\boldsymbol{N-1})_{-\frac{1}{N}}
\end{aligned}
\tag{2.24}
$$

with $\mathbf{n}_\pm = (N-1)_\pm$. Because of this decomposition of $sl_N$, one can imagine the Levi sub-algebra as the manifest invariance in dealing with the study of the $tH^{\mu_1}_{\gamma_0}$ lines in the CS gauge theory with $sl_N$ gauge symmetry. In this view, we use the projectors $\varrho_{\mathbf{1}}$ and $\varrho_{\boldsymbol{N-1}}$ of the irreducible parts of the decomposition $\boldsymbol{N} = \mathbf{1}_{\frac{N-1}{N}} \oplus (\boldsymbol{N-1})_{-\frac{1}{N}}$ as well as the identity $\varrho_{\mathbf{1}} + \varrho_{\boldsymbol{N-1}} = I_{id}$ to think of the adjoint form $\mu_1$ of the minuscule coweight as the sum of two contributions, one coming from $\mathbf{1}_{1-1/N}$ and the other from $(\boldsymbol{N-1})_{-1/N}$ like

$$
\mu_1 = \mu_1 \varrho_{\mathbf{1}} + \mu_1 \varrho_{\boldsymbol{N-1}}
\tag{2.25}
$$

The projectors $\varrho_{\boldsymbol{R}}$ appearing in the above relation are as in eqs(2.18-2.19). In this picture the 't Hooft line of the $sl_N$ gauge symmetry gets splited into two parallel "sub-lines" as represented in the Figure 6. This is our first result regarding the using the projector basis to understand the intrinsic properties of the L-operator in the A-series. Clearly, the two 't Hooft "sub-lines" in the Figure **6**-(b) are coincident in the external space $\mathbb{R}^2 \times \mathbb{CP}^1$ of the CS theory, but are lifted in the $sl_N$ internal space where the transitions between the two sublines

are generated by operators belonging to the nilpotent subalgebras $\mathbf{n}_\pm$.

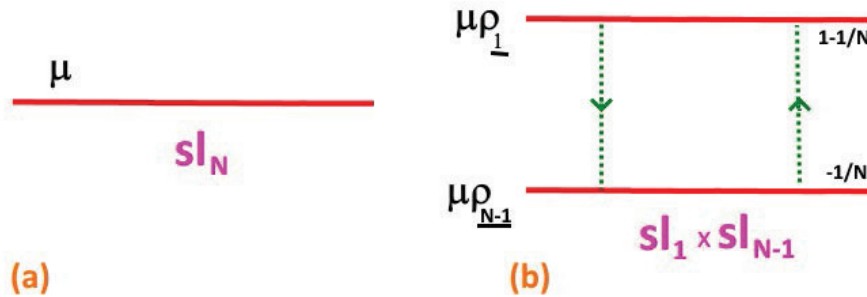

Figure 6: (**a**) Magnetic 'tHooft with charge $\mu_1$ from the point of view of global $sl_N$ symmetry. (**b**) The same line from the point of view of internal $sl_1 \oplus sl_{N-1}$. here, the line $\mu$ splits into two sublines $\mu\varrho_1$ and $\mu\varrho_{N-1}$ as described in eq(2.25).

Moreover, the decomposition $\boldsymbol{N} \to \mathbf{1}_{1-1/N} \oplus (\boldsymbol{N}-\mathbf{1})_{-1/N}$ can be extended to higher dimensional representations $\boldsymbol{R}$ of the $sl_N$ gauge symmetry. This extension follows with the previous discussion concerning $W^{\boldsymbol{R}}_{\xi_z}$ beyond the fundamental weight $\boldsymbol{N}$. For example, the antisymmetric $\boldsymbol{N} \wedge \boldsymbol{N}$, the symmetric $\boldsymbol{N} \vee \boldsymbol{N}$ and the $\boldsymbol{adj}$ representations of $sl_N$ decompose with respect to the minuscule coweight $\mu_1$ as follows

| $sl_N$ | $sl_1 \oplus sl_{N-1}$ |
|---|---|
| $\boldsymbol{N} \wedge \boldsymbol{N}$ | $\boldsymbol{F}_{1-\frac{2}{N}} \oplus (\boldsymbol{F} \wedge \boldsymbol{F})_{-\frac{2}{N}}$ |
| $\boldsymbol{N} \vee \boldsymbol{N}$ | $\mathbf{1}_{2-\frac{2}{N}} \oplus \boldsymbol{F}_{1-\frac{2}{N}} \oplus (\boldsymbol{F} \vee \boldsymbol{F})_{-\frac{2}{N}}$ |
| $\boldsymbol{N} \times \bar{\boldsymbol{N}}$ | $\mathbf{1}_{1-\frac{1}{N}}\bar{\mathbf{1}}_{\frac{1}{N}-1} + \mathbf{1}_{1-\frac{1}{N}}\bar{\boldsymbol{F}}_{\frac{1}{N}} + \boldsymbol{F}_{-\frac{1}{N}}\bar{\mathbf{1}}_{\frac{1}{N}-1} + \boldsymbol{F}_{-\frac{1}{N}}\bar{\boldsymbol{F}}_{\frac{1}{N}}$ |
| $\boldsymbol{adj}\,(sl_N)$ | $\boldsymbol{F}_{-1} \oplus \left[\mathbf{1}_0 \oplus \boldsymbol{adj}\,(sl_{N-1})_0\right] \oplus \boldsymbol{F}_{+1}$ |

(2.26)

where we have set $\boldsymbol{F} = \boldsymbol{N} - \mathbf{1}$. Notice also that compared to $\boldsymbol{N} \to \mathbf{1}_{1-1/N} \oplus (\boldsymbol{N}-\mathbf{1})_{-1/N}$, the symmetric $\boldsymbol{N} \vee \boldsymbol{N}$ reduces to three $sl_1 \oplus sl_{N-1}$ representations namely $\mathbf{1}_{2-\frac{2}{N}}$ and $\boldsymbol{F}_{1-\frac{2}{N}}$ as well as $(\boldsymbol{F} \vee \boldsymbol{F})_{-\frac{2}{N}}$; the same holds for $\boldsymbol{adj}\,(sl_N)$. This feature is interesting as it indicates that the corresponding Lax operators $\mathcal{L}_{\boldsymbol{N}\vee\boldsymbol{N}}$ and $\mathcal{L}_{adj(sl_N)}$ have a richer intrinsic structure compared to $\mathcal{L}_{\boldsymbol{N}}$, see subsection 2.3.

**2)** *Minuscule coweights $\mu_k$ for $2 \leq k \leq N-2$.*
Levi decompositions of $sl_N$ and its fundamental representation with respect to $\mu_k$ read as follows

$$
\begin{aligned}
\mu_k \quad : \quad sl_N \quad &\to \quad sl_k \oplus sl_{N-k} \oplus sl_1 \oplus k\,(N-k)_+ \oplus k\,(N-k)_- \\
\boldsymbol{N} \quad &\to \quad \boldsymbol{k}_{\frac{N-k}{N}} \oplus (\boldsymbol{N}-\boldsymbol{k})_{-\frac{k}{N}}
\end{aligned}
$$

(2.27)

where the Levi subalgebra is $sl_k \oplus sl_{N-k} \oplus sl_1$ and the nilpotent subalgebras are $\boldsymbol{k}\,(\boldsymbol{N}-\boldsymbol{k})_\pm$. For the example of $sl_4$ with $k=2$, we have

$$
\begin{aligned}
\mu_2 \quad : \quad sl_4 \quad &\to \quad sl_2 \oplus sl_2 \oplus sl_1 \oplus 4_+ \oplus 4_- \\
4 \quad &\to \quad 2_{+\frac{1}{2}} \oplus 2_{-\frac{1}{2}}
\end{aligned}
$$

(2.28)

Notice that for this case as well, we have a splitting picture as in the Figure **6**-(b) where eq(2.25) should be replaced by

$$\mu_k = \mu_k \varrho_{\mathbf{k}} + \mu_k \varrho_{\mathbf{N-k}} \tag{2.29}$$

**3**) *Minuscule coweight $\mu_{N-1}$*

The Levi- decomposition of $sl_N$ with respect to $\mu_{N-1}$ reads as follows

$$\begin{array}{rccc} \mu_{N-1} & : & sl_N & \to & sl_{N-1} \oplus sl_1 \oplus F_+ \oplus F_- \\ & & N & \to & 1_{\frac{N-1}{N}} \oplus (N-1)_{-\frac{1}{N}} \end{array} \tag{2.30}$$

It has a similar structure to eq(2.24), so we can omit the details regarding this $\mu_{N-1}$ case; it can also be recovered from the generic $\mu_k$ with $k = N - 1$.

## 2.3   the $\mathcal{L}$-operators in $sl_N$ theory

The expression of the $\mathcal{L}^{\mu_k}$-operator in terms of the adjoint form of the minuscule coweight $\mu_k$ and the Darboux coordinates $b^a$ and $c_a$ is given by

$$\mathcal{L}^{\mu_k}(z) = e^X z^{\mu_k} e^Y \tag{2.31}$$

with

$$X = \sum_{a=1}^{k(N-k)} b^a X_a \qquad , \qquad Y = \sum_{a=1}^{k(N-k)} c_a Y^a \tag{2.32}$$

In eq(2.31), the minuscule coweight acts like

$$[\mu_k, X_a] = +X_a \qquad , \qquad [\mu_k, Y^a] = -Y^a \tag{2.33}$$

with the adjoint action $\mu_k = \mu_k^i \varrho_i$ where $\varrho_i = |i\rangle \langle i|$ and where the $\mu_k^i$'s are fractions of the unity given by (2.16). See also the Figure **7**-(a,b) representing our vision regarding the topology of the L-operators of A-type series. For the expressions of the generators $X_a$ and $Y^a$ solving the constraints of eq(2.33), they are constructed below depending on the value of the level $k$.

### 2.3.1   't Hooft line with magnetic charge $\mu_1$

In the case of a 't Hooft line with a magnetic charge $\mu_1$ crossing a Wilson line $W_{\xi_z}^{\boldsymbol{R}=N}$ of $sl_N$, we have $N - 1$ generators $X_a$ and $N - 1$ generators $Y^a$ in the fundamental representation. These are $N \times N$ triangular matrices solving eq(2.33) and given by

$$\begin{array}{rcl} X_a & = & |1\rangle \langle a+1| \\ Y^a & = & |a+1\rangle \langle 1| \\ \mu_1 & = & \frac{N-1}{N} \varrho_1 - \frac{1}{N} (\varrho_2 + ... + \varrho_N) \end{array} \tag{2.34}$$

where we have set $\varrho_i = |i\rangle \langle i|$ with $\sum_{i=1}^N \varrho_i = I_{N \times N}$. Moreover, by taking $\varrho_{\bar{1}} = \varrho_2 + ... + \varrho_N$ with $\varrho_1 + \varrho_{\bar{1}} = I$, the adjoint form $\mu_1$ can be written in the following form

$$\mu_1 = \frac{N-1}{N} \varrho_1 - \frac{1}{N} \varrho_{\bar{1}} \tag{2.35}$$

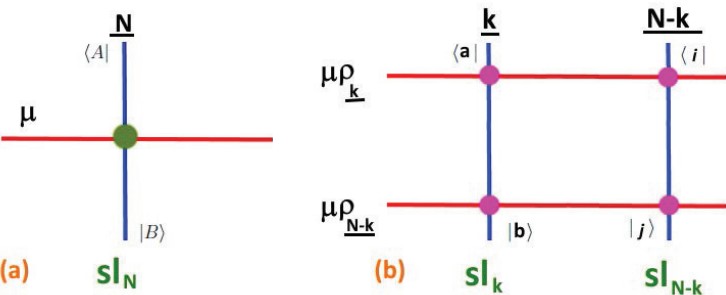

Figure 7: (a) A horizontal minuscule 't Hooft line with magnetic charge $\mu_k$ crossing a vertical Wilson line with electric charge $\boldsymbol{R} = \mathbf{N}$. The green dot describes the coupling given by the Lax operator $\langle A|L_{\boldsymbol{R}}^{\mu_k}|B\rangle$. (b) Intrinsic structure of the Lax operator taking into account the Levi decomposition of $sl_N$ with respect to $\mu_k$.

These projectors play an important role in the study of the L-operator of the 4D CS theory with $SL_N$ gauge invariance: (**1**) They single out the Levi charges of the two internal subspaces in the Levi decomposition $\boldsymbol{N} = \mathbf{1}_{1-1/N} \oplus (\boldsymbol{N}-\mathbf{1})_{-1/N}$. For example, by multiplying eq(2.35) first by $\varrho_1$ and then by $\varrho_{\bar{1}}$, we obtain

$$\mu_1 \varrho_1 = \frac{N-1}{N}\varrho_1 \quad , \quad \mu_1 \varrho_{\bar{1}} = -\frac{1}{N}\varrho_{\bar{1}} \tag{2.36}$$

which describe the two horizontal sublines in the Figure **6**-(b). (**2**) They allow to write interesting properties verified by the realisation (2.34) such as

$$\begin{aligned} X_a \varrho_1 &= 0 & , & \varrho_1 Y^a &= 0 \\ \varrho_{\bar{1}} X_a &= 0 & , & Y^a \varrho_{\bar{1}} &= 0 \end{aligned} \tag{2.37}$$

indicating that $\mathcal{L}_{\boldsymbol{R}}^{\mu_1}$ can be presented as a matrix with sub-blocks given in terms of the projectors $\varrho_1$ and $\varrho_{\bar{1}}$.

We can check the relations (2.33) by computing the quantities $\mu_1 X_a$ and $X_a \mu_1$ using the above realisation, we have

$$\mu_1 X_a = \frac{N-1}{N}X_a \quad , \quad X_a \mu_1 = -\frac{1}{N}X_a \tag{2.38}$$

thus giving $[\mu_1, X_a] = X_a$; the same can be done for the generators $Y^a$.

Now, in order to explicitly calculate the L-operator $\mathcal{L}_{\boldsymbol{R}}^{\mu_1}(z) = e^X z^{\mu_1} e^Y$, we need to evaluate the exponentials $e^X$ and $e^Y$ such that X and Y are given by

$$X = \sum_{a=1}^{N-1} b^a |1\rangle\langle a+1| \quad , \quad Y = \sum_{a=1}^{N-1} c_a |a+1\rangle\langle 1| \tag{2.39}$$

These matrices obey the property $X^2 = Y^2 = 0$. Because of this nilpotency, we can write $e^X = I + X$ and $e^Y = I + Y$, and consequently

$$\begin{aligned} \mathcal{L}_{\boldsymbol{R}}^{\mu_1}(z) &= (I+X)z^{\mu_1}(I+Y) \\ &= z^{\mu_1}I + Xz^{\mu_1} + z^{\mu_1}Y + Xz^{\mu_1}Y \end{aligned} \tag{2.40}$$

Using $\mu_1 = \frac{N-1}{N}\varrho_1 - \frac{1}{N}\varrho_{\bar{1}}$ with $\varrho_{\bar{1}} = I - \varrho_1$ and $z^{\mu_1} = z^{\mu_k^i}\varrho_i$, we can express the L-operator in terms of projectors as follows

$$
\begin{aligned}
\mathcal{L}_{\boldsymbol{R}}^{\mu_1}(z) \quad = \quad & z^{\frac{N-1}{N}}\varrho_1 + z^{-\frac{1}{N}}X\varrho_{\bar{1}}Y + \\
& z^{-\frac{1}{N}}\left(X\varrho_{\bar{1}} + \varrho_{\bar{1}}Y\right) + z^{-\frac{1}{N}}\varrho_{\bar{1}}
\end{aligned}
\tag{2.41}
$$

This form of the Lax operator is a result of the projectors basis that we choose above, this unique writing is particularly significant for the quiver description of the L-operator as well as for the straightforward extension to other electric charges of the 4D CS gauge theory with $sl_N$ gauge symmetry.

### 2.3.2 Magnetic charge $\mu_k$ with $2 \le k \le N-2$

In this generic case, we have $k(N-k)$ generators $X_\alpha$ and $k(N-k)$ generators $Y^\alpha$ generating the nilpotent subalgebras $\boldsymbol{k(N-k)}_+$ and $\boldsymbol{k(N-k)}_-$ of the Levi decomposition of $sl_N$ with respect to the minuscule coweight $\mu_k$. In fact, the $X_{i\alpha}$ and the $Y^{i\alpha}$ of $\mathbf{n}_\pm$ are $N \times N$ triangular matrices realised as follows

$$
\begin{aligned}
X_{i\alpha} &= |i\rangle\langle k+\alpha| \quad , \quad 1 \le i \le k \\
Y^{i\alpha} &= |k+\alpha\rangle\langle i| \quad , \quad 1 \le \alpha \le N-k
\end{aligned}
\tag{2.42}
$$

and the $\mu_k$ is given by

$$
\mu_k = \frac{N-k}{N}\Pi_k - \frac{k}{N}\Pi_{\bar{k}}
\tag{2.43}
$$

with

$$
\Pi_k = \sum_{l=1}^{k}\varrho_l \quad , \qquad \Pi_{\bar{k}} = \sum_{l=k+1}^{N}\varrho_l
\tag{2.44}
$$

The generators (2.42) satisfy the Levi decomposition conditions that read as

$$
\begin{aligned}
{[\mu_k, X_{ia}]} &= \left(\frac{N-l}{N} + \frac{l}{N}\right)X_{ia} &= X_{ia} \\
{[\mu_k, Y^{ia}]} &= \left(-\frac{l}{N} - \frac{N-l}{N}\right)Y^{ia} &= -Y^{ia}
\end{aligned}
\tag{2.45}
$$

This interesting realisation also obeys

$$
X_a\Pi_k = 0 \quad , \qquad \Pi_k Y^a = 0
\tag{2.46}
$$

which indicates the sub-blocks of the matrix $\mathcal{L}_{\boldsymbol{R}}^{\mu_k}$. The commutators $[X_{ia}, Y^{ia}]$ give the Cartan generators reading as $H_{ia} = \varrho_i - \varrho_a$ while the nilpotency $X_{i\alpha}X_{j\beta} = Y^{i\alpha}Y^{j\beta} = 0$ leads to $e^X = I + X$ and $e^Y = 1 + Y$. Using these features, we obtain

$$
\begin{aligned}
\mathcal{L}_{\boldsymbol{R}}^{\mu_k} \quad &= \quad (I+X)\,z^{\mu_k}\,(I+Y) \\
&= \quad z^{\mu_k}I + Xz^{\mu_k} + z^{\mu_k}Y + Xz^{\mu_k}Y
\end{aligned}
\tag{2.47}
$$

Moreover, using

$$
\mu_k = \frac{N-k}{N}\Pi_k - \frac{k}{N}\Pi_{\bar{k}}
\tag{2.48}
$$

with $\Pi_{\bar{k}} = I - \Pi_k$ and $z^{\mu_1} = z^{\mu_k^k} \Pi_k + z^{\mu_k^{\bar{k}}} \Pi_{\bar{k}}$, we can express the operator $\mathcal{L}^{(\mu_k)}$ in terms of the projectors as follows.

$$
\begin{aligned}
\mathcal{L}_{\mathbf{R}}^{\mu_k}(z) = \ & z^{\frac{N-k}{N}} \Pi_k + z^{-\frac{k}{N}} \Pi_{\bar{k}} + \\
& z^{-\frac{k}{N}} X \Pi_{\bar{k}} + z^{-\frac{k}{N}} \Pi_{\bar{k}} Y + z^{-\frac{k}{N}} X \Pi_{\bar{k}} Y
\end{aligned}
\tag{2.49}
$$

This is the generic form of the L-operator in the 4D Chern-Simons gauge theory with $sl_N$ gauge invariance.

### 2.3.3 Magnetic charge $\mu_{N-1}$

In this case, the $N-1$ generators $X_i$ and $N-1$ generators $Y^a$ are given by

$$
X_a = |1 + a\rangle \langle N| \qquad , \qquad Y^a = |N\rangle \langle 1 + a|
\tag{2.50}
$$

with $1 \leq a \leq N - 1$ and

$$
\mu_{N-1} = \frac{1}{N} \varrho_{\overline{N}} - \frac{N-1}{N} \varrho_N
\tag{2.51}
$$

The Lax operator reads as

$$
\mathcal{L}_{\mathbf{R}}^{\mu_{N-1}}(z) = z^{\frac{1}{N}} \varrho_{\overline{N}} + z^{\frac{1-N}{N}} \varrho_N + z^{\frac{1-N}{N}} X \varrho_N + z^{\frac{1-N}{N}} \varrho_N Y + z^{\frac{1-N}{N}} X \varrho_N Y
\tag{2.52}
$$

which corresponds to setting $k = N - 1$ in eq(2.49).

# 3 Topological gauge quivers: A- family

In this section, we introduce our quiver gauge representation for topological Lax operators in the 4D Chern-Simons theory. This graphical description was first proposed in [59] for the case of exceptional gauge symmetries $E_{6,7}$, it will be extended here for the ADE Lie algebras. We begin by giving a definition of these graphs, then we construct the topological quivers $Q_{\mathbf{R}}^{\mu_k}$ corresponding to the L-operators $\mathcal{L}_{\mathbf{R}}^{\mu_k}$ of $sl_N$ -type with $\mu_k$, $1 \leq k \leq N$ and $\mathbf{R} = \mathbf{N}$. We exploit this leading model to graphically describe the structure of Lax operators corresponding to representations of $sl_N$ beyond the fundamental $\mathbf{N}$. These $sl_N$ quivers are collectively listed in the Figure 29 in the conclusion section.

## 3.1 Motivating the topological quivers $Q_{\mathbf{R}}^{\mu_k}$

The quiver diagrams $Q_{\mathbf{R}}^{\mu_k}$ we want to construct in this section give a unified graphical representation of the L-operators of A-type. We refer to these graphs as topological gauge quivers; first because they concern a topological gauge theory (Chern-Simons), and second because they have a formal similarity with quiver diagrams $Q_{gauge}^{susy}$ in supersymmetric quiver gauge theories that we briefly recall here below.

- *Gauge quivers in supersymmetric theory*

In a supersymmetric quiver gauge theory with unitary gauge symmetry $G$ factorised as

$$G = \prod_{i=1}^{n_0} U(M_i) \tag{3.53}$$

and Lie algebra like $\boldsymbol{g} = \oplus_{i=1}^{n_0} u(M_i)$, where the gauge symmetry factors are imagined in type II strings as stacks of $M_i$ coincident D branes wrapping cycles in Calabi-Yau compactifications, we have a gauge quiver that denoted like $Q_{gauge}^{susy}$. It has: (**i**) $n_0$ nodes $\mathcal{N}_1, ..., \mathcal{N}_{n_0}$ corresponding to the gauge group factors $G_1, ..., G_{n_0}$ describing "adjoint matter" in the gauge theory transforming in the adjoint representations

$$adjU(M_i) = \boldsymbol{M}_i \times \bar{\boldsymbol{M}}_i \tag{3.54}$$

(**ii**) a number $n_{link}$ of links $L_{ij}$ between the nodes $(\mathcal{N}_i, \mathcal{N}_j)$ describing bi-fundamental matter transforming in the representations

$$\boldsymbol{M}_i \times \bar{\boldsymbol{M}}_j \in U(M_i) \times U(M_j) \tag{3.55}$$

- *Topological gauge quivers in 4D CS*

Based on the general aspects of supersymmetric quivers, we introduce our topological gauge quiver diagrams $Q_{\boldsymbol{R}}^{\mu_k}$ describing the L-operator in the topological 4D Chern Simons theory with A-type symmetry. These have similar features with $Q_{gauge}^{susy}$ that allow to interpret the phase space coordinates $b^a$ and $c_a$ in terms of topological variables and bi-fundamental matter. The topological property is inherited from the topological nature of line defects in 4D CS following from eqs.(2.31,2.32). In fact, By using the killing form of the $sl_N$ Lie algebra, we can write $b^a = tr(XY^a)$; and then by substituting $X = \log(\mathcal{L}^{\mu_k} e^{-Y} z^{-\mu_k})$, we end up with the following relation between $b^a$ and the topological line defect

$$b^a = tr\left(\log\left(\mathcal{L}^{\mu_k} e^{-Y} z^{-\mu_k}\right) Y^a\right)$$

Similar calculations for $c_a$ lead to

$$c_a = tr\left(\log\left(z^{-\mu_k} e^{-X} \mathcal{L}^{\mu_k}\right) X_a\right)$$

Concerning the interpretation of $b^a$ and $c_a$ as bi-fundamental matter, it follows from the decomposition of the gauge potential $\mathcal{A}^{[\mu]}$ in the Lie algebra $(\mathcal{A}^{[\mu]} \sim adj_{sl_N})$. From eq.(2.27), we have

$$
\begin{array}{ccccccccccccc}
sl_N & \to & sl_k & \oplus & sl_{N-k} & \oplus & sl_1 & \oplus & n_+ & \oplus & n_- & \\
adj_{sl_N} & \to & adj_{sl_k} & \oplus & adj_{sl_{N-k}} & \oplus & adj_{sl_1} & \oplus & (k, \overline{N-k}) & \oplus & (\bar{k}, N-k) & \\
\mathcal{A}^{[\mu]} & \to & \mathcal{A}_{sl_k} & \oplus & \mathcal{A}_{sl_{N-k}} & \oplus & \mathcal{A}_{sl_1} & \oplus & \{b^a\} & \oplus & \{c_a\} &
\end{array} \tag{3.56}
$$

where we learn that $b^a$ and $c_a$ sit in the bi-fundamental of the gauge symmetry $SL_k \times SL_{N-k}$. We can therefore associate to this symmetry a quiver diagram having ($i$) two nodes $\mathcal{N}_1$ and $\mathcal{N}_2$

respectively given by the adjoint $k \times \bar{k}$ and $(N-k) \times \overline{(N-k)}$ and $(ii)$ Two links $L_{12}$ and $L_{21}$ corresponding to the bifundamentals $(k, \overline{N-k})$ and $(\bar{k}, N-k)$. This quiver is constructed below (see the Figure 9). For the leading value $k = 1$ corresponding to $SL_1 \times SL_{N-1}$ , the topological quiver $Q_{\boldsymbol{R}}^{\mu_1}$ with electric Wilson line in the representation $\boldsymbol{R} = \boldsymbol{N}$ would have two nodes

$$\mathcal{N}_1 = (\mathbf{1} \times \bar{\mathbf{1}})_0 \qquad , \qquad \mathcal{N}_2 = (\boldsymbol{F} \times \bar{\boldsymbol{F}})_0 \tag{3.57}$$

where $\boldsymbol{F} = \boldsymbol{N} - 1$, and two links

$$L_{12} = \mathbf{1}_{1-1/N} \otimes \bar{\boldsymbol{F}}_{+1/N} \qquad , \qquad L_{21} = \boldsymbol{F}_{-1/N} \otimes \bar{\mathbf{1}}_{-1+1/N} \tag{3.58}$$

The group representation structure of the nodes and the links follows from the reduction of $\boldsymbol{N} \times \bar{\boldsymbol{N}}$ of the electric Wilson line into irreducible representations of $sl_1 \oplus sl_N$ as

$$\boldsymbol{N} \times \bar{\boldsymbol{N}} \quad \rightarrow \quad \begin{aligned} & \left(\mathbf{1}_{1-1/N} \times \bar{\mathbf{1}}_{-1+1/N}\right) + \left(\boldsymbol{F}_{-1/N} \times \bar{\boldsymbol{F}}_{+1/N}\right) + \\ & \left(\mathbf{1}_{1-1/N} \times \bar{\boldsymbol{F}}_{+1/N}\right) + \left(\boldsymbol{F}_{-1/N} \times \bar{\mathbf{1}}_{-1+1/N}\right) \end{aligned} \tag{3.59}$$

To demonstrate the explicit construction of the quivers $Q_{sl_N}^{\mu_k}$ associated to L-operators of the $SL_N$ CS theory, we begin by treating the particular case $k = 1$ to present the key idea, then we consider the generic $k \in [2, N-2]$ case.

## 3.2   Topological quiver of $\mathcal{L}_{\boldsymbol{R}}^{\mu_1}$

Thanks to the algebraic properties of the L-operator $\mathcal{L}_{\boldsymbol{R}}^{\mu_1}$ (2.41), in particular the special features induced from relations of the type $X\varrho_1 = 0$ and $\varrho_1 Y = 0$, and from identities like $X\bar{\varrho}_1 = X$ and $\bar{\varrho}_1 Y = Y$, the expression (2.41) of the operator $\mathcal{L}_{\boldsymbol{R}}^{\mu_1}$ reduces to the simple form

$$\mathcal{L}_{\boldsymbol{R}}^{\mu_1} = z^{-\frac{1}{N}} (I + X + Y + XY) \varrho_{\bar{1}} + z^{\frac{N-1}{N}} \varrho_1 \tag{3.60}$$

In matrix language, it reads simply as

$$\mathcal{L}_{\boldsymbol{R}}^{\mu_1} = z^{-\frac{1}{N}} \begin{pmatrix} I_{N-1} + XY & X \\ Y & z \end{pmatrix} \tag{3.61}$$

To derive this expression, we have also used $X\bar{\varrho}_1 Y = XY$. The above matrix relation is the well known form of $\mathcal{L}_{\boldsymbol{R}}^{\mu_1}$ in literature [17].

### 3.2.1   The L-operator in the projector basis

Here, we develop a method to cast the L-operator $\mathcal{L}_{\boldsymbol{R}}^{\mu_1}$ into pieces characterised by representations of the Levi subalgebra $\boldsymbol{l}_{\mu_1}$. The expression (3.60) involves the projectors $\varrho_1$ and $\bar{\varrho}_1$ on the representations of the Levi subalgebra $sl_1 \oplus sl_{N-1}$. The presence of these projectors in the explicit construction of $\mathcal{L}_{\boldsymbol{R}}^{\mu_1}$ is interesting in the sense that it can be remarkably presented into a four sub-block matrix as follows

$$\mathcal{L}_{\boldsymbol{R}}^{\mu_1} = \begin{pmatrix} [z^{\frac{N-1}{N}} + z^{-\frac{1}{N}} XY]\varrho_1 & z^{-\frac{1}{N}} X \varrho_{\bar{1}} \\ z^{-\frac{1}{N}} \varrho_{\bar{1}} Y & z^{-\frac{1}{N}} \varrho_{\bar{1}} \end{pmatrix} \tag{3.62}$$

Moreover, due to the properties of the realisation of the nilpotent generators $X_\alpha$ and $Y^\alpha$ (2.34), the above expression reads also as

$$\mathcal{L}_{\boldsymbol{R}}^{\mu_1} = \begin{pmatrix} \varrho_1[z^{\frac{N-1}{N}} + z^{-\frac{1}{N}}XY]\varrho_1 & z^{-\frac{1}{N}}\varrho_1 X \varrho_{\bar 1} \\ z^{-\frac{1}{N}}\varrho_{\bar 1}Y\varrho_1 & z^{-\frac{1}{N}}\varrho_{\bar 1}\varrho_{\bar 1} \end{pmatrix} \tag{3.63}$$

thus opening a window on a formal similarity between the structure of $\mathcal{L}_{\boldsymbol{R}}^{\mu_1}$ and known graphs in supersymmetric gauge theory, especially in supersymmetric quiver gauge theories embedded in type II strings [62]. There, properties of quiver gauge theories are encoded in a graph $Q_{gauge}^{susy}$ with nodes $\mathcal{N}_i$ and links $L_{ij}$ having interpretations in terms of BPS particles, brane wrapping cycles and singularity [63]. In our situation concerning 4D CS theory, the proposed quiver $Q_{\boldsymbol{R}}^{\mu_1}$ shares general aspects with $Q_{gauge}^{susy}$, and is termed as a topological quiver because the underlying 4D CS theory is a topological theory and also due to other related features including gauge symmetry and gapless massless particle states [64,65]. The topological quiver $Q_{\boldsymbol{R}}^{\mu_1}$ for the A- family with $\boldsymbol{R} = \boldsymbol{N}$ has two nodes $\mathcal{N}_i$ and two links $L_{ij}$ thought of as

$$\begin{array}{llll} \mathcal{N}_1 &= \langle \varrho_1 \mathcal{L} \varrho_1 \rangle &, & L_{1\bar 1} &= \langle \varrho_1 \mathcal{L} \varrho_{\bar 1} \rangle \\ \mathcal{N}_{\bar 1} &= \langle \varrho_{\bar 1} \mathcal{L} \varrho_{\bar 1} \rangle &, & L_{\bar 1 1} &= \langle \varrho_{\bar 1} \mathcal{L} \varrho_1 \rangle \end{array} \tag{3.64}$$

By exhibiting the dependence into the Darboux coordinates $b^\alpha$ and $c_\alpha$, we can put the $\mathcal{L}_{\boldsymbol{R}}^{\mu_1}$ into an interesting form where $b^\alpha$ and $c_\alpha$ can be interpreted in terms of topological bi-fundamental matter of $SL_1 \times SL_{N-1}$. Using the Killing form, we can define these topological bi-matter in terms of the links $L_{1\bar 1}$ and $L_{\bar 1 1}$ as

$$b^\alpha = z^{\frac{1}{N}} Tr\left(L_{1\bar 1}Y^\alpha\right) \qquad , \qquad c_\alpha = z^{\frac{1}{N}} Tr\left(L_{\bar 1 1}X_\alpha\right) \tag{3.65}$$

This QFT interpretation of $b^\alpha$ and $c_\alpha$ is borrowed from the above mentioned supersymmetric quiver gauge theories. This aspect regarding $b^\alpha$ and $c_\alpha$ can be explicitly exhibited by replacing $X = b^\alpha X_a$ and $Y = c_a Y^\alpha$ as well as $XY = (b^\alpha c_a)\varrho_1$, we end up with the known $\mathcal{L}_{\boldsymbol{R}}^{\mu_1}$ [17] expressed here in a condensed form using projectors as

$$\mathcal{L}_{\boldsymbol{R}}^{\mu_1} = \begin{pmatrix} [z^{\frac{N-1}{N}} + z^{-\frac{1}{N}}\mathbf{b}^T\mathbf{c}]\varrho_1 & z^{-\frac{1}{N}}\mathbf{b}^T\varrho_{\bar 1} \\ z^{-\frac{1}{N}}\varrho_{\bar 1}\mathbf{c} & z^{-\frac{1}{N}}\varrho_{\bar 1} \end{pmatrix} \tag{3.66}$$

In this oscillator realisation, the $b^\alpha$ and $c_\alpha$ appear indeed as fundamental quantities stretching between the two nodes and $\mathbf{b}^T\mathbf{c}$ as topological adjoint matter of the Levi subsymmetry group $SL_1 \times SL_{N-1}$.

### 3.2.2 Formal expression of $\mathcal{L}_{sl_N}^{\mu_1}$ and the quiver $Q_{sl_N}^{\mu_1}$

The above derivation of $\mathcal{L}_{\boldsymbol{R}}^{\mu_1}$ (3.63) can be stated in a formal way for any representation $\boldsymbol{R}$. In this case, one may think of $\mathcal{L}_{\boldsymbol{R}}^{\mu_1}$ like $\mathcal{L}_{sl_N}^{\mu_1}$ with $sl_N$ refering to any representation $\boldsymbol{R}$ of the gauge symmetry. But here let us sill keep $\boldsymbol{R} = \boldsymbol{N}$ and use the property $\varrho_1 + \varrho_{\bar 1} = I_{id}$ to cast $\mathcal{L}_{sl_N}^{\mu_1}$ in different but equivalent ways: First as $I_{id}\mathcal{L}^{\mu_1}$ and $\mathcal{L}^{\mu_1}I_{id}$ reading explicitly like

$$\begin{aligned} \mathcal{L}_{sl_N}^{\mu_1} &= \varrho_1\mathcal{L}^{\mu_1} + \varrho_{\bar 1}\mathcal{L}^{\mu_1} \\ &= \mathcal{L}^{\mu_1}\varrho_1 + \mathcal{L}^{\mu_1}\varrho_{\bar 1} \end{aligned} \tag{3.67}$$

Second, we can also use the form $I_{id}\mathcal{L}^{\mu_1}I_{id}$ and the resolution of the identity $I_{id}$ in terms of the projectors to express the Lax operator as $(\varrho_1 + \varrho_{\bar{1}})\,\mathcal{L}^{\mu_1}\,(\varrho_1 + \varrho_{\bar{1}})$; thus leading to

$$\mathcal{L}^{\mu_1}_{sl_N} = \varrho_1\mathcal{L}^{\mu_1}\varrho_1 + \varrho_1\mathcal{L}^{\mu_1}\varrho_{\bar{1}} + \varrho_{\bar{1}}\mathcal{L}^{\mu_1}\varrho_1 + \varrho_{\bar{1}}\mathcal{L}^{\mu_1}\varrho_{\bar{1}} \tag{3.68}$$

By help of $\varrho_1^2 = \varrho_1$ and $\varrho_{\bar{1}}^2 = \varrho_{\bar{1}}$ as well as $\varrho_1\varrho_{\bar{1}} = 0$, we can present $\mathcal{L}^{\mu_1}_{sl_N}$ in the operator basis $(\varrho_1, \varrho_{\bar{1}})$ like a 2×2 matrix blocks as follows

$$\mathcal{L}^{\mu_1}_{sl_N} = \begin{pmatrix} \varrho_1\mathcal{L}^{\mu_1}\varrho_1 & \varrho_1\mathcal{L}^{\mu_1}\varrho_{\bar{1}} \\ \varrho_{\bar{1}}\mathcal{L}^{\mu_1}\varrho_1 & \bar{\varrho}_1\mathcal{L}^{\mu_1}\varrho_{\bar{1}} \end{pmatrix} \tag{3.69}$$

This structure given here for $sl_N$ was behind the topological gauge quiver used in [62] to study the exceptional 't Hooft lines $E_6$ and $E_7$. In this investigation, we further develop this approach to 4D CS gauge theory with $sl_N$ gauge symmetry and beyond. For the case of minuscule coweight $\mu_1$ of $sl_N$, the topological gauge quiver $Q^{\mu_1}_{sl_N}$ is depicted in the Figure **8**.

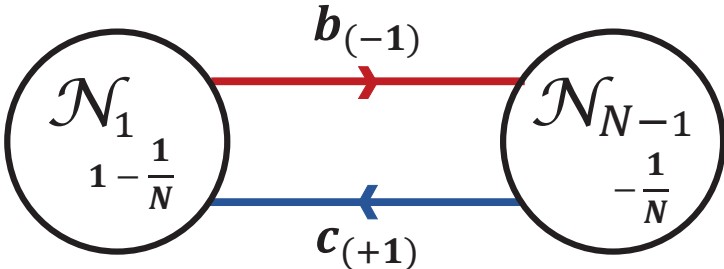

Figure 8: The topological quiver $Q^{\mu_1}_{sl_N}$ representing $\mathcal{L}^{\mu_1}_{sl_N}$ of $sl_N$. It has 2 nodes and 2 links. The nodes describe self-dual topological matter and the links describe topological bi-matter.

Its two nodes $\mathcal{N}_1$ and $\mathcal{N}_{\bar{1}}$ are given by $\mathcal{N}_1 = \varrho_1\mathcal{L}^{\mu_1}\varrho_1$ and $\mathcal{N}_{\bar{1}} = \varrho_{\bar{1}}\mathcal{L}^{\mu_1}\varrho_{\bar{1}}$ respectively interpreted as topological adjoint matter of $SL_1$ and $SL_{N-1}$. Below, we refer to this adjoint matter as topological self dual matter; it is uncharged under the minuscule coweight operator. The links $L_{ij}$ between the nodes are given by $L_{1\bar{1}} = \varrho_1\mathcal{L}^{\mu_1}\varrho_{\bar{1}}$ and $L_{\bar{1}1} = \varrho_{\bar{1}}\mathcal{L}^{\mu_1}\varrho_1$. They carry charges under $SL_1 \times SL_{N-1}$; and are interpreted in terms of topological bi-fundamental matter.

### 3.3   Topololgical quivers: case $2 \leq k \leq N - 2$

Here, we generalise the construction of subsection 3.2 regarding the minuscule coweight $\mu_1$ to the generic minuscule coweight $\mu_k$ with $2 \leq k \leq N - 2$.

### 3.3.1 Generic projectors $\Pi_k$ and $\Pi_{\bar{k}}$

In the generic case, the expression (2.49) involves the projectors $\Pi_k$ and $\Pi_{\bar{k}}$ on the representations of the Levi subalgebra $sl_k \oplus sl_{N-k} \oplus sl_1$. Using the properties

$$X\Pi_k = 0 \qquad , \qquad \Pi_k Y = 0 \tag{3.70}$$

and the identities

$$X\Pi_{\bar{k}} = X \qquad , \qquad \Pi_{\bar{k}} Y = Y \tag{3.71}$$

leading to $X\Pi_{\bar{k}} Y = XY$, the expression of the L- operator $\mathcal{L}_{\boldsymbol{R}}^{\mu_k}$ takes the simple form

$$\mathcal{L}_{\boldsymbol{R}}^{\mu_k} = z^{-\frac{k}{N}} (\Pi_{\bar{k}} + X + Y + XY) + z^{\frac{N-k}{N}} \Pi_k \tag{3.72}$$

In the projector basis $(\Pi_k, \Pi_{\bar{k}})$, this $\mathcal{L}_{\boldsymbol{R}}^{\mu_k}$ can be presented in block matrices like

$$\mathcal{L}_{\boldsymbol{R}}^{\mu_k} = \begin{pmatrix} (z^{\frac{N-k}{N}} + z^{-\frac{k}{N}} XY)\Pi_k & z^{-\frac{k}{N}} X\Pi_{\bar{k}} \\ z^{-\frac{k}{N}} \Pi_{\bar{k}} Y & z^{-\frac{k}{N}} \Pi_{\bar{k}} \end{pmatrix} \tag{3.73}$$

By exhibiting the dependence into the Darboux coordinates while substituting $X = b^{i\alpha} X_{ia}$ and $Y = c_{j\beta} Y^{j\beta}$ as well as $XY = b^{i\alpha} c_{i\alpha}$, we obtain

$$\mathcal{L}_{\boldsymbol{R}}^{\mu_k} = \begin{pmatrix} z^{-\frac{k}{N}} (z + b^{i\alpha} c_{i\alpha})\Pi_k & z^{-\frac{k}{N}} b^{i\alpha} X_{ia}\Pi_{\bar{k}} \\ z^{-\frac{k}{N}} c_{j\beta}\Pi_{\bar{k}} Y^{j\beta} & z^{-\frac{k}{N}} \Pi_{\bar{k}} \end{pmatrix} \tag{3.74}$$

### 3.3.2 Constructing the topological quivers $\mathbf{Q}_{\boldsymbol{R}}^{\mu_k}$

By using the property $\Pi_k + \Pi_{\bar{k}} = I$, we can cast $\mathcal{L}_{\boldsymbol{R}}^{\mu_k}$ as follows

$$\mathcal{L}_{\boldsymbol{R}}^{\mu_k} = (\Pi_k + \Pi_{\bar{k}}) \mathcal{L}^{\mu_k} (\Pi_k + \Pi_{\bar{k}}) \tag{3.75}$$

Using $\Pi_k \Pi_{\bar{k}} = 0$, we can put this $\mathcal{L}^{(\mu_k)}$ into the following matrix form

$$\mathcal{L}_{\boldsymbol{R}}^{\mu_k} = \begin{pmatrix} \Pi_k \mathcal{L}^{\mu_k} \Pi_k & \Pi_k \mathcal{L}^{\mu_k} \Pi_{\bar{k}} \\ \Pi_{\bar{k}} \mathcal{L}^{\mu_k} \Pi_k & \Pi_{\bar{k}} \mathcal{L}^{\mu_k} \Pi_{\bar{k}} \end{pmatrix} \tag{3.76}$$

The topological gauge quiver $\mathbf{Q}_{\boldsymbol{R}}^{\mu_k}$ associated with this L-operator has two nodes $\mathcal{N}_k, \mathcal{N}_{\bar{k}}$ and two links $L_{k\bar{k}}, L_{\bar{k}k}$. It is depicted by the Figure **9**.

The two nodes are given by

$$\mathcal{N}_k = \Pi_{\bar{k}} \mathcal{L}^{\mu_k} \Pi_k \qquad , \qquad \mathcal{N}_{\bar{k}} = \Pi_{\bar{k}} \mathcal{L}^{\mu_k} \Pi_{\bar{k}} \tag{3.77}$$

they describe topological adjoint matter of $SL_k$ and $SL_{N-k}$; and interpreted as topological self dual matter. The two links relating the two nodes. They are given by

$$\Pi_k \mathcal{L}^{\mu_k} \Pi_{\bar{k}} \qquad , \qquad \Pi_{\bar{k}} \mathcal{L}^{\mu_k} \Pi_k \tag{3.78}$$

they describe bi-fundamental matter of $SL_k \times SL_{N-k}$ and are interpreted as topological bi-matter. These bi-fundamental matters are precisely given by the Darboux variables $b^{i\alpha}$ and $c_{ia}$.

To end this section, notice the three following features:

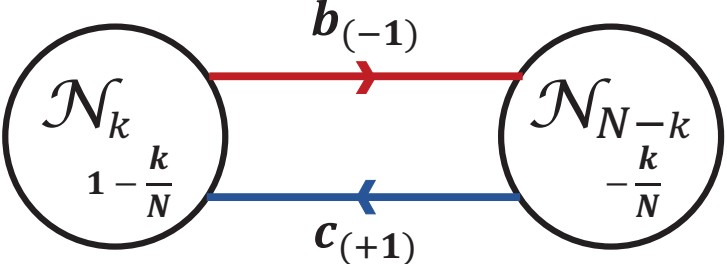

Figure 9: The topological quiver representing $\mathcal{L}^{\mu_k}$ of $sl_N$. It has 2 nodes and 2 links. The nodes describe self-dual topological matter and the links describe bi-matter.

**(1)** the topological quiver $Q_{\boldsymbol{R}}^{\mu_1}$ of the operator $\mathcal{L}_{\boldsymbol{R}}^{\mu_1}$ appears just as the leading quiver of the k-family $Q_{\boldsymbol{R}}^{\mu_k}$ associated with the family $\mathcal{L}_{\boldsymbol{R}}^{\mu_k}$. So, the topological quiver $Q_{\boldsymbol{R}}^{\mu_{N-1}}$ of the $\mathcal{L}_{\boldsymbol{R}}^{\mu_{N-1}}$ turns out be just the last member of the k-family. We omit its description.

**(2)** the quiver $Q_{\boldsymbol{R}}^{\mu_k}$ given in this section concern Wilson lines with quantum states in the fundamental $\boldsymbol{R} = \boldsymbol{N}$. For Wilson lines in other representations of $sl_N$ like the completely antisymmetric $\boldsymbol{N}^{\wedge k}$ and the completely symmetric $\boldsymbol{N}^{\vee n}$, we can construct the associated the L-operators and the corresponding quivers $Q_{\boldsymbol{R}}^{\mu_k}$. Examples of the topological quivers $Q_{\boldsymbol{N}^{\wedge k}}^{\mu_1}$ and $Q_{\boldsymbol{N}^{\vee n}}^{\mu_1}$ are given in Figure **29**. Their Levi charges reported on the nodes can be read from the decomposition (2.26). As an illustration, the quiver $Q_{\boldsymbol{N}^{\vee 3}}^{\mu_1}$ corresponding to the representation the symmetric $\boldsymbol{N} \vee \boldsymbol{N} \vee \boldsymbol{N}$ is depicted by the Figure **10**.

**(3)** An interesting topological quiver diagram $Q_{\boldsymbol{adj}(sl_N)}^{\mu_k}$ given by the Figure **11**. It is the one associated with the adjoint representation; that is $\boldsymbol{R} = \boldsymbol{adj}(sl_N)$. From the decomposition given by eq(2.26), we see that $\boldsymbol{adj}(sl_N)$ splits as $\boldsymbol{n}_- \oplus \boldsymbol{l}_{\mu_k} \oplus \boldsymbol{n}_+$ with $\boldsymbol{l}_{\mu_k} = \boldsymbol{adj}(sl_k)_0 + \boldsymbol{adj}(sl_{N-k})_0 + sl_1$ and $\boldsymbol{n}_\pm = \boldsymbol{k}(\boldsymbol{N} - \boldsymbol{k})_\pm$. The second concerns $sl_1$ with the representation $\boldsymbol{1}_0 = \boldsymbol{1}_{1-1/N} \times \bar{\boldsymbol{1}}_{-1+1/N}$.

# 4  Vector 't Hooft lines of $D_N$- type

In this section, we study the class of vector-like L-operators $\mathcal{L}_{so_{2N}}^{vect}$ in the 4D Chern-Simons theory with $SO_{2N}$ gauge symmetry. This is a sub-family of the family of D- type Lax operators which contains moreover the Lax operators $\mathcal{L}_{so_{2N}}^{spin}$ of the spinorial class to be studied in the next section. Because $SO_4 = SU_2 \times SU_2$ and $SO_6 \sim D_3$ is isomorphic to $SL_4$, we assume that $N \geq 4$ so that the first element of the $D_N$ series is given by $SO_8$.

Notice that the general aspects of the present construction are similar to those introduced in

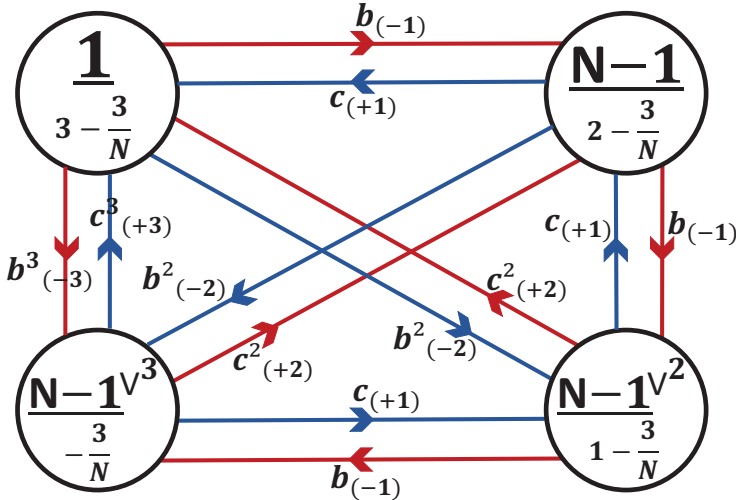

Figure 10: The topological quiver $Q_R^{\mu_1}$ for the representation $\boldsymbol{R} = \boldsymbol{N} \vee \boldsymbol{N} \vee \boldsymbol{N}$. This quiver has four nodes and 12 links.

the previous sections. The 't Hooft line $\text{tH}_{\gamma_0}^{\mu}$ is taken as the horizontal x-axis of $\mathbb{R}^2$ and the $W_{\xi_z}^{\boldsymbol{R}}$ is chosen as the vertical y-axis; the $z$ is a generic position in the holomorphic line $\mathbb{CP}^1$, and $\boldsymbol{R}$ is a given representation of $so_{2N}$. Moreover, most of the features associated to the derivation of Lax operators from 4D CS with $SO_{2N}$ gauge symmetry have been considered in [52, 60]. Therefore, we focus here on analysing the internal algebraic structure of this theory allowing to illustrate the key elements of the quiver gauge $Q_{so_{2N}}^{vect}$ associated to $\mathcal{L}_{so_{2N}}^{vect}$. This quiver constitutes a necessary part in the unified theory chain in the sense that it links the A-type symmetries to the exceptional ones, and allows to indirectly include the B-type symmetries thanks to its similarity with the minuscule coweight of the $so_{2N+1}$ Lie algebra.

## 4.1  Vector lines $\text{tH}_{\gamma_0}^{\mu_1}$ and their L-operators

We begin by recalling that minuscule 't Hooft lines within the $D_N$ family of 4D CS theory are magnetically charged with magnetic charge given by the minuscule coweights $\mu$ of $D_N$. Because there are three minuscule coweights in the $D_N$ Lie algebras given by $\mu_1, \mu_{N-1}, \mu_N$ (see the Figures **13** and **17**), we distinguish three types of 't Hooft lines $\text{tH}_{\gamma_0}^{\mu}$ in the 4D Chern-Simons theory with orthogonal gauge symmetry $SO_{2N}$ that we can refer to as

$$\text{tH}_{\gamma_0}^{\mu_1} = \text{tH}_{\gamma_0}^{vect} \quad , \quad \text{tH}_{\gamma_0}^{\mu_N} = \text{tH}_{\gamma_0}^{spin} \quad , \quad \text{tH}_{\gamma_0}^{\mu_{N-1}} = \text{tH}_{\gamma_0}^{cospin} \tag{4.79}$$

The coweights $\mu_1, \mu_{N-1}, \mu_N$ are respectively dual to the vector representation $2N$, the spinor representation $2_L^{N-1}$ and the cospinor representation $2_R^{N-1}$. Here, we first focus on the vector-like $\text{tH}_{\gamma_0}^{\mu_1}$ and move in the next section to the study of $\text{tH}_{\gamma_0}^{\mu_{N-1}}$ and $\text{tH}_{\gamma_0}^{\mu_N}$. To fix the ideas, we illustrate in the Figure **12** the Levi splitting characterising $\text{tH}_{\gamma_0}^{vect}$. This intrinsic structure will be derived and commented later on.

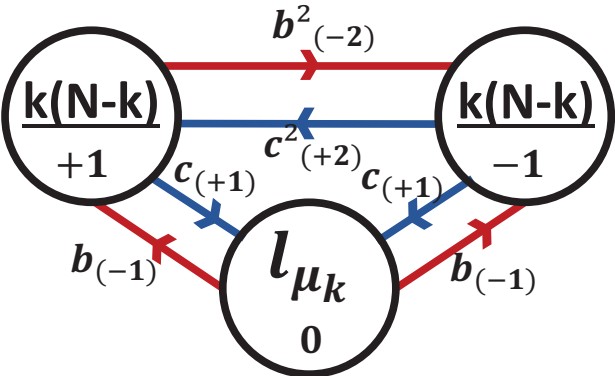

Figure 11: The topological quiver $Q^{\mu_k}_{adj(sl_N)}$ for the adjoint representation of $sl_N$. It has three nodes $\mathcal{N}_0 = l_{\mu_k}$ and $n_\pm = k\,(N-k)_\pm$.

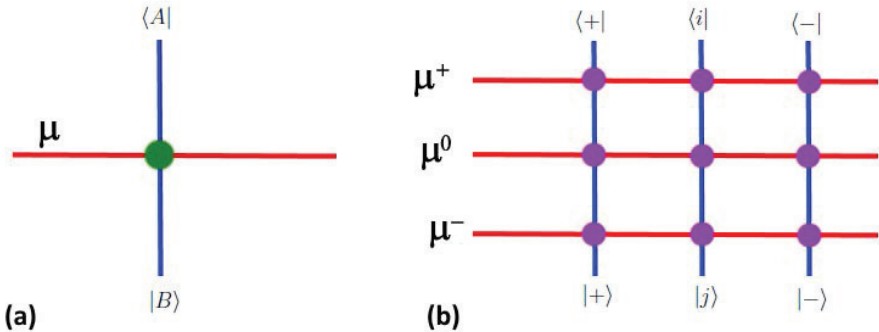

Figure 12: (a) A horizontal vector-like 't Hooft line with magnetic charge $\mu_1$ crossing an electrically charged vertical Wilson line. The green dot refers to the coupling between the two lines; it is given by the Lax operator $\mathcal{L}^{\mu_1}_R$. (b) Intrinsic structure of the Lax operator interpreted in terms of a topological gauge quiver with three nodes and 6 links.

### 4.1.1 Vectorial $\mathrm{tH}^{vect}_{\gamma_0}$ line: magnetic charge

The fundamental coweight $\mu_1$ is the dual to the simple root $\alpha_1$ of the $so_{2N}$ Lie algebra. By taking the N simple roots of $SO_{2N}$ as $\alpha_i = e_i - e_{i+1}$ for $i \in [1, N-1]$ and $\alpha_N = e_{N-1} + e_N$; it follows that the value of the minuscule coweight constrained as $\mu_1 \alpha_i = \delta_{i1}$ can be solved like $\mu_1 = e_1$. In terms of the simple roots, we have

$$\mu_1 = \alpha_1 + ... + \alpha_{N-2} + \frac{1}{2}\left(\alpha_{N-1} + \alpha_N\right) \tag{4.80}$$

Notice that by setting N=3 in this relation, the resulting $\mu_1$ takes the value $\alpha_1 + \frac{1}{2}\left(\alpha_2 + \alpha_3\right)$ which can be compared with the fundamental weight $\tilde{\mu}_2 = \tilde{\alpha}_2 + \frac{1}{2}(\tilde{\alpha}_1 + \tilde{\alpha}_3)$ of the $sl_4$ Lie algebra which is isomorphic to $so_6$. Here, the $\tilde{\alpha}_i$'s stand for the simple roots of $sl_4$.

From the Dynkin diagram of the $D_N$ Lie algebras given in Figure **13**, we can see that the

Levi decomposition $\boldsymbol{l}_{\mu_1} \oplus n_+ \oplus n_-$ of $so_{2N}$ with respect to the vectorial coweight $\mu_1$ is given by $\boldsymbol{l}_{\mu_1} = so_2 \oplus so_{2N-2}$ and $n_{\pm} = (2N-2)_{\pm}$ with the charge symmetry $so_2 \sim sl_1$. As such, the dimensions of the **adj** $[so_{2N}]$ and the vector $\boldsymbol{2N}$ representations split as follows

$$
\begin{aligned}
N(2N-1) &= 1_0 + (N-1)(2N-3)_0 + (2N-2)_+ + (2N-2)_- \\
\boldsymbol{2N} &= \boldsymbol{2}_0 + (\boldsymbol{2N-2})_0
\end{aligned}
\qquad (4.81)
$$

where we have also exhibited the charge of $so_2$. To construct the L-operator of the tH$^{vect}_{\gamma_0}$ line represented graphically by the Figure **12-(a)**, we need the adjoint action of the coweight $\mu_1$ and the explicit expressions of the generators of the nilpotent subalgebras $n_{\pm}$.

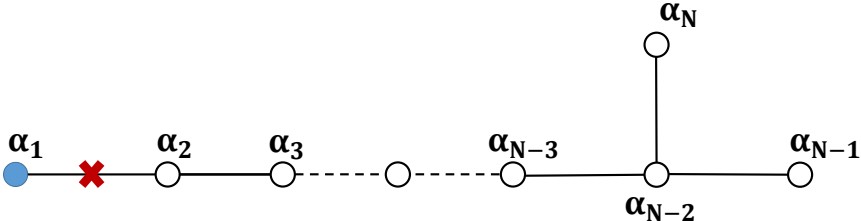

Figure 13: Dynkin diagram of $D_N$ Lie algebras where the $N$ simple roots $\alpha_i$ are exhibited. The Levi decomposition of $so_{2N} \to so_2 \oplus so_{2N-2}$ using the vector coweight is given by cutting the simple root $\alpha_1$.

The $2N-2$ generators of $n_+$ are denoted by $X_i$ and their homologues generating $n_-$ are denoted like $Y^i$, their realisation should solve the Levi decomposition constraint $[\mu_1, n_{\pm}] = \pm n_{\pm}$ and $[n_q, n_q] = 0$ with $q = \pm$.
To get this solution, we consider (**i**) an electric vertical Wilson line as in the Figure **12-(a)**

$$
W^{\boldsymbol{R=2N}}_{\xi_z} \qquad , \qquad \xi_z = \{(x,y)\,|\,x=0; -\infty < y < \infty\} \qquad (4.82)
$$

with incoming vector-like states $\langle A|$ (A=1,..., 2N) and outgoing $|B\rangle$ ones propagating along the line $\xi_z$. (**ii**) a horizontal 't Hooft line defect tH$^{vect}_{\gamma_0}$ with the magnetic charge $\mu_1$;

$$
tH^{vect}_{\gamma_0} \qquad , \qquad \gamma_0 = \mathbb{R}^2_{y\leq 0} \cap \mathbb{R}^2_{y\geq 0} \qquad (4.83)
$$

In this case, we can split the vector representation $|B\rangle$ of SO$_{2N}$ as a direct sum $|\beta\rangle \oplus |j\rangle$ where $|\beta\rangle$ is a vector of $so_2$ and $|j\rangle$ a vector of $so_{2N-2}$. Moreover, we use the isomorphism $so_2 \sim sl_1$ to split $|\beta\rangle$ as $|+\rangle$ and $|-\rangle$. Eventually, the 2N states split as

$$
|B\rangle = \begin{pmatrix} |0\rangle \\ |j\rangle \\ |\bar{0}\rangle \end{pmatrix} \equiv \begin{pmatrix} |+\rangle \\ |j\rangle \\ |-\rangle \end{pmatrix}, \qquad 1 \leq j \leq M \qquad (4.84)
$$

where we have set $M = 2N-2$ and considered the splitting of the $\boldsymbol{2N}$ vector as $\boldsymbol{1}_+ \oplus (\boldsymbol{2N-2})_0 \oplus \boldsymbol{1}_-$ such that the Levi subalgebra is $sl_1 \oplus so_{2N-2}$. In this vector states basis (4.84), the operators $X_i$ and $Y^i$ generating the nilpotent subalgebras are given by

$$
\begin{aligned}
X_i &= |+\rangle \langle i| - |i\rangle \langle -| \\
Y^i &= |i\rangle \langle +| - |-\rangle \langle i|
\end{aligned}
\qquad (4.85)
$$

The action of these operators $X_i$ and $Y^i$ on the vector representation of $so_{2N}$ can be visualized in the the Figure **14** describing the splitting of the **2N** vector. As for the adjoint action of the minuscule coweight, it is given by a particular linear combination of projectors $\varrho_R$ on the irreducible representations $\mathbf{1}_\pm$ and $(\mathbf{2N-2})_0$ of the $so_2 \oplus so_{2N-2}$ Levi subalgebra as follows

$$\mu_1 = \varrho_+ - \varrho_- \tag{4.86}$$

with $\varrho_+ = |+\rangle \langle+|$ and $\varrho_- = |-\rangle \langle-|$.

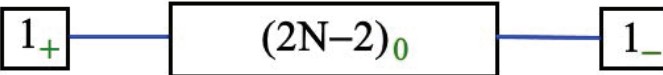

Figure 14: A graphic representation of the splitting of the vector 2N representation under the vectorial Levi decomposition. The projectors on these three blocks are $\varrho_+ = |+\rangle \langle+|$, $\sum \varrho_i = \sum |i\rangle \langle i|$ and $\varrho_- = |-\rangle \langle-|$.

Because of the vanishing $so_2$ charge of $(\mathbf{2N-2})_0$, the minuscule coweight has no dependence on the projector

$$\Pi_0 = \sum_i \varrho_i \tag{4.87}$$

with $\varrho_i = |i\rangle \langle i|$. Notice that $X_i$ and $Y^j$ satisfy some characteristic relations like for example $X_i Y^j = \delta_j^j \varrho_+ + |i\rangle \langle j|$ indicating that

$$Tr\left(X_i Y^i\right) = 2\delta_j^j \tag{4.88}$$

From the realisation of eqs(4.85-4.86) we can deduce that $[\mu_1, X_i] = +X_i$, $[\mu_1, Y^i] = -Y^i$ and $[X_i, Y^i] = \mu_1$. Other useful and simplifying relations are listed below

$$\begin{array}{rclcrcl}
X_i X_j & = & -\delta_{ij} |+\rangle \langle-| & , & X_i X_j X_l & = & 0 \\
Y^i Y^j & = & -\delta^{ij} |-\rangle \langle+| & , & Y^i Y^j Y^l & = & 0
\end{array} \tag{4.89}$$

and

$$\begin{array}{rclcrcl}
\varrho_- X_i & = & 0 & , & X_i \varrho_+ & = & 0 \\
\varrho_+ Y^i & = & 0 & , & Y^i \varrho_- & = & 0
\end{array} \tag{4.90}$$

as well as

$$\begin{array}{rclcrcl}
X_i \varrho_- & = & -|i\rangle \langle-| & , & \varrho_+ X_i & = & |+\rangle \langle i| \\
\varrho_- Y^i & = & -|-\rangle \langle i| & , & Y^i \varrho_+ & = & |i\rangle \langle+|
\end{array} \tag{4.91}$$

By considering the linear combinations

$$X = b^i X_i \in \mathbf{n}_+ \quad , \quad Y = c_i Y^i \in \mathbf{n}_- \tag{4.92}$$

where $b^i$ and $c_i$ are the phase space coordinates, we can calculate their powers $X^n$ and $Y^n$; and then $e^X$ and $e^Y$. We find that $X^2 = -\mathbf{b}^2 E$, $Y^2 = -\mathbf{c}^2 F$ and $X^3 = Y^3 = 0$ where we have set $\mathbf{b}^2 = b^i \delta_{ij} b^j$ and $\mathbf{c}^2 = c_i \delta^{ij} c_j$ as well as $E = |+\rangle \langle -|$ and $F = |-\rangle \langle +|$ satisfying $[E, F] = \mu_1$ and $Tr(EF) = 1$. We also have

$$
\begin{aligned}
b^i &= \tfrac{1}{2} Tr(XY^i) &, \qquad \mathbf{b}^2 &= -Tr(X^2 F) \\
c_i &= \tfrac{1}{2} Tr(X_i Y) &, \qquad \mathbf{c}^2 &= -Tr(Y^2 E)
\end{aligned}
\tag{4.93}
$$

Moreover, we have

$$
\begin{aligned}
\varrho_- X &= 0 &, \qquad \varrho_+ X &= b^i |+\rangle \langle i| &, \qquad X\varrho_- &= -b^i |i\rangle \langle -| \\
\varrho_+ Y &= 0 &, \qquad \varrho_- Y &= -c_i |-\rangle \langle i| &, \qquad \varrho_- Y &= -c_i |-\rangle \langle i|
\end{aligned}
\tag{4.94}
$$

### 4.1.2  Vector- like $\text{tH}_{\gamma_0}^{vect}$ line: building the L-operator

Using the properties $X^3 = Y^3 = 0$ indicating that $e^X = I + X + \tfrac{1}{2}X^2$ and equivalently for $e^Y$; then putting back into the expression of the L-operator namely $\mathcal{L} = e^X z^{\mu_1} e^Y$, we obtain

$$
\begin{aligned}
\mathcal{L} = \; & z^{\mu_1} + X z^{\mu_1} + z^{\mu_1} Y + \\
& \tfrac{1}{2} z^{\mu_1} Y^2 + \tfrac{1}{2} X^2 z^{\mu_1} + X z^{\mu_1} Y \\
& + \tfrac{1}{2} X z^{\mu_1} Y^2 + \tfrac{1}{2} X^2 z^{\mu_1} Y + \tfrac{1}{4} X^2 z^{\mu_1} Y^2
\end{aligned}
\tag{4.95}
$$

with higher monomial given by $X^2 z^{\mu_1} Y^2$. Replacing $z^{\mu_1} = z\varrho_+ + z^{-1}\varrho_-$ and using eq(4.90) indicating that

$$
X z^{\mu_1} = z^{-1} X \varrho_-, \qquad z^{\mu_1} Y = z^{-1} \varrho_- Y
\tag{4.96}
$$

the above L-operator reads as follows

$$
\begin{aligned}
\mathcal{L} = \; & z\varrho_+ + z^{-1}\varrho_- + z^{-1} X \varrho_- + z^{-1} \varrho_- Y + \\
& \tfrac{1}{2} z^{-1} \varrho_- Y^2 + \tfrac{1}{2} z^{-1} X^2 \varrho_- + z^{-1} X \varrho_- Y \\
& + \tfrac{1}{2} z^{-1} X \varrho_- Y^2 + \tfrac{1}{2} z^{-1} X^2 \varrho_- Y + \tfrac{1}{4} z^{-1} X^2 \varrho_- Y^2
\end{aligned}
\tag{4.97}
$$

This operator has a remarkable dependence on the projector $\varrho_-$. Using the non vanishing $\varrho_+ X_i X_j \varrho_- = -\delta_{ij} E$ and $\varrho_- Y^i Y^j \varrho_+ = -\delta^{ij} F$ as well as $\varrho_+ X_i X_j \varrho_- Y^k Y^l \varrho_+ = \delta_{ij}\delta^{kl} \varrho_+$, we have

$$
\begin{aligned}
\varrho_+ \mathcal{L} \varrho_+ &= z\varrho_+ + \tfrac{1}{4} z^{-1} \varrho_+ X^2 \varrho_- Y^2 \varrho_+ \\
\varrho_+ \mathcal{L} \Pi_0 &= \tfrac{1}{2} z^{-1} \varrho_+ X^2 \varrho_- Y \Pi_0 \\
\varrho_+ \mathcal{L} \varrho_- &= \tfrac{1}{2} z^{-1} \varrho_+ X^2 \varrho_-
\end{aligned}
\tag{4.98}
$$

and

$$
\begin{aligned}
\Pi_0 \mathcal{L} \varrho_+ &= \tfrac{1}{2} z^{-1} \Pi_0 X \varrho_- Y^2 \varrho_+ \\
\Pi_0 \mathcal{L} \Pi_0 &= z^{-1} \Pi_0 X \varrho_- Y \Pi_0 = z^{-1} b^i \text{E}_i^j c_j \\
\Pi_0 \mathcal{L} \varrho_- &= z^{-1} \Pi_0 X \varrho_-
\end{aligned}
\tag{4.99}
$$

with $\text{E}_i^j = |i\rangle \langle j|$, and

$$
\begin{aligned}
\varrho_- \mathcal{L} \varrho_+ &= \tfrac{1}{2} z^{-1} \varrho_- Y^2 \varrho_+ \\
\varrho_- \mathcal{L} \Pi_0 &= z^{-1} \varrho_- Y \Pi_0 \\
\varrho_- \mathcal{L} \varrho_- &= z^{-1} \varrho_-
\end{aligned}
\tag{4.100}
$$

Substituting $X\varrho_- = -b^i x_i$ and $\varrho_- Y = -c_i y^i$ as well as $X^2\varrho_- = -\mathbf{b}^2 E$ and $\varrho_- Y^2 = -\mathbf{c}^2 F$ by help of eqs(4.91,4.93,4.94), we obtain

$$
\begin{aligned}
\mathcal{L}_{D_N}^{vect} =\ & \left(z + \tfrac{1}{4}z^{-1}\mathbf{b}^2\mathbf{c}^2\right)\varrho_+ + z^{-1}\varrho_- - z^{-1}b^i x_i - z^{-1}c_i y^i + \\
& -\tfrac{1}{2}z^{-1}\mathbf{c}^2 F - \tfrac{1}{2}z^{-1}\mathbf{b}^2 E + z^{-1}\left(b^i c_j\right)x_i y^j \\
& +\tfrac{1}{2}z^{-1}\left(b^i\mathbf{c}^2\right)x_i F + \tfrac{1}{2}z^{-1}\mathbf{b}^2 c_i E y^i
\end{aligned}
\tag{4.101}
$$

## 4.2 Topological quiver $\mathrm{Q}_{so_{2N}}^{vect}$ : case of the vector 't Hooft line

From the realisation eqs(4.85-4.86) and the diagram of the Figure **12**, we learn that the Lax operator $\mathcal{L}_{so_{2N}}^{vect}$ has an intrinsic structure that can be represented by a topological gauge quiver $\mathrm{Q}_{so_{2N}}^{vect}$. To draw this topological quiver diagram, we use the projectors $\varrho_+, \varrho_-$ and $\Pi_0 = \sum \varrho_i$, singling out the representations of the Levi subgroup $SO_2 \times SO_{2N-2}$ of the orthogonal symmetry $SO_{2N}$, to cast eq(4.97) as follows

$$
\mathcal{L}_{so_{2N}}^{vect} = \begin{pmatrix}
\varrho_+\mathcal{L}\varrho_+ & \varrho_+\mathcal{L}\Pi_0 & \varrho_+\mathcal{L}\varrho_- \\
\Pi_0\mathcal{L}\varrho_+ & \Pi_0\mathcal{L}\Pi_0 & \Pi_0\mathcal{L}\varrho_- \\
\varrho_-\mathcal{L}\varrho_+ & \varrho_-\mathcal{L}\Pi_0 & \varrho_-\mathcal{L}\varrho_-
\end{pmatrix}
\tag{4.102}
$$

In this decomposition, we have used the relation $\varrho_+ + \Pi_0 + \varrho_- = I_{id}$ and $\varrho_\pm\Pi_0 = \varrho_+\varrho_- = 0$. Finally, we recover the matrix representation in agreement with [67]

$$
\mathcal{L}_{so_{2N}}^{vect} = z^{-1}\begin{pmatrix}
z^2 + \tfrac{1}{4}\mathbf{b}^2\mathbf{c}^2 & \tfrac{1}{2}\mathbf{b}^2 c_i & -\tfrac{1}{2}\mathbf{b}^2 \\
\tfrac{1}{2}b^j\mathbf{c}^2 & b^j c_i & -b^j \\
-\tfrac{1}{2}\mathbf{c}^2 & -c_i & 1
\end{pmatrix}
\tag{4.103}
$$

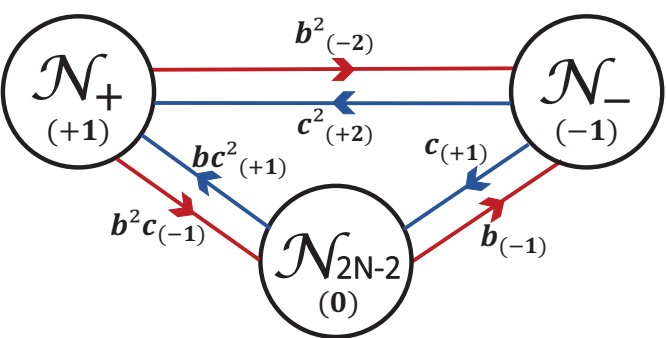

Figure 15: The topological quiver representing $\mathcal{L}_{so_{2N}}^{vect}$. It has three nodes and 6 links. The nodes describe self-dual topological matter and the links describe topological bi-matter.

The topological gauge quiver $\mathrm{Q}_{so_{2N}}^{vect}$ representing the above vector like $\mathcal{L}_{so_{2N}}^{vect}$ is given by the Figure **15**. The $\mathrm{Q}_{so_{2N}}^{vect}$ has three nodes $\mathcal{N}_+$, $\mathcal{N}_{2N-2}$ and $\mathcal{N}_-$ given by

$$
\mathcal{N}_+ \equiv \langle\varrho_+\mathcal{L}\varrho_+\rangle \quad , \quad \mathcal{N}_{2N-2} \equiv \langle\Pi_0\mathcal{L}\Pi_0\rangle \quad , \quad \mathcal{N}_- \equiv \langle\varrho_-\mathcal{L}\varrho_-\rangle
\tag{4.104}
$$

It has 3+3 links $L_{ij}$ with $i, j = 0, \pm$ interpreted as topological *bi-fundamental matter* $SO_2 \times SO_{2N-2}$ reading as

$$L_{+0} = \langle \varrho_+ \mathcal{L} \Pi_0 \rangle \quad , \quad L_{0+} = \langle \Pi_0 \mathcal{L} \varrho_+ \rangle \quad , \quad L_{-+} = \langle \varrho_- \mathcal{L} \varrho_+ \rangle$$
$$L_{+-} = \langle \varrho_+ \mathcal{L} \varrho_- \rangle \quad , \quad L_{0-} = \langle \Pi_0 \mathcal{L} \varrho_- \rangle \quad , \quad L_{-0} = \langle \varrho_- \mathcal{L} \Pi_0 \rangle$$

$$(4.105)$$

Notice that The Darboux coordinates can be expressed in terms of the operator $\mathcal{L}_{so_{2N}}^{vect}$ and the generators $X_i$, $Y^i$ and the minuscule coweight $\mu_1$ as follows :

$$b^i = zTr\left(\mu_1 Y^i \mathcal{L}_{so_{2N}}^{vect}\right), \qquad c_i = zTr\left(\mathcal{L}_{so_{2N}}^{vect} X_i \mu_1\right) \qquad (4.106)$$

While $b^i$ and $c_i$ sit respectively in the vector representation of $SO_{2N-2}$ and its transpose, they carry opposite unit charges under the minuscule coweight $\mu_1$.

As for the Darboux, we also have their composites that appear in the expression of the L-operator, they are scalars of $SO_{2N-2}$ and carry non trivial $SO_2$ charges. They are given by

$$\mathbf{b}^2 = -2zTr\left(F\mathcal{L}_{so_{2N}}^{vect}\right), \qquad \mathbf{c}^2 = -2zTr\left(E\mathcal{L}_{so_{2N}}^{vect}\right) \qquad (4.107)$$

where E and F are related to the minuscule coweight operator as $[E, F] = \mu_1$. Interesting composites of the Darboux coordinates that transform non trivially under $SO_2$ are given by

$$\mathbf{b}^2 c_i = 2zTr\left(\mu_1 F \mathcal{L}_{so_{2N}}^{vect} X_i\right), \qquad b^i \mathbf{c}^2 = 2zTr\left(\mu_1 Y^i \mathcal{L}_{so_{2N}}^{vect} E\right) \qquad (4.108)$$

# 5 Spinorial 't Hooft lines of $D_N$- type

This section is a continuation to the previous one, it concerns the operators $\mathcal{L}_{so_{2N}}^{spin}$. Here, we introduce the two spinorial like 't Hooft lines of $D_N$ type denoted as $tH_{\gamma_0}^{\mu_{N-1}}$ and $tH_{\gamma_0}^{\mu_N}$ and construct the associated Lax operators $\mathcal{L}_{so_{2N}}^{spin}$. We cast their special properties in the associated topological quivers $Q_{so_{2N}}^{spin}$. We also treat exotic cases where the electric charges are given by representations beyond the (anti)fundamental of the $so_{2N}$ Lie algebra.

## 5.1 't Hooft line with magnetic charges $\mu_{N-1}$ and $\mu_N$

Besides the vectorial $\mu_1 = e_1$ given by eq(4.80), the $SO_{2N}$ has moreover two other minuscule coweights $\mu_{N-1}$ and $\mu_N$. These coweights yield the magnetic charges of the two spinorial-like 't Hooft lines :

$$tH_{\gamma_0}^{\mu_{N-1}} \quad , \quad tH_{\gamma_0}^{\mu_N}$$

These line defects are represented similarly to the vector-like line $tH_{\gamma_0}^{\mu_1}$ of previous section as depicted in Figure **16** where the $tH_{\gamma_0}^{\mu_N}$ couples to a vertical Wilson line $W_{\xi_z}^{\mathbf{R}}$ carrying internal states $|A\rangle$ belonging to some representation $\mathbf{R}$ of $so_{2N}$. Interesting candidates for $\mathbf{R}$ are given by the vectorial and the spinorials, namely

$$\mathbf{R} = 2N \quad , \quad \mathbf{R} = 2_L^{N-1} \quad , \quad \mathbf{R} = 2_R^{N-1} \quad , \quad \mathbf{R} = 2^N \qquad (5.109)$$

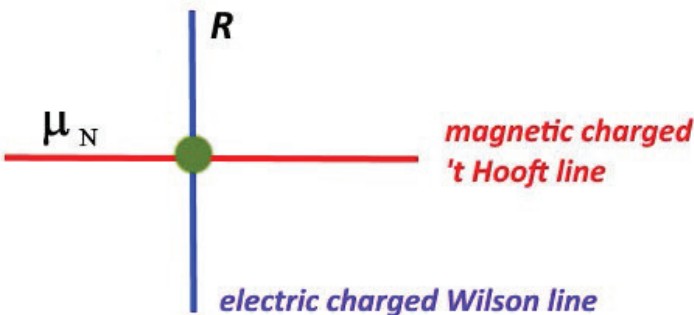

Figure 16: A horizontal 't Hooft line of D- type with spinor-like magnetic charge given by the minuscule coweight $\mu_N$ of $SO_{2N}$ couples to a vertical Wilson line characterized by a representation $\boldsymbol{R}$ of $so_{2N}$.

So depending on the electric charge of the Wilson line $W_{\xi_z}^{\boldsymbol{R}}$, one distinguishes various kinds of $\mathcal{L}$-operators that, generally speaking, can be labeled as follows

$$\mathcal{L}_{\boldsymbol{R}}^{\mu_s} = e^{X_{\boldsymbol{R}}} z^{\mu_s} e^{Y_{\boldsymbol{R}}} \tag{5.110}$$

with a spinor-like minuscule coweight $\mu_s$ of $SO_{2N}$. For an electric representation $\boldsymbol{R}$ with dimension $d_{\boldsymbol{R}}$, we have a Lax operator $\mathcal{L}_{\boldsymbol{R}}^{\mu}$ described by a $d_{\boldsymbol{R}} \times d_{\boldsymbol{R}}$ matrix whose entries are functions of the Darboux coordinates. These phase space coordinates labeled as $\left(b^{[ij]}, c_{[ij]}\right)$ appear in the expression of the $X_{\boldsymbol{R}}$ and $Y_{\boldsymbol{R}}$ as follows

$$X_{\boldsymbol{R}} = b^{[ij]} X_{[ij]}^{\boldsymbol{R}} \qquad , \qquad Y_{\boldsymbol{R}} = c_{[ij]} Y_{\boldsymbol{R}}^{[ij]} \tag{5.111}$$

Where in these expansions, the $X_{[ij]}^{\boldsymbol{R}}$ and $Y_{\boldsymbol{R}}^{[ij]}$ are generators of the nilpotent subalgebras $\mathbf{n}_{\pm}^{\boldsymbol{R}}$ issued from the Levi decomposition of $so_{2N}$. In fact, for the spinor-like coweights $\mu_s = \mu_{N-1}$ or $\mu_N$, we have the following Levi decomposition of $so_{2N}$

$$so_{2N} \rightarrow \boldsymbol{l}_{\mu_s} \oplus \boldsymbol{n}_+ \oplus \boldsymbol{n}_- \tag{5.112}$$

where $\boldsymbol{l}_{\mu_s} = gl_N$. This can be directly read from the Figure **17** where we see that the fundamental coweight $\mu_{N-1}$ is the dual of the simple root $\alpha_{N-1} = e_{N-1} - e_N$, while $\mu_N$ is the dual of $\alpha_N = e_{N-1} + e_N$.

Notice that by cutting the root $\alpha_{N-1}$ from **17-a**, we end up with the Dynkin diagram of an $sl_N$ Lie algebra with the following simple roots :

$$\alpha_1, ..., \alpha_{N-2}; \alpha_N \tag{5.113}$$

And if instead, we cut the root $\alpha_N$ as in **17-b**, we also end up with the Dynkin diagram of an $sl'_N$ Lie algebra having the simple roots :

$$\alpha_1, ..., \alpha_{N-2}; \alpha_{N-1} \tag{5.114}$$

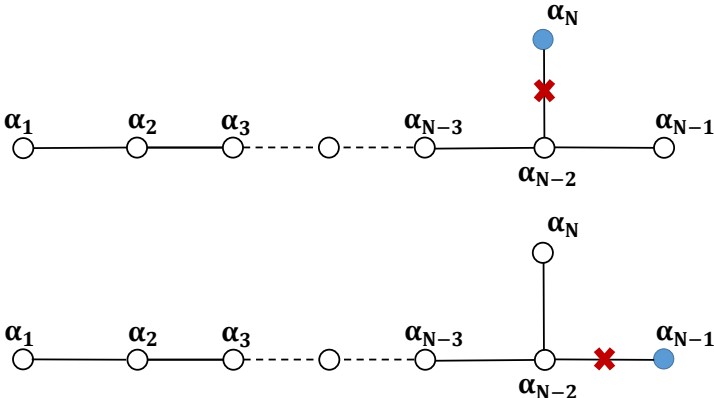

Figure 17: Dynkin diagram of $D_N$ Lie algebras where the two Levi decompositions with respect to the spinorial coweights are illustrated by : $(a)$ removing the simple root $\alpha_N$ for the minuscule coweight $\mu_N$. $(b)$ removing the simple root $\alpha_{N-1}$ for the minuscule coweight $\mu_{N-1}$.

The two $sl_N$ and $sl'_N$ are isomorphic, they are related by the exchange $\alpha_N \leftrightarrow \alpha_{N-1}$. We can therefore focus our analysis on the minuscule $tH_{\gamma_0}^{\mu_N}$ since the calculations are similar for $tH_{\gamma_0}^{\mu_{N-1}}$. Notice however that the expressions of the coweights in terms of the $e_i$ weight vector basis are given by

$$
\begin{aligned}
\mu_{N-1} &= \tfrac{1}{2}\left(e_1 + ... + e_{N-1} - e_N\right) \\
\mu_N &= \tfrac{1}{2}\left(e_1 + ... + e_{N-1} + e_N\right)
\end{aligned}
\tag{5.115}
$$

## 5.2  Magnetic charge $\mu_N$ and the link between $SO_{2N}$ and $SL_N$

Here, we study the Levi decomposition of $so_{2N}$ with respect to $\mu_N$ in order to explore intrinsic aspects of the coupling between the minuscule $tH_{\gamma_0}^{\mu_N}$ and the Wilson line in a representation $\mathbf{R}$ of $so_{2N}$ that is usually taken as the vectorial $\mathbf{2N}$. Particularly, we extend the results here for Wilson lines in the spinorial representation $\mathbf{2}^N$ where we build the graphic representation of their remarkable coupling with 't Hooft lines; see Figure **18-(a)**.

### 5.2.1  Spinorial 't Hooft line $tH_{\gamma_0}^{\mu_N}$

As shown by the Figure **17-a** without $\alpha_N$, there is a close relationship between $SO_{2N}$ and $SL_N$. It is given by the Levi decomposition $so_{2N} \rightarrow \boldsymbol{l}_{\mu_N} \oplus \boldsymbol{n}_+ \oplus \boldsymbol{n}_-$ with respect to the coweight $\mu_N$ of the $SO_{2N}$ gauge symmetry of the CS theory. In this decomposition, we have the following dimension splitting

$$
N\left(2N - 1\right) = N^2 + \frac{1}{2}N\left(N - 1\right) + \frac{1}{2}N\left(N - 1\right)
\tag{5.116}
$$

and the subalgebra structures

$$
\begin{aligned}
\boldsymbol{l}_{\mu_N} &= sl_1 \oplus sl_N \\
\boldsymbol{n}_+ &= \boldsymbol{N}_{+\frac{1}{2}} \wedge \boldsymbol{N}_{+\frac{1}{2}} \\
\boldsymbol{n}_- &= \boldsymbol{N}_{-\frac{1}{2}} \wedge \boldsymbol{N}_{-\frac{1}{2}}
\end{aligned}
\tag{5.117}
$$

with $sl_1 \oplus sl_N \sim gl_N$ and $[sl_1, \boldsymbol{n}_\pm] = \boldsymbol{n}_\pm$ indicating that

$$
\left[ sl_1, \boldsymbol{N}_{\pm\frac{1}{2}} \right] = \pm \frac{1}{2} \boldsymbol{N}_{\pm\frac{1}{2}}
\tag{5.118}
$$

We also have the $\boldsymbol{R}_{so_{2N}}$ representations' splitting

| repres $\boldsymbol{R}_{so_{2N}}$ | repres $\boldsymbol{R}_{gl_N}$ |
|---|---|
| $2\boldsymbol{N}$ | $\boldsymbol{N}_{+\frac{1}{2}} \oplus \boldsymbol{N}_{-\frac{1}{2}}$ |
| $2\boldsymbol{N} \wedge 2\boldsymbol{N}$ | $adj_0 \oplus \left( \boldsymbol{N}_{+\frac{1}{2}} \wedge \boldsymbol{N}_{+\frac{1}{2}} \right) \oplus \left( \boldsymbol{N}_{-\frac{1}{2}} \wedge \boldsymbol{N}_{-\frac{1}{2}} \right)$ |
| $2\boldsymbol{N} \vee 2\boldsymbol{N}$ | $adj_0 \oplus \left( \boldsymbol{N}_{+\frac{1}{2}} \vee \boldsymbol{N}_{+\frac{1}{2}} \right) \oplus \left( \boldsymbol{N}_{-\frac{1}{2}} \vee \boldsymbol{N}_{+\frac{1}{2}} \right)$ |
| $2^N$ | $\oplus_{k=0}^{N} \boldsymbol{N}_{q_k}^{\wedge k}$ |

$$
\tag{5.119}
$$

where $2\boldsymbol{N}$ describes vector- like states, $2\boldsymbol{N} \wedge 2\boldsymbol{N}$ the antisymmetric (adjoint) and $2\boldsymbol{N} \vee 2\boldsymbol{N}$ the symmetric. The $2^N$ states describe a Dirac-type spinor reducible into left handed and right handed Weyl spinors as follows

$$
2^N = 2_L^{N-1} \oplus 2_R^{N-1}
\tag{5.120}
$$

Notice that the wedge product $\wedge^k \boldsymbol{N}$ is the k-th anti-symmetrisation order (for short $\boldsymbol{N}^{\wedge k}$) of the tensor product of k representation $\boldsymbol{N}$. Its dimension is equal to $\frac{N!}{(N-k)!k!}$. As illustrating examples of the degrees of freedom described by such wedge products, we give below the reductions associated with the leading gauge symmetry groups

| $so_{2N}$ | $2\boldsymbol{N}$ | $2^N$ | $2_L^{N-1}$ | $2_R^{N-1}$ |
|---|---|---|---|---|
| $so_6$ | $6$ | $8$ | $4_L$ | $4_R$ |
| $so_8$ | $8$ | $16$ | $8_L$ | $8_R$ |
| $so_{10}$ | $10$ | $32$ | $16_L$ | $16_R$ |
| $so_{12}$ | $12$ | $64$ | $32_L$ | $32_R$ |

$$
\tag{5.121}
$$

where we have also given the $so_6$ which is isomorphic to $sl_4$ with no Levi charge operator $sl_1$. The Levi decompositions with respect to $\mu_N$ of the above spinorial representations $2^N$ are given by the sum of two bloks: $(i)$ the first block involving the even powers $\boldsymbol{N}^{\wedge 2l}$, it corresponds to Weyl spinor; say $2_L^{N-1}$. $(ii)$ the second block having the odd powers $\boldsymbol{N}^{\wedge 2l+1}$ and corresponding to $2_R^{N-1}$. So, we have:

| $so_{2N}$ | $2_L^{N-1}$ | $2_R^{N-1}$ |
|---|---|---|
| $so_6$ | $4_L = 1 + 3^{\wedge 2}$ | $4_R = 3^{\wedge 1} + 3^{\wedge 3}$ |
| $so_8$ | $8_L = 1 + 4^{\wedge 2} + 4^{\wedge 4}$ | $8_R = 4^{\wedge 1} + 4^{\wedge 3}$ |
| $so_{10}$ | $16_L = 1 + 5^{\wedge 2} + 5^{\wedge 4}$ | $16_R = 5^{\wedge 1} + 5^{\wedge 3} + 5^{\wedge 5}$ |
| $so_{12}$ | $32_L = 1 + 6^{\wedge 2} + 6^{\wedge 4} + 6^{\wedge 6}$ | $32_R = 6^{\wedge 1} + 6^{\wedge 3} + 6^{\wedge 5}$ |

$$
\tag{5.122}
$$

By assuming the $\mathbf{2}_L^{N-1}$ and the $\mathbf{2}_R^{N-1}$ as traceless, we can exhibit the Levi charges in the above relations leading to

| $so_{2N}$ | $\mathbf{2}_L^{N-1}$ | | $\mathbf{2}_R^{N-1}$ | |
|---|---|---|---|---|
| $so_6$ | $\mathbf{4}_L$ | $= 1_{+3/4}+3_{-1/4}$ | $\mathbf{4}_R$ | $= 3_{+1/4}+1_{-3/4}$ |
| $so_8$ | $\mathbf{8}_L$ | $= 1_{+1}+6_0+1_{-1}$ | $\mathbf{8}_R$ | $= 4_{+1/2}+4_{-1/2}$ |
| $so_{10}$ | $\mathbf{16}_L$ | $= 15_{5/4}+10_{+1/4}+5_{-3/4}$ | $\mathbf{16}_R$ | $= 5_{+3/4}+10_{-1/4}+1_{-5/4}$ |
| $so_{12}$ | $\mathbf{32}_L$ | $= 1_{+3/2}+15_{+1/2}+15_{-1/2}+1_{-3/2}$ | $\mathbf{32}_R$ | $= 6_{+1}+20_0+6_{-1}$ |

$$(5.123)$$

Thanks to the reduction of $so_{2N}$ representations in terms of $gl_N$ ones like in eqs(5.119), one can construct various kinds of spinorial-like Lax couplings depending on the electric representation $\mathbf{R}$ hosted by the Wilson line $W_{\gamma_z}^{\mathbf{R}}$ crossing the tH$_{\gamma_0}^{\mu_N}$ line. Two of such couplings are studied here below:

- *Case of electric $\mathbf{R}_s = \mathbf{2}^N$*

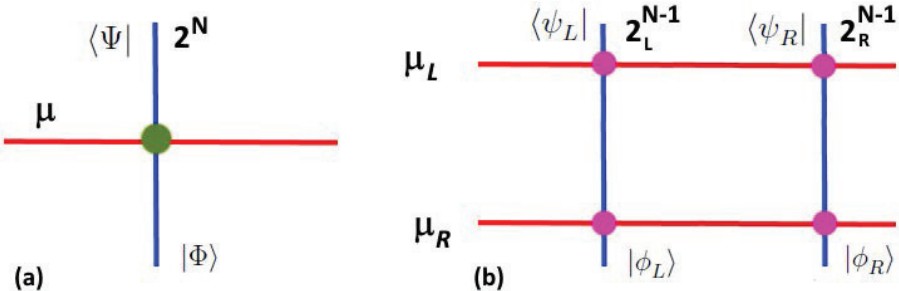

Figure 18: On the left, a horizontal spinorial like 't Hooft line crossing a vertical Wilson line carrying internal fermionic states $\Psi = (\psi_L, \psi_R)$. On the right, the structure of the coupling under the Levi decomposition showing chiral and antichiral Weyl states traveling along vertical lines.

In this case, the coupling is given by the interaction between the spinorial tH$_{\gamma_0}^{\mu_N}$ and a Wilson $W_{\gamma_z}^{\mathbf{R}}$ line with electric representation $\mathbf{R}_s = \mathbf{2}^N$ as illustrated by the Figure $\mathbf{18}$-$\mathbf{(a)}$. The quantum states propagating along the vertical Wilson line form a Dirac spinor $\Psi = \Psi_L \oplus \Psi_R$. By using the projector $\Pi_L$ on the left handed spinor and the projector $\Pi_R$ on the right handed one, we can use the properties $\Pi_L + \Pi_R = I_{id}$ and $\Pi_L \Pi_R = 0$ to decompose the action of the minuscule coweight on $\mathbf{2}^N$ like

$$\mu = \Pi_L \mu + \Pi_R \mu \qquad \leftrightarrow \qquad \mu = \mu_L + \mu_R \qquad (5.124)$$

This splitting is illustrated by the Figure $\mathbf{18}$-$\mathbf{(b)}$ where the states propagating in the two vertical Wilson lines are given by the left handed $\Psi_L$ and the right handed $\Psi_R$ Weyl spinors. In this case, the L-operator decomposes into four blocks as follows

$$\mathcal{L}_{\mathbf{R}_s}^{\mu} = \begin{pmatrix} \Pi_L \mathcal{L} \Pi_L & \Pi_L \mathcal{L} \Pi_R \\ \Pi_R \mathcal{L} \Pi_L & \Pi_R \mathcal{L} \Pi_R \end{pmatrix} \qquad (5.125)$$

Notice that in this expression of $\mathcal{L}^{\mu}_{\boldsymbol{R}_s}$, we have not yet implemented the the Levi decomposition; we have only exhibited the chiral and anti-chiral structure of the Dirac spinor. To implement the effect of the Levi decomposition, we introduce other types of projectors

$$P_{\mathbf{N}^{\wedge k}} = \left|\mathbf{N}^{\wedge k}\right\rangle \left\langle \mathbf{N}^{\wedge k}\right| \tag{5.126}$$

that give the reduction $\mathbf{2}^N = \oplus_k \mathbf{N}^{\wedge k}$ and eqs(5.119-5.123). This leads to a more complicated structure of this specific type of coupling; we will come back to this case later for further development.

- *Case of electric* $\boldsymbol{R}_v = \mathbf{2}N$

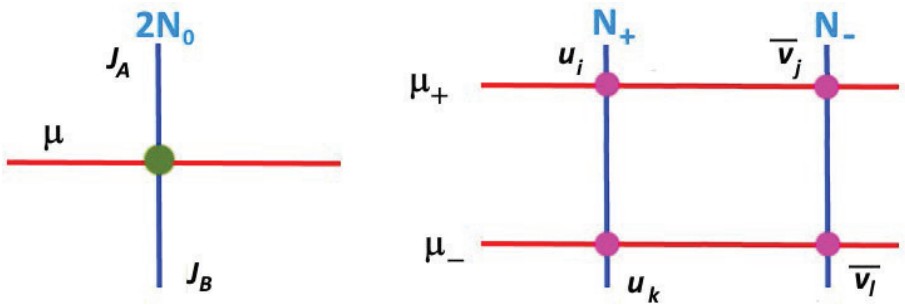

Figure 19: On the left, a horizontal spinorial like 't Hooft line crossing a vertical Wilson line carrying a bosonic current $J^A = \Psi\Gamma^A\Psi$. On the right, The splitting of the current into two currents $u_i = \Psi\Upsilon_i\Phi$ and $\bar{v}^i = \Psi\bar{\Upsilon}^i\Phi$ traveling along the vertical lines.

In this case, the spinorial tH$^{\mu_N}_{\gamma_0}$ crosses a Wilson $W^{\boldsymbol{R}}_{\xi_z}$ line with electric representation $\boldsymbol{R} = \mathbf{2}N$ as shown by the Figure **19-(a)**. This representation $\boldsymbol{R}$ can be related to the previous $\boldsymbol{R}_s = \mathbf{2}^N$ because it can also be viewed as an $so_{2N}$ electric current $J_A$ given by the Dirac bi-linear like

$$J_A = \langle\Psi|\,\Gamma_A\,|\Phi\rangle \tag{5.127}$$

where the 2N Gammas $\Gamma_A$ are $2^N \times 2^N$ Dirac matrices. To deal with this $so_{2N}$ Wilson lines, it is interesting to use the new basis

$$\Upsilon_l = \frac{1}{\sqrt{2}}\left(\Gamma_l + i\Gamma_{N+1}\right), \qquad \bar{\Upsilon}^l = \frac{1}{\sqrt{2}}\left(\Gamma_l - i\Gamma_{N+1}\right) \tag{5.128}$$

Then by putting back into (5.127), we find that the $so_{2N}$ electric current $J_A$ decomposes as two (covariant and contravariant) $gl_N$ currents given by

$$u_i = \Psi\Upsilon_i\Phi \quad , \quad \bar{v}^i = \Psi\bar{\Upsilon}^i\Phi \tag{5.129}$$

The $u_i$ transforms in the fundamental $N_+$ of the Levi subalgebra $gl_N$; and the $\bar{v}^i$ transforms in the anti- fundamental $N_-$. Using the projector $\varrho_+$ on the representation $N_+$ and the projector $\varrho_-$ on the representation $N_-$, we can express the L-operator as follows, see also Figure **19-(b)**.

$$\mathcal{L}^{\mu}_{\boldsymbol{R}_v} = \begin{pmatrix} \varrho_+\mathcal{L}\varrho_+ & \varrho_+\mathcal{L}\varrho_- \\ \varrho_-\mathcal{L}\varrho_+ & \varrho_-\mathcal{L}\varrho_- \end{pmatrix} \tag{5.130}$$

### 5.2.2 Levi and nilpotent subalgebras within $so_{2N}$

To model properties of the spinorial 't Hooft lines in 4D Chern-Simons theory with $SO_{2N}$ symmetry characterized by the following Levi decomposition with respect to $\mu_n$

$$so_{2N} \to (\boldsymbol{N}_- \wedge \boldsymbol{N}_-) \oplus gl_N \oplus (\boldsymbol{N}_+ \wedge \boldsymbol{N}_+) \tag{5.131}$$

where $\boldsymbol{N}_\pm$ stand for $\boldsymbol{N}_{\pm 1/2}$, it is interesting to recall some useful tools concerning the euclidian Dirac spinors in higher dimensions and the algebra of Gamma matrices.

In even 2N dimensions, the Dirac spinor $|\psi_{Dirac}\rangle$ has $2^N$ components and decomposes as a sum of two Weyl spinors like $|\psi_L\rangle + |\psi_R\rangle$ where

$$
\begin{aligned}
|\psi_L\rangle &= \Pi_L |\psi_{Dirac}\rangle \\
|\psi_R\rangle &= \Pi_R |\psi_{Dirac}\rangle
\end{aligned}
\tag{5.132}
$$

and

$$
\begin{aligned}
\Pi_L &= \tfrac{1}{2} (I + \Gamma_{2N+1}) \\
\Pi_r &= \tfrac{1}{2} (I - \Gamma_{2N+1})
\end{aligned}
\tag{5.133}
$$

The $\psi_L$ and $\psi_R$ are Weyl spinors transforming in $2_L^{N-1}$ and $2_R^{N-1}$ while the $\Pi_L$ and the $\Pi_R$ are the spin projectors encountered earlier reading as follows

$$\Gamma_L = \begin{pmatrix} \boldsymbol{I} & \boldsymbol{0} \\ \boldsymbol{0} & \boldsymbol{0} \end{pmatrix}, \qquad \Gamma_R = \begin{pmatrix} \boldsymbol{0} & \boldsymbol{0} \\ \boldsymbol{0} & \boldsymbol{I} \end{pmatrix} \tag{5.134}$$

The identity and the zeros appearing in these matrices live in $2^{N-1}$ dimensions. The $\Gamma_{2N+1}$ is the chiral operator given by

$$\Gamma_{A_1}\Gamma_{A_2}....\Gamma_{A_{2N}} = (i)^N \varepsilon_{A_1.....A_{2N}} \Gamma_{2N+1} \tag{5.135}$$

where $\varepsilon_{A_1.....A_{2N}}$ is the completely antisymmetric tensor with $\varepsilon_{1...2N} = 1$ and $\Gamma_A$ obeying the Clifford algebra of a 2N dimension euclidian space.

$$\Gamma_A\Gamma_B + \Gamma_B\Gamma_A = 2\delta_{AB} \tag{5.136}$$

The relations (5.131) and (5.135) allow to split the 2N Gamma matrices $\Gamma_A$ into two subsets that will be used later to construct a new basis for the Gammas that is compatible with $gl_N$,

$$
\begin{aligned}
\Gamma_i \\
\Gamma_{N+i}
\end{aligned}
\quad , \qquad i = 1, ..., N
\tag{5.137}
$$

Recall also that the generators $J_{[AB]}$ of the $so_{2N}$ spinor representation are defined by the commutators

$$\Gamma_{AB} = \frac{1}{2i} [\Gamma_A, \Gamma_B] \tag{5.138}$$

As for $sl_1 \oplus sl_N$, the $so_{2N}$ algebra also has N commuting diagonal generators $H_l$ realised in terms of the Gamma matrices as

$$H_l = \frac{1}{2i} [\Gamma_l, \Gamma_{N+l}] = -i\Gamma_l\Gamma_{N+l} \quad , \qquad l = 1, ..., N \tag{5.139}$$

To exhibit the realisation of the $sl_1 \oplus sl_N$ representations within the $so_{2N}$ orthogonal symmetry group, we substitute the spliting $\Gamma_A = (\Gamma_i, \Gamma_{N+l})$ into the $N(2N-1)$ generators $\Gamma_{AB}$ of $so_{2N}$ and we obtain the following antisymmetric 2×2 block matrix

$$\Gamma_{AB} = \begin{pmatrix} \Gamma_{[ij]} & \hat{\Gamma}_i^j \\ -\hat{\Gamma}_j^i & \tilde{\Gamma}^{[ij]} \end{pmatrix} \tag{5.140}$$

This decomposition contains:
(**a**) the $N^2$ operators $\hat{\Gamma}_i^j$ generating $N_+ \otimes N_-$ of the Levi subalgebra $sl_1 \oplus sl_N$.
(**b**) the $\frac{1}{2}N(N-1)$ operators $\Gamma_{[ij]}$ generating the $N_+ \wedge N_+$ nilpotent subalgebras.
(**c**) the $\frac{1}{2}N(N-1)$ operators $\bar{\Gamma}^{[ij]}$ generating the $N_- \wedge N_-$ dual nilpotent subalgebra.

## 5.3   Nilpotent subalgebras and L-operator

In order to explicitly realise the generators $\Gamma_{[ij]}, \tilde{\Gamma}^{[ij]}$ and $\hat{\Gamma}_i^j$ appearing in the decomposition (5.140) and consequently the generators $X_{[ij]}$ and $Y^{[kl]}$ of the nilpotent subalgebras $\mathbf{n}_\pm$, we first think of the set of the 2N Dirac matrices $\Gamma_A = (\Gamma_i, \Gamma_{N+1})$ as follows,

$$\Upsilon_l = \frac{1}{\sqrt{2}} \left( \Gamma_l + i\Gamma_{N+1} \right), \qquad \bar{\Upsilon}^l = \frac{1}{\sqrt{2}} \left( \Gamma_l - i\Gamma_{N+1} \right) \tag{5.141}$$

This new Gamma matrix basis satisfy the Clifford algebra

$$\begin{aligned}
\Upsilon_i \bar{\Upsilon}^j + \bar{\Upsilon}^j \Upsilon_i &= 2\delta_i^j \\
\Upsilon_i \Upsilon_j + \Upsilon_j \Upsilon_i &= 0 \\
\bar{\Upsilon}^k \bar{\Upsilon}^l + \bar{\Upsilon}^l \bar{\Upsilon}^k &= 0
\end{aligned} \tag{5.142}$$

Then, we consider the two $gl_N$ vector currents $u_i = \langle \xi | \Upsilon_i | \psi \rangle$ and $\bar{v}^i = \langle \psi | \bar{\Upsilon}^i | \xi \rangle$ of eq(5.129) constructed out of bilinears of the Dirac fermions and use them to construct $\Gamma_{[ij]}, \tilde{\Gamma}^{[ij]}$ and $\hat{\Gamma}_i^j$. These two currents transform in the $N_+$ and $N_-$ representation of $sl_1 \oplus sl_N$.

### 5.3.1   Realising the nilpotent generators of $n_\pm$

First, using the N+N complex variables $u_i$ and $\bar{v}^i$, we build the translation operators $\bar{\partial}^i = \partial/\partial u_i$ and $\partial_i = \partial/\partial \bar{v}^i$ as well as the rotations

$$\begin{aligned}
X_{[ij]} &= u_i \partial_j - u_j \partial_i & , & & Z_i^l &= u_i \bar{\partial}^l - \bar{v}^l \partial_i \\
Y^{[ij]} &= \bar{v}^i \bar{\partial}^j - \bar{v}^j \bar{\partial}^i & , & & H &= \frac{1}{2} Tr \left( Z_i^l \right)
\end{aligned} \tag{5.143}$$

In these relations, the operator

$$H = \frac{1}{2} \sum_i \left( u_i \bar{\partial}^i - \bar{v}^i \partial_i \right) \tag{5.144}$$

is the charge generator of $sl_1$. It acts on the complex variables like

$$H u_i = +\frac{1}{2} u_i, \qquad H \bar{v}^i = -\frac{1}{2} \bar{v}^i \tag{5.145}$$

We also have $X_{[ij]}\bar{v}^l = \delta^l_j u_i - \delta^l_i u_j$ and $Y^{[ij]}u_l = \delta^j_l \bar{v}^i - \delta^i_l \bar{v}^j$ as well as $Z^j_i u_k = u_i \delta^j_k$ and $Z^j_i \bar{v}^l = -\bar{v}^j \delta^l_i$. The above operators (5.143-5.144) obey interesting commutation relations such as

$$
\begin{aligned}
\left[X_{[ij]}, Y^{[kl]}\right] &= \left(\delta^k_j Z^l_i - \delta^k_i Z^l_j\right) - \left(\delta^l_j Z^k_i - \delta^l_i Z^k_j\right) \\
\left[X_{[ij]}, X_{[kl]}\right] &= 0 \\
\left[Y^{[ij]}, Y^{[kl]}\right] &= 0
\end{aligned}
\tag{5.146}
$$

For particular values of the labels, we obtain

$$
\begin{aligned}
\left[X_{[ij]}, Y^{[jl]}\right] &= (N-2)Z^l_i + 2\delta^l_i H \\
\left[Z^j_i, X_{kl}\right] &= +\left(\delta^j_k X_{[il]} - \delta^j_l X_{[ik]}\right) \\
\left[Z^j_i, Y^{[kl]}\right] &= -\left(\delta^k_i Y^{[jl]} - \delta^l_i Y^{[jk]}\right)
\end{aligned}
\tag{5.147}
$$

and

$$
\begin{aligned}
\left[H, X_{[kl]}\right] &= +X_{[kl]} \\
\left[H, Y^{[kl]}\right] &= -Y^{[kl]}
\end{aligned}
\tag{5.148}
$$

In order to introduce similar notations to the ones used in the previous sections, we associate to the variables $u_i$ and $\bar{v}^i$ the kets

$$
u_i \to \left|+\frac{1}{2}, i\right\rangle \quad , \quad \bar{v}^i \to \left|-\frac{1}{2}, i\right\rangle
\tag{5.149}
$$

and to the translation operators $\bar{\partial}^i = \partial/\partial u_i$ and $\partial_i = \partial/\partial \bar{v}^i$ the following bras

$$
\bar{\partial}^i \to \left\langle -\frac{1}{2}, i\right| \quad , \quad \partial_i \to \left\langle +\frac{1}{2}, i\right|
\tag{5.150}
$$

We use moreover the following notation

$$
\begin{aligned}
\langle -, j|+, i\rangle &= \delta^j_i & , \quad \langle +, j|+, i\rangle &= 0 \\
\langle +, j|-, i\rangle &= \delta^j_i & , \quad \langle -, j|-, i\rangle &= 0
\end{aligned}
\tag{5.151}
$$

to realise the operators $X_{[ij]}, Y^{[kl]}$ and $Z^l_i$ as

$$
\begin{aligned}
X_{[ij]} &= |+, i\rangle\langle +, j| - |+, j\rangle\langle +, i| \\
Y^{[kl]} &= |-, k\rangle\langle -, l| - |-, l\rangle\langle -, k| \\
Z^l_i &= |+, i\rangle\langle -, l| - |-, l\rangle\langle +, i|
\end{aligned}
\tag{5.152}
$$

We also have $X_{[ij]}Y^{[kl]} = U^{[kl]}_{[ij]}$ with

$$
U^{[kl]}_{[ij]} = \delta^k_j |+, i\rangle\langle -, l| - \delta^k_i |+, j\rangle\langle -, l| - \delta^l_j |+, i\rangle\langle -, k| + \delta^l_i |+, j\rangle\langle -, k|
\tag{5.153}
$$

as well as

$$
H = \frac{1}{2}\varrho^+ - \frac{1}{2}\varrho^-
\tag{5.154}
$$

where

$$
\begin{aligned}
\Pi^+ &= \sum_i \varrho_i^+ & , & & \varrho_i^+ &= |+, i\rangle\, \langle -, i| \\
\Pi^- &= \sum_i \varrho_i^- & , & & \varrho_i^- &= |-, i\rangle\, \langle +, i|
\end{aligned}
\tag{5.155}
$$

with the properties $\Pi^+ X_{[ij]} = X_{[ij]}$ and $Y^{[kl]}\Pi^+ = Y^{[kl]}$. Notice also that using (5.151), we have

$$
X_{ij}X_{kl} = 0 \qquad , \qquad Y^{[ij]}Y^{[kl]} = 0
\tag{5.156}
$$

and

$$
X_{[ij]}Y^{[jl]} = |+, i\rangle\, \langle -, l| + \delta_i^l \Pi^+
\tag{5.157}
$$

### 5.3.2 Building the Lax operator $\mathcal{L}_{\boldsymbol{R}_v}^{\mu_N}$

Now, we are finally able to explicitly calculate the expression of the spinorial Lax operator of the 4D CS theory with $SO_{2N}$ gauge symmetry. This operator $\mathcal{L}_{\boldsymbol{R}_v}^{\mu_N}$ describing the coupling of Figure **19-(b)** is generally given by

$$
\mathcal{L}_{\boldsymbol{R}_v}^{\mu_N} = e^{X_{\boldsymbol{R}_v}} z^{\mu_N} e^{Y_{\boldsymbol{R}_v}}
\tag{5.158}
$$

where the $X_{\boldsymbol{R}_v}$ and $Y_{\boldsymbol{R}_v}$ are $2N \times 2N$ matrices given by the following linear combinations

$$
X_{\boldsymbol{R}_v} = b^{[ij]} X_{[ij]}^{\boldsymbol{R}_v}, \qquad Y_{\boldsymbol{R}_v} = c_{[ij]} Y_{\boldsymbol{R}_v}^{[ij]}
\tag{5.159}
$$

such that the antisymmetric $b^{[ij]}$ and $c_{[ij]}$ are Darboux coordinates satisfying the Poisson Braket

$$
\left\{ b^{[ij]}, c_{[kl]} \right\}_{PB} = \delta_k^i \delta_l^j - \delta_l^i \delta_k^j
\tag{5.160}
$$

The adjoint form $\mu_N$ of the minuscule coweight in (5.158) is given by

$$
\mu_N = \frac{1}{2}\Pi^+ - \frac{1}{2}\Pi^-
\tag{5.161}
$$

where the projectors $\Pi^\pm$ are as given in (5.155) with the properties $\Pi^+ + \Pi^- = I_{id}$ and $\Pi^+\Pi^- = 0$. This allows us to write

$$
z^{\mu_N} = z^{\frac{1}{2}}\Pi^+ + z^{-\frac{1}{2}}\Pi^-
\tag{5.162}
$$

Moreover, because of the properties (5.156), the matrices X and Y (5.159) are nilpotent with degree 2, that is $X^2 = Y^2 = 0$. Therefore, the L-operator expands as

$$
\mathcal{L}_{\boldsymbol{R}_v}^{\mu_N} = z^{\mu_N} + X z^{\mu_N} + z^{\mu_N} Y + X z^{\mu_N} Y
\tag{5.163}
$$

By substituting $z^{\mu_N}$ by its expression (5.162) and using the properties $X\Pi^+ = 0$ and $\Pi^+ Y = 0$, we end up with

$$
\mathcal{L}_{\underline{\boldsymbol{R}}_v}^{\mu_N} = z^{\frac{1}{2}}\Pi^+ + z^{-\frac{1}{2}}\Pi^- + z^{-\frac{1}{2}}X\Pi^- + z^{-\frac{1}{2}}\Pi^- Y + z^{-\frac{1}{2}}X\Pi^- Y
\tag{5.164}
$$

And by putting $X = b^{[ij]}X_{[ij]}$ and $Y = c_{[kl]}Y^{[kl]}$, the Lax operator $\mathcal{L}^{\mu_N}_{\boldsymbol{R}_v}$ can be also expressed like

$$
\begin{aligned}
\mathcal{L}^{\mu_N}_{\boldsymbol{R}_v} &= z^{\frac{1}{2}}\Pi^+ + z^{-\frac{1}{2}}\Pi^- + 8z^{-\frac{1}{2}}(b^{[ik]}E^j_i c_{[kj]}) \\
&\quad + (2z^{-\frac{1}{2}}b^{[ij]})X_{[ij]}\Pi^- + (2z^{-\frac{1}{2}}c_{[kl]})\Pi^- Y^{[kl]}
\end{aligned}
\tag{5.165}
$$

where $E^k_i = |+,i\rangle\langle -,k|$. Moreover, using

$$
\begin{aligned}
Tr\left(X_{[ij]}Y^{[kl]}\right) &= 2\left(\delta^l_i\delta^k_j - \delta^l_j\delta^k_i\right) \\
Tr\left(XY^{[ij]}\right) &= -2b^{[ij]} \\
Tr\left(X_{[ij]}Y\right) &= -2c_{[ij]}
\end{aligned}
\tag{5.166}
$$

we have

$$
b^{[ij]} = -\frac{1}{4}z^{\frac{1}{2}}Tr\left(Y^{[ij]}\mathcal{L}^{\mu_N}\right), \qquad c_{[ij]} = -\frac{1}{4}z^{\frac{1}{2}}Tr\left(X_{[ij]}\mathcal{L}^{\mu_N}\right)
\tag{5.167}
$$

The expression of the L-operator in the basis $|+,i\rangle, |-,j\rangle$ defined in eq(5.140) reads as follows

$$
\mathcal{L}^{\mu_N}_{\boldsymbol{R}_v} = z^{-\frac{1}{2}}\begin{pmatrix} 2c_{[ij]} & z\delta^i_j + 8b^{[ik]}c_{[kj]} \\ \delta^i_j & 2b^{[ij]} \end{pmatrix}
\tag{5.168}
$$

## 5.4 Topological quiver $Q^{\mu_N}_{\boldsymbol{R}_v}$ of $\mathcal{L}^{\mu_N}_{\boldsymbol{R}_v}$

In order to construct the topological gauge quiver $Q^{\mu_N}_{\boldsymbol{R}_v}$ associated to the spinor coweight and the fundamental representation of the D-type symmetry, we begin by rewriting the $\mathcal{L}^{\mu_N}_{\boldsymbol{R}_v}$ in the projector basis $(\Pi^+, \Pi^-)$ of the representation $2N = N_+ \oplus N_-$.
Using the properties of the $gl_N$ projectors on $N_+\oplus N_-$, in particular $(\Pi^+)^2 = \Pi^+$, $(\Pi^-)^2 = \Pi^-$ and

$$
\Pi^+ + \Pi^- = I_{id} \quad , \quad \Pi^+\Pi^- = 0
\tag{5.169}
$$

as well as $\Pi^+ X = X$ and $Y\Pi^+ = Y$, we can rewrite the Lax operator (5.164) as follows

$$
\mathcal{L}^{\mu_N}_{\underline{\boldsymbol{R}}_v} = \begin{pmatrix} z^{\frac{1}{2}}\Pi^+ + z^{-\frac{1}{2}}\Pi^+ X\Pi^- Y\Pi^+ & z^{-\frac{1}{2}}X\Pi^- \\ z^{-\frac{1}{2}}\Pi^- Y & z^{-\frac{1}{2}}\Pi^- \end{pmatrix}
\tag{5.170}
$$

Moreover, by using the remarkable properties $X\Pi^- = X$ and $\Pi^- Y = Y$ that can be checked with the explicit realisations $X_{[ij]} = |+,i\rangle\langle +,j| - |+,j\rangle\langle +,i|$ and $Y^{[kl]} = |-,k\rangle\langle -,l| - |-,l\rangle\langle -,k|$, the term $X\Pi^- Y\Pi^+$ reduces to $XY\Pi^+$ and the eq(5.170) becomes

$$
\mathcal{L}^{\mu_N}_{\boldsymbol{R}_v} = z^{-\frac{1}{2}}\begin{pmatrix} z + \Pi^+(XY)\Pi^+ & X\Pi^- \\ \Pi^- Y & \Pi^- \end{pmatrix}
\tag{5.171}
$$

The nodes $\mathcal{N}_1$ and $\mathcal{N}_2$ of the topological gauge quiver $Q^{\mu_N}_{\boldsymbol{R}_v}$ representing $\mathcal{L}^{\mu_N}_{\boldsymbol{R}_v}$ as depicted in Figure **20** are given by the diagonal entries of the matrix (5.171)

$$
\mathcal{N}_1 \equiv \Pi_+\mathcal{L}\Pi_+ \quad , \quad \mathcal{N}_2 \equiv \Pi_-\mathcal{L}\Pi_-
\tag{5.172}
$$

They are interpreted in terms of topological self-dual matter in the sense that they have no $sl_1$ Levi charge. This feature is manifestly exhibited by their dependence into the monomials

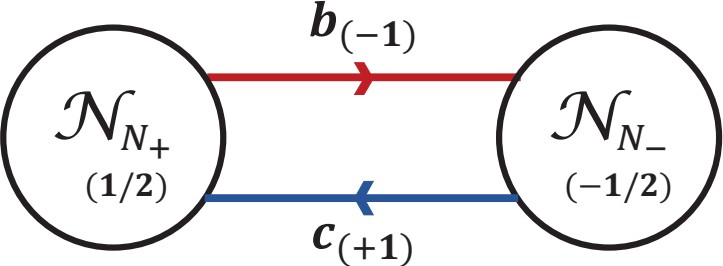

Figure 20: The topological quiver $Q^{\mu_N}_{\boldsymbol{R}_v}$ representing the operator $\mathcal{L}^{\mu_N}_{\boldsymbol{R}_v}$. It has 2 nodes $\mathcal{N}_1, \mathcal{N}_2$; and 2 links $L_{12}, L_{21}$. The nodes describe self-dual topological matter and the links describe topological bi-matter.

$b^{[ik]}c_{[kj]}$ that are neutral under $sl_1$ because the Darboux coordinates $b^{[ik]}$ and $c_{[kj]}$ have opposite charges. On the other hand, the two links are given by

$$L_{1\to2} \equiv \Pi_+ \mathcal{L} \Pi_- \qquad , \qquad L_{2\to1} \equiv \Pi_- \mathcal{L} \Pi_+ \tag{5.173}$$

They are remarkably equivalent to the Darboux coordinates $b^{[ij]}$ and $c_{[ij]}$ and are interpreted in terms of topological bi-fundamental matter of $sl_1 \oplus sl_N$. The $sl_1$ charges data for the $Q^{\mu_N}_{\boldsymbol{R}_v}$ is collected in the following table

$$\begin{array}{|c|c|c|c|c|}
\hline
\text{Quiver} & \mathcal{N}_1 & \mathcal{N}_2 & L_{1\to2} & L_{2\to1} \\
\hline
sl_1 & +\frac{1}{2} & -\frac{1}{2} & -1 & +1 \\
\hline
\end{array} \tag{5.174}$$

where we remark that the transition from the topological quiver node $\mathcal{N}_1$ to the $\mathcal{N}_2$ is given by the link $L_{1\to2}$ carrying a Levi charge $-1$; while the reverse transition is given by the link $L_{2\to1}$ with Levi charge $+1$.

# 6 Exceptional $E_6$ 't Hooft lines

This section is dedicated to the 4D Chern-Simons having as gauge symmetry the $E_6$ group. This case is characterized by two minuscule 't Hooft lines $tH^{\mu_1}_{\gamma_0}$ and $tH^{\mu_5}_{\gamma_0}$, and therefore two types of minuscule Lax operators $\mathcal{L}^{\mu_1}_{e_6}$ and $\mathcal{L}^{\mu_5}_{E_6}$ that we need to study in order to build the associated topological gauge quivers $Q^\mu_{e_6}$.

## 6.1 Minuscule coweights and Levi subalgebras of $E_6$

We begin by describing the interesting properties of the finite dimensional exceptional Lie algebra $\boldsymbol{e}_6$ that are useful for our construction. This is a simply laced Lie algebra with

dimension 78 and rank 6; its algebraic properties are described by the root system $\Phi_{e_6}$ generated by six simple roots $\alpha_i$. The intersection between these simple roots is represented in the Dynkin diagram $\mathcal{D}_{e_6}$ depicted in the Figure **21** and having the symmetric Cartan matrix $K_{e_6} = \alpha_i . \alpha_j$ given by :

$$
K_{e_6} = \begin{pmatrix}
2 & -1 & 0 & 0 & 0 & 0 \\
-1 & 2 & -1 & 0 & 0 & 0 \\
0 & -1 & 2 & -1 & 0 & -1 \\
0 & 0 & -1 & 2 & -1 & 0 \\
0 & 0 & 0 & -1 & 2 & 0 \\
0 & 0 & -1 & 0 & 0 & 2
\end{pmatrix}
\tag{6.175}
$$

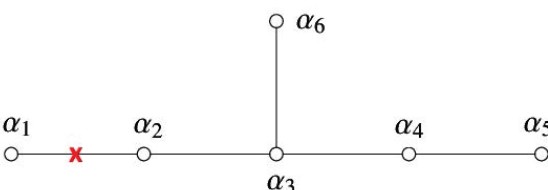

Figure 21: The Dynkin Diagram of $e_6$ having six nodes labeled by the simple roots $\alpha_i$. The cross ($\times$) indicates the cutted node in the Levi decomposition with respect to $\mu_1$, the Levi subalgebra in this case is given by $so(10) \oplus so(2)$.

The root system $\Phi_{e_6}$ contains 72 roots generated by the simple root basis $\{\alpha_i\}_{1 \leq i \leq 6}$, it has 36 positive roots $\alpha \in \Phi_{e_6}^+$ and 36 negative ones $-\alpha \in \Phi_{e_6}^-$. All of these roots have length $\alpha^2 = 2$ and are realised in the Euclidean $\mathbb{R}^8$ generated by the unit vector basis $\{\epsilon_i\}_{1 \leq i \leq 8}$ as follows

$$
\begin{array}{rcll}
\text{E}_6 \quad : \quad \alpha_1 &=& \frac{1}{2}\left(\epsilon_1 - \epsilon_2 - \epsilon_3 - \epsilon_4 - \epsilon_5 - \epsilon_6 - \epsilon_7 + \epsilon_8\right) & \\
\alpha_i &=& \epsilon_i - \epsilon_{i-1} \quad , \quad i = 1, 2, 3, 4, 5 & \\
\alpha_6 &=& \epsilon_1 + \epsilon_2 &
\end{array}
\tag{6.176}
$$

From the Figure **21**, we learn that the Dynkin diagram $\mathcal{D}_{e_6}$ is invariant under a manifest $\mathbb{Z}_2^{aut}$ outer- automorphism symmetry exchanging four simple roots and leaving invariant $\alpha_3$ and $\alpha_6$. It acts like $\alpha_i \rightarrow \alpha_{6-i}$ with $i = 1, ..., 5$, by exchanging $\alpha_2$ with $\alpha_4$ and $\alpha_1$ with $\alpha_5$. In permutation symmetry language, the $\mathbb{Z}_2^{aut}$ is generated by the double transposition $(15)(24)$, i.e:

$$
\mathbb{Z}_2^{aut} = \{I_{id}, (15)(24)\}
\tag{6.177}
$$

The 36+36 roots $\alpha$ of the $e_6$ Lie algebra can be organised as follows

| root | realisation | labels | number |
|------|-------------|--------|--------|
| $\beta_{ij}^+$ | $+\epsilon_i + \epsilon_j$ | $1 \leq j < i \leq 5$ | 20 |
| $\beta_{ij}^-$ | $-\epsilon_i - \epsilon_j$ | $2 \leq j < i \leq 5$ | 20 |
| $\gamma_{q_i}^+$ | $+\frac{1}{2}\left(q_i \epsilon_i - \epsilon_6 - \epsilon_7 + \epsilon_8\right)$ | $\Pi_{i=1}^5 q_i = 1$ | 16 |
| $\gamma_{q_i}^-$ | $-\frac{1}{2}\left(q_i \epsilon_i - \epsilon_6 - \epsilon_7 + \epsilon_8\right)$ | $\Pi_{i=1}^5 q_i = 1$ | 16 |

$$
\tag{6.178}
$$

where the five $q_i$ can take the values $\pm 1$ with the constraint $\Pi q_i = 1$.

Regarding the fundamental coweights $\omega_i$ of the six fundamental representations of the Lie algebra of $E_6$, they are given by the duality relation $\omega^i.\alpha_j = \delta^i_j$; this equation can either be solved in terms of roots, or by using the weight unit vectors $\epsilon_l$. The $\omega_i$ read in terms of the simple roots as follows

| fund- $\omega_i$ | in terms of roots | height | Repres |
|---|---|---|---|
| $\omega_1$ | $\frac{4}{3}\alpha_1 + \frac{5}{3}\alpha_2 + 2\alpha_3 + \frac{4}{3}\alpha_4 + \frac{2}{3}\alpha_5 + \alpha_6$ | 8 | $27_+$ |
| $\omega_2$ | $\frac{5}{3}\alpha_1 + \frac{10}{3}\alpha_2 + 4\alpha_3 + \frac{8}{3}\alpha_4 + \frac{4}{3}\alpha_5 + 2\alpha_6$ | 15 | $351_+$ |
| $\omega_3$ | $2\alpha_1 + 4\alpha_2 + 6\alpha_3 + 4\alpha_4 + 2\alpha_5 + 3\alpha_6$ | 21 | $2925_0$ |
| $\omega_4$ | $\frac{4}{3}\alpha_1 + \frac{8}{3}\alpha_2 + 4\alpha_3 + \frac{10}{3}\alpha_4 + \frac{5}{3}\alpha_5 + 2\alpha_6$ | 15 | $351_-$ |
| $\omega_5$ | $\frac{2}{3}\alpha_1 + \frac{4}{3}\alpha_2 + 2\alpha_3 + \frac{5}{3}\alpha_4 + \frac{4}{3}\alpha_5 + \alpha_6$ | 8 | $27_-$ |
| $\omega_6$ | $\alpha_1 + 2\alpha_2 + 3\alpha_3 + 2\alpha_4 + \alpha_5 + 2\alpha_6$ | 11 | $78_0$ |

$$(6.179)$$

From these expressions, we see that the outer-automorphism symmetry $\mathbb{Z}_2^{aut}$ discussed above can be manifestly exhibited as follows,

$$
\begin{aligned}
\omega_1 + \omega_5 &= 2\left(\alpha_1 + \alpha_5\right) + 3\left(\alpha_2 + \alpha_4\right) + 4\alpha_3 + 2\alpha_6 \\
\omega_2 + \omega_4 &= 3\left(\alpha_1 + \alpha_5\right) + 6\left(\alpha_2 + \alpha_4\right) + 8\alpha_3 + 4\alpha_6 \\
\omega_3 &= 2\left(\alpha_1 + \alpha_5\right) + 4\left(\alpha_2 + \alpha_4\right) + 6\alpha_3 + 3\alpha_6 \\
\omega_6 &= \left(\alpha_1 + \alpha_5\right) + 2\left(\alpha_2 + \alpha_4\right) + 3\alpha_3 + 2\alpha_6
\end{aligned}
$$

$$(6.180)$$

Moreover, by using (6.176) and $\alpha_i \to \alpha_{6-i}$ with $\alpha_0 \equiv \alpha_6$, one can write down the action of the outer-automorphism symmetry $\mathbb{Z}_2^{aut}$ on the weight vector basis $\epsilon_i$. In what follows, we will be particularly interested into: (1) the representation $78_0$, associated with the simple root $\alpha_6$, and (2) the $27_\pm$ associated with $\alpha_1$ and $\alpha_5$.

The two minuscule coweights $\mu_1$ and $\mu_5$ that are dual to the $\alpha_1$ and $\alpha_5$ of the $e_6$ are respectively associated with the fundamentals $27_+$ and $27_-$ as shown in table (6.179). Being related by $\mathbb{Z}_2^{aut}$, we focus below on one of the two minuscule coweights, say $\mu = \omega_1$; Similar results can be derived for $\mu_5$.

### 6.1.1   The $e_6$ algebra and the representation 78

There are different ways to decompose the root system of the $e_6$ Lie algebra. The interesting Levi decomposition with respect to charges of the minuscule coweight $\mu = \mu_1$ considered here reads as follows

$$e_6 \to so_2 \oplus so_{10} \oplus \mathbf{16}_+ \oplus \mathbf{16}_- \qquad (6.181)$$

From this splitting, we learn that the Levi subalgebra $\boldsymbol{l}_\mu = so_2 \oplus so_{10}$ and the nilpotent subalgebras $\boldsymbol{n}_\pm = \mathbf{16}_\pm$. The root system $\Phi_{e_6}$ containing the 72 roots of $e_6$ is therefore decomposed in terms of two subsets: a subset $\Phi_{so_{10}}$, and a subset given by the complement $\Phi_{e_6} \backslash \Phi_{so_{10}}$; they are described here below as they play an important role in the construction of the Lax operator $\mathcal{L}^\mu_{e_6}$.

- *Roots within $\Phi_{so_{10}}$*

The subset $\Phi_{so_{10}}$ contains 40 roots $\beta_{so_{10}}$, 20 positive and 20 negative; they define the step operators $Z_{\pm\beta_{so_{10}}}$ generating $so_{10}$ within $\boldsymbol{e}_6$. It is generated by the simple roots

$$\alpha_2, \quad \alpha_3, \quad \alpha_4, \quad \alpha_5, \quad \alpha_6 \tag{6.182}$$

and has the usual symmetry properties of the root system of $so_{10}$. The root subsystem $\Phi_{so_{10}} \subset \Phi_{e_6}$ can be defined as containing the roots $\beta_{so_{10}}$ with no dependence into $\alpha_1$, formally

$$\frac{\delta\beta_{so_{10}}}{\delta\alpha_1} = 0 \tag{6.183}$$

This can be noticed by cutting the node $\alpha_1$ in the Dynkin diagram of the Figure **21**, where we recover the Dynkin diagram of $so_{10}$ and a free node $\alpha_1$ associated with the $so_{10}$ spinor representations $16_\pm$ charged under $so_2$.

- *Roots outside $\Phi_{so_{10}}$*

This is the complementary subset of $\Phi_{so_{10}}$ within $\Phi_{e_6}$; it is given by $\Phi_{e_6}\backslash\Phi_{so_{10}}$ and reads directly from the root system of $e_6$ by considering only the roots $\beta_{e_6}$ with a dependence into $\alpha_1$ :

$$\delta\beta_{so_{10}}/\delta\alpha_1 \neq 0 \tag{6.184}$$

This subset contains 32 roots of spinorial type as they linearly depend on the simple root $\alpha_1$ which is spinorial-like. The importance of these roots is that they define the 16 step operators $X_{+\beta}$ generating the nilpotent subalgebra $\mathbf{16}_+$ and 16 step operators $X_{-\beta} = Y^\beta$ generating $\mathbf{16}_-$.

### 6.1.2 Decomposing the representation 27

As for the adjoint representation of $e_6$, the fundamental representation also decomposes in terms of representations of $so_2 \oplus so_{10}$. This representation is interesting in our study as it will be taking as the electric charge of the Wilson line $W_{\xi_z}^{\boldsymbol{R}}$ where $\boldsymbol{R} = \mathbf{27}_+$. Generally speaking, given a representation $\boldsymbol{R}_{e_6}$ of the algebra $e_6$, it can be decomposed into a direct sum of representations of $so_2 \oplus so_{10}$. such as

$$\boldsymbol{R}_{e_6} = \sum_l n_l \left(\boldsymbol{R}_l^{so_{10}}, \boldsymbol{R}_l^{so_2}\right) \tag{6.185}$$

where $n_l$ are some positive integers. In the case of the representation $\boldsymbol{R}_{e_6} = 27$, we have the following reduction [68]

$$\mathbf{27} = (\mathbf{1}, -\frac{4}{3}) + (\mathbf{10}, +\frac{2}{3}) + (\mathbf{16}, -\frac{1}{3}) \tag{6.186}$$

that we can simply write as $27 = 1_{-4/3} + 10_{2/3} + 16_{-1/3}$. Notice that by cutting the simple root $\alpha_1$ in the Dynkin diagram, the $SO_{10}$ representations get charged under $SO_2$; these charges play the role of a "glue" between these representations within the 27. This property

is manifested by the constraint that the sum (or the trace) of the charges of the 27 states with respect to $SO_2 \sim E_6/SO_{10}$ must vanish. Notice moreover that these charges can be also observed in the following relation

$$\omega_1 - \omega_5 = \frac{2}{3}\alpha_1 + \frac{1}{3}\alpha_2 - \frac{1}{3}\alpha_4 - \frac{2}{3}\alpha_5 \tag{6.187}$$

where $\alpha_2$ stands for the spinorial of $SO_{10}$ and $\alpha_5$ for the vectorial.

In order to understand the structure of the 27 states in the fundamental representation of $e_6$, we refer to the weight diagram of the Figure **22** where we have a top state $|\xi_1\rangle$ with weight $\xi_1 = \omega_1$ and a bottom state $\xi_{27} = -\omega_5$. The other 25 states in between can be generated either by starting from the $|\xi_1\rangle$ and successively acting on it by the step operators $(E_\beta)^\dagger = E_{-\beta}$ where $\beta$ a positive root of $e_6$, or by acting on the bottom state $|\xi_{27}\rangle$ with $(E_{-\beta})^\dagger = E_\beta$. The subspaces of the **27** representation correspond in the figure **22** to :

$$
\begin{array}{ll}
|\mathbf{1}\rangle & |\xi_1\rangle_{-4/3} \;\; = |\omega_1\rangle \\
\downarrow & \\
|\mathbf{16}\rangle & |\xi_\alpha\rangle_{+1/3} \\
\downarrow & \\
|\mathbf{10}\rangle & |\xi_i\rangle_{-2/3}
\end{array}
\tag{6.188}
$$

such that the top state $|\xi_1\rangle$ is an $SO_{10}$ singlet, the 16 states $|\xi_2\rangle,...,|\xi_{17}\rangle$ constitute a chiral spinor of $SO_{10}$, and the ten states $|\xi_{18}\rangle,...,|\xi_{27}\rangle$ form a vector of $SO_{10}$.

## 6.2 Minuscule $E_6$ 't Hooft operator

We can now use the collected mathematical tools concerning the exceptional Lie algebra $e_6$ to calculate the Lax operator $\mathcal{L}^\mu_{e_6}$ describing the coupling of an exceptional minuscule 't Hooft line $tH^\mu_{\gamma_0}$ with magnetic charge $\mu = \mu_1$ interacting with a Wilson line $W^{\boldsymbol{R}}_{\xi_z}$ with electric charge $\boldsymbol{R} = \mathbf{27}$.

### 6.2.1 Realizing the generators of the nilpotent subalgebras

To construct the 't Hoof line operator $\mathcal{L}^\mu_{e_6}$ of the exceptional $E_6$ Chern-Simons theory in 4D, we begin by building the generators of the nilpotent subalgebras that appear in the Levi factorisation -based formula [52]

$$\mathcal{L}^\mu_{E_6} = e^X z^\mu e^Y \tag{6.189}$$

where $\mu = \omega_1$ and the nilpotent matrix operators are given by

$$X = \sum_{\beta=1}^{16} b^\beta X_\beta \quad , \quad Y = \sum_{\beta=1}^{16} c_\beta Y^\beta \tag{6.190}$$

In these expansions, the sixteen $b^\beta$ and the sixteen $c_\beta$ are the 16+16 Darboux coordinates of the phase space of the exceptional $E_6$ 't Hooft line $tH^\mu_{\gamma_0}$. They satisfy the Poisson bracket

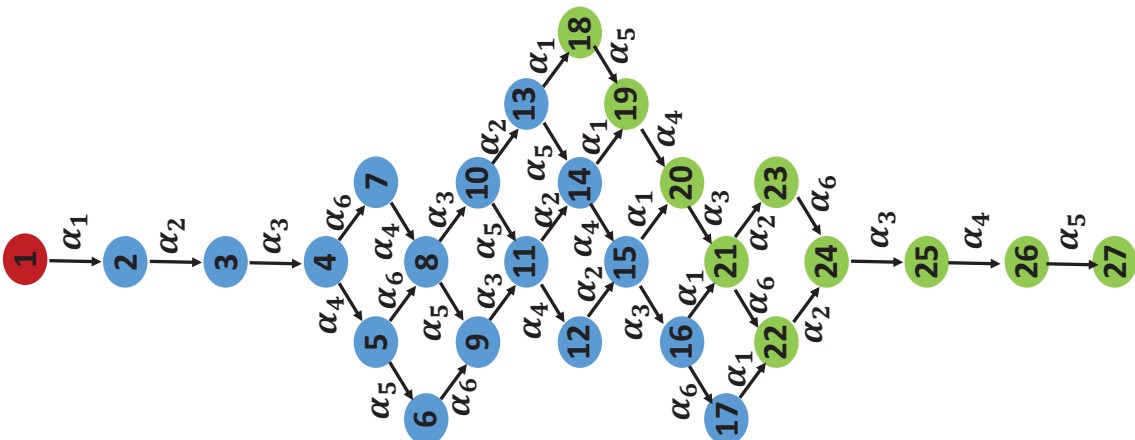

Figure 22: The weight diagram of the representation 27 of the exceptional Lie algebra $e_6$ where every state $|\xi_A\rangle$ is simply represented by the node carrying its number. The states 1+16+10 of the $SO_{10}$ sub-representations of $e_6$ are represented by different colors.

$\{b^\gamma, c_\beta\} = \delta^\gamma_\beta$ that must be promoted to a commutator in the study of interacting quantum lines. $X_\beta$ and $Y^\beta$ are the generators of the nilpotent subalgebras $\mathbf{16}_+$ and $\mathbf{16}_-$. The charge operator $\mu$ of the Levi subalgebra associated with the minuscule coweight can be presented as

$$\mu = -\frac{4}{3}\varrho_{\mathbf{1}} + \frac{2}{3}\varrho_{\mathbf{10}} - \frac{1}{3}\varrho_{\mathbf{16}} \tag{6.191}$$

where $\varrho_{\mathbf{1}}$, $\varrho_{\mathbf{10}}$ and $\varrho_{\mathbf{16}}$ are projectors on the $so_2 \oplus so_{10}$ representation subspaces making the $\mathbf{27}$ fundamental of $E_6$ as given by eq(6.186). By denoting the 27 states $|\xi_A\rangle$ of this representation as

| Groups | $E_6$ | $SO_{10} \times SO_2$ | | |
|---------|-------|-------|-------|-------|
| States | $|\xi_A\rangle$ | $|v_i\rangle$ | $|s_\alpha\rangle$ | $|\varphi\rangle$ |
| Repres | $\mathbf{27}_0$ | $\mathbf{10}_{+2/3}$ | $\mathbf{16}_{-1/3}$ | $\mathbf{1}_{-4/3}$ |

$$\tag{6.192}$$

following the splitting formally represented in the picture $\mathbf{23}$,

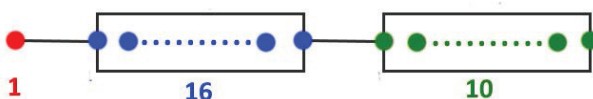

Figure 23: A graphical illustration of the Levi decomposition of the representation 27 of $e_6$ in terms of representations of $so_{10}$.

we can write the projectors $\varrho_{\mathbf{R}}$ on the fundamental representation of $e_6$ as

$$\varrho_{\mathbf{10}} = \sum_{l=1}^{10} |v_l\rangle \langle v^l|, \qquad \varrho_{\mathbf{16}} = \sum_{\beta=1}^{16} |s_\beta\rangle \langle s^\beta|, \qquad \varrho_{\mathbf{1}} = |\varphi\rangle \langle \varphi| \qquad (6.193)$$

Using the state basis kets $|v_l\rangle$, $|s_\beta\rangle$ and $|\varphi\rangle$ satisfying the orthogonality properties $\langle\varphi|v_l\rangle = \langle\varphi|s_\beta\rangle = \langle v_l|s_\beta\rangle = 0$, we realise the generators $X_\beta$ and $Y^\beta$ of the nilpotent subalgebras like

$$\begin{aligned} X_\beta &= |v_i\rangle (\Gamma^i)_{\beta\gamma} \langle s^\gamma| + |s_\beta\rangle \langle\varphi| \\ Y^\beta &= |\varphi\rangle \langle s^\beta| + |s_\gamma\rangle (\Gamma_i)^{\beta\gamma} \langle v^i| \end{aligned} \qquad (6.194)$$

where the $\Gamma_i$'s are Gamma matrices satisfying the usual Clifford algebra in ten dimensional space, namely $\Gamma_i\Gamma_j + \Gamma_j\Gamma_i = 2\delta_{ij}$. Moreover, if we adobt the short notations $|\mathbf{1}\rangle$, $|\mathbf{10}\rangle$ and $|\mathbf{16}\rangle$ to refer to the singlet state $|\varphi\rangle$, the vector $|v_l\rangle$ and the spinor $|s_\beta\rangle$, we can express the projectors more simply like $\varrho_{\mathbf{1}} = |\mathbf{1}\rangle \langle\mathbf{1}|$, and $\varrho_{\mathbf{10}} = |\mathbf{10}\rangle \langle\mathbf{10}|$ as well as $\varrho_{\mathbf{16}} = |\mathbf{16}\rangle \langle\mathbf{16}|$. Then, we also end up with the following expressions for the nilpotent generators (6.194) :

$$\begin{aligned} X_\beta &= |\mathbf{10}\rangle \langle\mathbf{16}| + |\mathbf{16}\rangle \langle\mathbf{1}| \\ Y^\beta &= |\mathbf{1}\rangle \langle\mathbf{16}| + |\mathbf{16}\rangle \langle\mathbf{10}| \\ \mu &= \tfrac{2}{3} |\mathbf{10}\rangle \langle\mathbf{10}| - \tfrac{1}{3} |\mathbf{16}\rangle \langle\mathbf{16}| - \tfrac{4}{3} |\mathbf{1}\rangle \langle\mathbf{1}| \end{aligned} \qquad (6.195)$$

We can check that this realisation solves the Levi decomposition constraints, namely

$$[\mu, X_\beta] = X_\beta \qquad , \qquad [\mu, Y^\beta] = -Y^\beta \qquad (6.196)$$

We have for example $\mu X_\beta = \tfrac{2}{3} |\mathbf{10}\rangle \langle\mathbf{16}| - \tfrac{1}{3} |\mathbf{16}\rangle \langle\mathbf{1}|$ and $X_\beta\mu = -\tfrac{1}{3} |\mathbf{10}\rangle \langle\mathbf{16}| - \tfrac{4}{3} |\mathbf{16}\rangle \langle\mathbf{1}|$, thus leading to $[\mu, X_\beta] = X_\beta$. Notice that this realisation leads to

$$\begin{aligned} X_\alpha X_\beta &= |v_i\rangle (\Gamma^i)_{\alpha\beta} \langle\varphi| \\ Y^\alpha Y^\beta &= |\varphi\rangle (\Gamma_i)^{\beta\alpha} \langle v^i| \end{aligned} \qquad (6.197)$$

and

$$X_\alpha X_\beta X_\gamma = 0 \qquad , \qquad Y^\alpha Y^\beta Y^\gamma = 0 \qquad (6.198)$$

We also have as interesting properties $X_\beta\varrho_{\mathbf{10}} = 0$ and $\varrho_{\mathbf{10}}Y^\beta = 0$, as well as

$$\begin{aligned} X_\beta\varrho_{\mathbf{1}} &= X_\beta \quad , & \varrho_{\mathbf{1}}Y^\beta &= Y^\beta \\ X_\beta\varrho_{\mathbf{16}} &= X_\beta \quad , & \varrho_{\mathbf{16}}Y^\beta &= Y^\beta \end{aligned} \qquad (6.199)$$

From these relations and the linear combinations $X = b^\beta X_\beta$ and $Y = c_\beta Y^\beta$ given by (6.190), we learn that $X^3 = Y^3 = 0$ while

$$X^2 = 2V^i |v_i\rangle \langle 0| \qquad , \qquad Y^2 = 2W_i |0\rangle \langle v^i| \qquad (6.200)$$

where we have set

$$V^i = \frac{1}{2}b^\alpha (\Gamma^i)_{\alpha\beta} b^\beta \qquad , \qquad W_i = \frac{1}{2}c_\alpha (\Gamma_i)^{\alpha\beta} c_\beta \qquad (6.201)$$

In terms of the short notations, we have $X_\alpha X_\beta \sim |\mathbf{10}\rangle \langle\mathbf{1}|$ and $Y^\alpha Y^\beta \sim |\mathbf{1}\rangle \langle\mathbf{10}|$ as well as $X^2 = 2\mathbf{V} |\mathbf{10}\rangle \langle\mathbf{1}|$ and $Y^2 = 2\mathbf{W} |\mathbf{1}\rangle \langle\mathbf{10}|$ where $\mathbf{V}$ and $\mathbf{W}$ are the vectors appearing in (6.200).

### 6.2.2 Constructing the operator $\mathcal{L}^{\mu}_{e_6}$

For the final step, we use the nilpotency feature of X and Y yielding the finite expansions $e^X = I + X + \frac{1}{2}X^2$ and $e^Y = I + Y + \frac{1}{2}Y^2$ as well as $z^{\mu}e^Y = z^{\mu} + z^{\mu}Y + \frac{1}{2}z^{\mu}Y^2$. Moreover, by replacing with

$$z^{\mu} = z^{-\frac{4}{3}}\varrho_{\mathbf{1}} + z^{\frac{2}{3}}\varrho_{\mathbf{10}} + z^{-\frac{1}{3}}\varrho_{\mathbf{16}} \tag{6.202}$$

and $\varrho_{\mathbf{10}}Y = 0$, we obtain

$$
\begin{aligned}
z^{\mu}e^Y &= z^{-4/3}\varrho_{\mathbf{1}} + z^{-1/3}\varrho_{\mathbf{16}} + z^{2/3}\varrho_{\mathbf{10}} \\
&\quad + z^{-4/3}\varrho_{\mathbf{1}}Y + z^{-1/3}\varrho_{\mathbf{16}}Y + \frac{1}{2}z^{-4/3}\varrho_{\mathbf{1}}Y^2
\end{aligned} \tag{6.203}
$$

Substituting this into $e^X z^{\mu} e^Y$ and using the property $X\varrho_{\mathbf{10}} = 0$, we finally find the expression of the L-operator we are looking for :

$$
\begin{aligned}
\mathcal{L}^{\mu}_{e_6} &= z^{-\frac{4}{3}}\varrho_{\mathbf{1}} + z^{-1/3}\varrho_{\mathbf{16}} + z^{2/3}\varrho_{\mathbf{10}} + z^{-4/3}\varrho_{\mathbf{1}}Y + z^{-1/3}\varrho_{\mathbf{16}}Y + \\
&\quad z^{-\frac{4}{3}}X\varrho_{\mathbf{1}} + z^{-1/3}X\varrho_{\mathbf{16}} + z^{-\frac{4}{3}}X\varrho_{\mathbf{1}}Y + \\
&\quad \frac{1}{2}z^{-\frac{4}{3}}\varrho_{\mathbf{1}}Y^2 + z^{-1/3}X\varrho_{\mathbf{16}}Y + \frac{1}{2}z^{-\frac{4}{3}}X\varrho_{\mathbf{1}}Y^2 + \\
&\quad \frac{1}{2}z^{-\frac{4}{3}}X^2\varrho_{\mathbf{1}} + \frac{1}{2}z^{-\frac{4}{3}}X^2\varrho_{\mathbf{1}}Y + \frac{1}{4}z^{-\frac{4}{3}}X^2\varrho_{\mathbf{1}}Y^2
\end{aligned} \tag{6.204}
$$

Notice that each one of the $z^{\mu}$, $e^X$ and $e^Y$ has 3 monomials leading in general to 81 monomialss for the $\mathcal{L}^{\mu}_{e_6}$. However, The above expression was simplified thanks to useful properties such as $X\varrho_{\mathbf{10}} = 0$ and $\varrho_{\mathbf{10}}Y = 0$ and the other ones mentioned above. It can be further expressed in terms of Darboux ccordinates by substituting the following relations

$$
\begin{aligned}
X\varrho_{\mathbf{1}} &= b^{\beta} & , & & X^2\varrho_{\mathbf{1}} &= b^{\beta}\Gamma^i_{\beta\gamma}b^{\gamma} \\
\varrho_{\mathbf{1}}Y &= c_{\alpha} & , & & X\varrho_{\mathbf{1}}Y^2 &= b^{\alpha}c_{\beta}\Gamma^{\beta\gamma}_i c \\
X\varrho_{\mathbf{1}}Y &= b^{\beta}c_{\alpha} & , & & X^2\varrho_{\mathbf{1}}Y^2 &= b^{\beta}\Gamma^i_{\beta\gamma}b^{\gamma}c_{\beta}\Gamma^{\beta\gamma}_i c_{\gamma}
\end{aligned} \tag{6.205}
$$

and

$$
\begin{aligned}
X\varrho_{\mathbf{16}} &= b^{\gamma}\Gamma^i_{\gamma\beta} \\
\varrho_{\mathbf{16}}Y &= \Gamma^{\gamma\beta}_i c_{\gamma} \\
X\varrho_{\mathbf{16}}Y &= b^{\gamma}\Gamma^i_{\gamma\beta}\Gamma^{\gamma\beta}_i c_{\gamma} \\
\varrho_{\mathbf{1}}Y^2 &= c_{\beta}\Gamma^{\beta\gamma}_i c_{\gamma}
\end{aligned} \tag{6.206}
$$

and

$$X\varrho_{\mathbf{16}}Y^2 = 0 \quad , \quad X^2\varrho_{\mathbf{16}}Y^2 = 0 \tag{6.207}$$

## 6.3 Topological gauge quiver $\mathbf{Q}^{\mu}_{e_6}$

In this subsection, we construct the topological gauge quiver $\mathbf{Q}^{\mu}_{e_6}$ associated with the operator $\mathcal{L}^{\mu}_{e_6}$ (6.204). First, we give the matrix form of the L-operator in terms of the phase variables $b^{\beta}$ and $c_{\beta}$ to underline their field theory interpretation in terms of topological bi-matter. Then, we derive the quiver representation $\mathbf{Q}^{\mu}_{e_6}$ using the projectors $\varrho_{\mathbf{1}}$, $\varrho_{\mathbf{10}}$ and $\varrho_{\mathbf{16}}$ on the sub-representations of $so_{10}$ within the 27 of E$_6$.

By ordering the above mentioned projectors like $(\varrho_{\mathbf{10}}, \varrho_{\mathbf{16}}, \varrho_{\mathbf{1}})$ and thinking of them as representing the sub-blocks of the matrix; the operator $\mathcal{L}_{e_6}^{\mu}$ is put as follows

$$\mathcal{L}_{e_6}^{\mu} = \begin{pmatrix} z^{\frac{2}{3}}\varrho_{\mathbf{10}} + z^{-\frac{1}{3}}X\varrho_{\mathbf{16}}Y + \frac{1}{4}z^{-\frac{4}{3}}X^2\varrho_{\mathbf{1}}Y^2 & z^{-\frac{1}{3}}X\varrho_{\mathbf{16}} + \frac{1}{2}z^{-\frac{4}{3}}X^2\varrho_{\mathbf{1}}Y & \frac{1}{2}z^{-\frac{4}{3}}X^2\varrho_{\mathbf{1}} \\ z^{-\frac{1}{3}}\varrho_{\mathbf{16}}Y + \frac{1}{2}z^{-\frac{4}{3}}X\varrho_{\mathbf{1}}Y^2 & z^{-\frac{1}{3}}\varrho_{\mathbf{16}} + z^{-\frac{4}{3}}X\varrho_{\mathbf{1}}Y & z^{-\frac{4}{3}}X\varrho_{\mathbf{1}} \\ \frac{1}{2}z^{-\frac{4}{3}}\varrho_{\mathbf{1}}Y^2 & z^{-\frac{4}{3}}\varrho_{\mathbf{1}}Y & z^{-\frac{4}{3}}\varrho_{\mathbf{1}} \end{pmatrix}$$
(6.208)

Substituting eqs(6.194) and (6.200) into the expansions $X = b^{\beta}X_{\beta}$ and $Y = c_{\beta}Y^{\beta}$ as well as into their squares $X^2$ and $Y^2$, we obtain

$$\mathcal{L}_{e_6}^{\mu} = \begin{pmatrix} z^{\frac{2}{3}} + z^{-\frac{1}{3}}b^{\beta}c_{\beta} + \frac{1}{4}z^{-\frac{4}{3}}V^iW_i & z^{-\frac{1}{3}}b^{\beta}\Gamma^i_{\beta\gamma} + \frac{1}{2}z^{-\frac{4}{3}}V^ic_{\beta} & \frac{1}{2}z^{-\frac{4}{3}}V^i \\ z^{-\frac{1}{3}}c_{\beta}\Gamma^{\beta\gamma}_i + \frac{1}{2}z^{-\frac{4}{3}}b^{\beta}W_i & z^{-\frac{1}{3}} + z^{-\frac{4}{3}}b^{\beta}c_{\beta} & z^{-\frac{4}{3}}b^{\beta} \\ \frac{1}{2}z^{-\frac{4}{3}}W_i & z^{-\frac{4}{3}}c_{\beta} & z^{-\frac{4}{3}} \end{pmatrix}$$
(6.209)

where $V^i = \frac{1}{2}\mathbf{b}\Gamma^i\mathbf{b}$ and $W_i = \frac{1}{2}\mathbf{c}\Gamma_i\mathbf{c}$. This is the most convenient expression of the coupling between 'tH$_{\gamma_0}^{\mu_{e_6}}$ and $W_{\xi_z}^{27}$ in the $E_6$ CS theory allowing to derive the associated topological quiver $Q_{e_6}^{\mu}$. In fact, by writing the L-operator like $\left\langle \varrho_{\mathbf{R}_i} | \mathcal{L}^{\mu} | \varrho_{\mathbf{R}_j} \right\rangle$, which is

$$\mathcal{L}_{ij}^{\mu} = \varrho_{\mathbf{R}_i}\mathcal{L}^{\mu}\varrho_{\mathbf{R}_j} \tag{6.210}$$

We have in terms of the projectors :

$$\mathcal{L}_{e_6}^{\mu} = \begin{pmatrix} \varrho_{\mathbf{10}}\mathcal{L}^{\mu}\varrho_{\mathbf{10}} & \varrho_{\mathbf{10}}\mathcal{L}^{\mu}\varrho_{\mathbf{16}} & \varrho_{\mathbf{10}}\mathcal{L}^{\mu}\varrho_{\mathbf{1}} \\ \varrho_{\mathbf{16}}\mathcal{L}^{\mu}\varrho_{\mathbf{10}} & \varrho_{\mathbf{16}}\mathcal{L}^{\mu}\varrho_{\mathbf{16}} & \varrho_{\mathbf{16}}\mathcal{L}^{\mu}\varrho_{\mathbf{1}} \\ \varrho_{\mathbf{1}}\mathcal{L}^{\mu}\varrho_{\mathbf{10}} & \varrho_{\mathbf{1}}\mathcal{L}^{\mu}\varrho_{\mathbf{16}} & \varrho_{\mathbf{1}}\mathcal{L}^{\mu}\varrho_{\mathbf{1}} \end{pmatrix}$$
(6.211)

This directly indicates that the topological gauge quiver $Q_{e_6}^{\mu}$ has three nodes $\mathcal{N}_1, \mathcal{N}_2, \mathcal{N}_3$ and six links, three $L_{ij}$ and three $L_{ji}$ with $i > j = 1, 2, 3$, as depicted by the Figure **24**.
The $\mathcal{N}_i$ nodes are associated with the diagonal enties of (6.211), namely

$$\mathcal{N}_1 \equiv \varrho_{\mathbf{10}}\mathcal{L}^{\mu}\varrho_{\mathbf{10}} \quad , \quad \mathcal{N}_2 \equiv \varrho_{\mathbf{16}}\mathcal{L}^{\mu}\varrho_{\mathbf{16}} \quad , \quad \mathcal{N}_3 \equiv \varrho_{\mathbf{1}}\mathcal{L}^{\mu}\varrho_{\mathbf{1}} \tag{6.212}$$

We will refer to them in terms of the $SO_2 \times SO_{10}$ representations as follows

$$\begin{array}{rcl} \mathcal{N}_1 & : & \mathbf{10}_{+2/3} \\ \mathcal{N}_2 & : & \mathbf{16}_{-1/3} \\ \mathcal{N}_3 & : & \mathbf{1}_{-4/3} \end{array}$$
(6.213)

The $L_{ij}$ links of the quiver $Q_{e_6}^{\mu}$ are given by the off diagonal terms $\varrho_{\mathbf{R}_i}\mathcal{L}^{\mu}\varrho_{\mathbf{R}_j}$ with $i \neq j$. These links transform in the fundamental representations of $SO_2 \times SO_{10}$ knowing that **10** and **16** and their duals are fundamental representations of $SO_{10}$. The explicit expressions of these links are given in the following table

| link | Repres | bi-matter | link | Repres | bi-matter |
|---|---|---|---|---|---|
| $L_{1\to2}$ | $\mathbf{16}_{-\frac{1}{3}} \times \overline{\mathbf{10}}_{-\frac{2}{3}}$ | $\mathbf{b}, \mathbf{b}^2\mathbf{c}$ | $L_{2\to1}$ | $\mathbf{10}_{\frac{2}{3}} \times \overline{\mathbf{16}}_{\frac{1}{3}}$ | $\mathbf{c}, \mathbf{bc}^2$ |
| $L_{2\to3}$ | $\mathbf{1}_{-\frac{4}{3}} \times \overline{\mathbf{16}}_{+\frac{1}{3}}$ | $\mathbf{b}$ | $L_{2\to1}$ | $\mathbf{16}_{-\frac{1}{3}} \times \overline{\mathbf{1}}_{\frac{4}{3}}$ | $\mathbf{c}$ |
| $L_{1\to3}$ | $\mathbf{1}_{-\frac{4}{3}} \times \overline{\mathbf{10}}_{-\frac{2}{3}}$ | $\mathbf{b}^2$ | $L_{3\to1}$ | $\mathbf{10}_{\frac{2}{3}} \times \overline{\mathbf{1}}_{+\frac{4}{3}}$ | $\mathbf{c}^2$ |

(6.214)

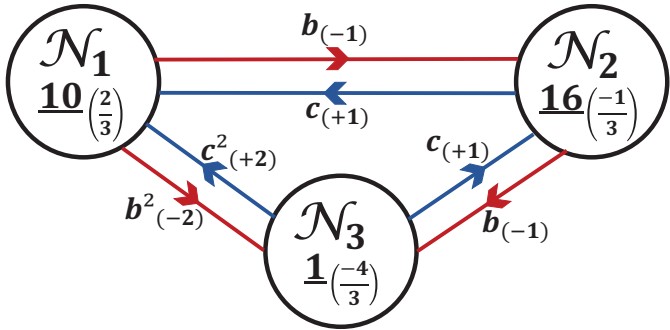

Figure 24: $\mathcal{L}_{E_6}$ as a topological quiver with 3 nodes and 6 links. The nodes are given by the self-dual $R_i \otimes \bar{R}_i$ and the links by bi-matter $R_i \otimes \bar{R}_j$. In addition to $SO_{10}$ representations, the Darboux coordinates $b^\alpha$, $c_\alpha$ carry $SO_2$ charges given by $q = \pm 1$. The fundamental vector-like matter $V^i$ and $W_i$ carry $-2$ and $+2$.

# 7 Minuscule line defects in $E_7$ CS theory

In this section, we complete the study undertaken in this paper regarding the minuscule L-operators of ADE type by investigating the case of 4D Chern Simons theory with exceptional $E_7$ gage symmetry. Just as before, we treat this theory by studying the properties of interacting minuscule 't Hooft and Wilson lines, and construct the Lax operators $\mathcal{L}^\mu_{e_7}$ and the associated topological gauge quivers $Q^\mu_{e_7}$.

## 7.1 Levi subalgebra of $E_7$ and weights of the $56_{e_7}$

First, we begin by recalling the useful aspects of the $e_7$ Lie algebra that will play an important role in our construction. In particular, the root system $\Phi_{e_7}$ containing 126 roots is generated by seven simple roots $\alpha_i$ realised as follows

$$
\begin{aligned}
\mathrm{E}_7 \quad : \quad \alpha_1 &= \tfrac{1}{2}\left(\epsilon_1 - \epsilon_2 - \epsilon_3 - \epsilon_4 - \epsilon_5 - \epsilon_6 - \epsilon_7 + \epsilon_8\right) \\
\alpha_i &= \epsilon_i - \epsilon_{i-1} \quad , \qquad i = 2,3,4,6 \\
\alpha_7 &= \epsilon_1 + \epsilon_2
\end{aligned}
\tag{7.215}
$$

The Dynkin diagram underlying the gauge symmetry of the 4D CS theory with $E_7$ symmetry is given by the Figure 25 where the seven simple roots $\alpha_i$ are exhibited.
The associated Cartan matrix $K_{e_7}$ reads as

$$
K_{e_7} = \begin{pmatrix}
2 & -1 & 0 & 0 & 0 & 0 & 0 \\
-1 & 2 & -1 & 0 & 0 & 0 & 0 \\
0 & -1 & 2 & -1 & 0 & 0 & -1 \\
0 & 0 & -1 & 2 & -1 & 0 & 0 \\
0 & 0 & 0 & -1 & 2 & 0 & 0 \\
0 & 0 & 0 & 0 & 0 & 2 & 0 \\
0 & 0 & -1 & 0 & 0 & 0 & 2
\end{pmatrix}
\tag{7.216}
$$

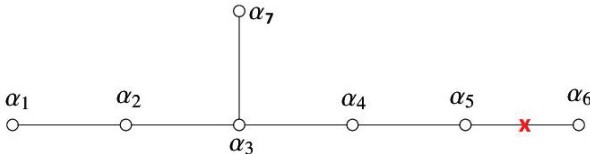

Figure 25: Dynkin Diagram of $E_7$ having seven nodes labeled by the simple roots $\alpha_i$. The cross ($\times$) indicates the root cut by the Levi decomposition where the Levi subgroup is $SO_2 \times E_6$.

It describes the intersection matrix $\alpha_i.\alpha_j$ while its inverse gives the fundamental coweights of $E_7$. One of these coweights is particularly interesting for our present study; the $\mu$ dual to $\alpha_6$ is the only minuscule coweight of $e_7$.

### 7.1.1 Minuscule coweight of $E_7$

From the Cartan matrix $K_{e_7}$, we can learn useful informations regarding the Lie algebra $e_7$ and its representations, in particular the expressions of fundamental weights $\omega_i$ in terms of simple roots :

| fund- $\omega_i$ | in terms of roots |
|---|---|
| $\omega_1$ | $2\alpha_1 + 3\alpha_2 + 4\alpha_3 + 3\alpha_4 + 2\alpha_5 + \alpha_6 + 2\alpha_7$ |
| $\omega_2$ | $3\alpha_1 + 6\alpha_2 + 8\alpha_3 + 6\alpha_4 + 4\alpha_5 + 2\alpha_6 + 4\alpha_7$ |
| $\omega_3$ | $4\alpha_1 + 8\alpha_2 + 12\alpha_3 + 9\alpha_4 + 6\alpha_5 + 3\alpha_6 + 6\alpha_7$ |
| $\omega_4$ | $3\alpha_1 + 6\alpha_2 + 9\alpha_3 + \frac{15}{2}\alpha_4 + 5\alpha_5 + \frac{5}{2}\alpha_6 + \frac{9}{2}\alpha_7$ |
| $\omega_5$ | $2\alpha_1 + 4\alpha_2 + 6\alpha_3 + 5\alpha_4 + 4\alpha_5 + 2\alpha_6 + 3\alpha_7$ |
| $\omega_6$ | $\alpha_1 + 2\alpha_2 + 3\alpha_3 + \frac{5}{2}\alpha_4 + 2\alpha_5 + \frac{3}{2}\alpha_6 + \frac{3}{2}\alpha_7$ |
| $\omega_7$ | $2\alpha_1 + 4\alpha_2 + 6\alpha_3 + \frac{9}{2}\alpha_4 + 3\alpha_5 + \frac{3}{2}\alpha_6 + \frac{7}{2}\alpha_7$ |

$$(7.217)$$

The exceptional Lie algebra $e_7$ has one minuscule coweight $\mu$ given by $\omega_6$, thus the corresponding Levi decomposition $\boldsymbol{n}_- \oplus \boldsymbol{l}_\mu \oplus \boldsymbol{n}_+$ for this algebra is given by

$$\boldsymbol{l}_\mu = so_2 \oplus e_6 \qquad , \qquad \boldsymbol{n}_\pm = 27_\pm \qquad (7.218)$$

The dimensions of $\boldsymbol{n}_\pm$ can be calculated by dispatching the algebraic dimensions of $e_7$ with respect to $so_2 \oplus e_6$, in fact we have $133 = 1 + 78 + 27 + 27$. This Levi decomposition with respect to the minuscule coweight $\mu$ requires the following adjoint actions

$$[\mu, \boldsymbol{n}_\pm] = \pm\boldsymbol{n}_\pm \qquad , \qquad [\boldsymbol{n}_+, \boldsymbol{n}_-] = 0 \qquad (7.219)$$

These constraints show that the 27 generators $X_\beta$ of the nilpotent algebra $\boldsymbol{n}_+$ and the 27 generators $Y^\beta$ of the algebra $\boldsymbol{n}_-$ have opposite $so_2$ charges $\pm 1$, which is important to consider when realising the action of $X_\beta$ and $Y^\beta$ on the electrically charged quantum states $|A\rangle$ that we take in the fundamental representation of $E_7$.

### 7.1.2 Representation $\underline{56}$ of the $e_7$ Lie algebra

The fundamental representation of the $\boldsymbol{e}_7$ algebra has 56 dimensions, it is self dual and pseudo-real [69]. Its weight diagram is given by the Figure **26** where the weight $\xi_0$ of the top state $|\xi_0\rangle$ corresponds to the minuscule coweight $\omega_6$ while the weight $\xi_{55}$ of the bottom state $|\xi_{55}\rangle$ is precisely $-\omega_6$, meaning that we have $\xi_0 + \xi_{55} = 0$.

Under the Levi decomposition associated to the minuscule $\mu$, the fundamental representation **56** decomposes as a reducible sum of $so_2 \oplus e_6$ representations as follows

$$
\begin{aligned}
\mathbf{56}_0 &= \mathbf{28}_+ \oplus \mathbf{28}_- \\
\mathbf{28}_+ \oplus \mathbf{28}_- &= \mathbf{1}_{3/2} \oplus \mathbf{27}_{+1/2} \oplus \mathbf{27}_{-1/2} \oplus \mathbf{1}_{-3/2}
\end{aligned}
\tag{7.220}
$$

where we have four $e_6$ representations, two singlets $\mathbf{1}_{\pm 3/2}$ and two fundamentals $\mathbf{27}_{\pm 1/2}$.

In the diagram of Figure **27**, the 28 weights of $\mathbf{28}_+$ are labeled by the subset $W_+ = \{|\xi_i\rangle\}_{0 \le i \le 27}$ and the 28 weights of the $\mathbf{28}_-$ by $W_- = \{|\xi_i\rangle\}_{28 \le i \le 55}$. Weights $\xi_i$ in the set $W_+ \cup W_-$ obey some special features that characterize this exceptional algebra and that will be helpful for the construction of the operator $\mathcal{L}_{e_7}^\mu$, they are listed below

$$
\begin{aligned}
\xi_{27} &= \xi_0 - \beta_{\max} &,\quad \xi_{27} + \xi_{28} &= \xi_0 + \xi_{55} \\
\xi_{28} &= \xi_{55} + \beta_{\max} &,\quad \xi_i + \xi_{55-i} &= \xi_0 + \xi_{55} \\
\xi_i &= \xi_0 - \gamma_i &,\quad \xi_{55-i} &= \xi_{55} + \gamma_i
\end{aligned}
\tag{7.221}
$$

for a generic root $\gamma_i$ in the nilpotent $\mathbf{27}_+$ and where $\beta_{\max}$ is given by

$$
\beta_{\max} = 2\alpha_1 + 3\alpha_2 + 4\alpha_3 + 3\alpha_4 + 2\alpha_5 + \alpha_6 + 2\alpha_7
\tag{7.222}
$$

We also have

$$
\xi_0 - \xi_{55} = 2\omega_6 \quad,\quad \xi_i - \xi_{55-i} = 2\omega_6 - 2\gamma_i
\tag{7.223}
$$

The list of the ten weights $\xi_A, A = 1, ..., 10$ represented by blue dots in the Figure **26** is given in the following table in terms of the seven $\omega_i$'s,

$$
\begin{aligned}
\xi_1 &= \omega_5 - \omega_6 &,\quad \xi_6 &= \omega_2 - \omega_7 \\
\xi_2 &= \omega_4 - \omega_5 &,\quad \xi_7 &= \omega_1 + \omega_3 - \omega_7 - \omega_2 \\
\xi_3 &= \omega_3 - \omega_4 &,\quad \xi_8 &= \omega_1 + \omega_4 - \omega_3 \\
\xi_4 &= \omega_7 + \omega_2 - \omega_3 &,\quad \xi_9 &= \omega_1 + \omega_5 - \omega_4 \\
\xi_5 &= \omega_1 + \omega_7 - \omega_2 &,\quad \xi_{10} &= \omega_1 + \omega_6 - \omega_5
\end{aligned}
\tag{7.224}
$$

while the next sixteen states (8+8) represented in the Figure **26** by yellow and magenta colored dots (from $\xi_{19}$ to $\xi_{26}$) are listed here

$$
\begin{aligned}
\xi_{11} &= \omega_7 - \omega_1 &,\quad \xi_{15} &= \omega_5 + \omega_2 - \omega_4 - \omega_1 \\
\xi_{12} &= -\omega_7 - \omega_1 &,\quad \xi_{16} &= \omega_5 + \omega_3 - \omega_4 - \omega_2 \\
\xi_{13} &= \omega_2 + \omega_4 - \omega_3 - \omega_1 &,\quad \xi_{17} &= \omega_7 + \omega_5 - \omega_3 \\
\xi_{14} &= \omega_4 - \omega_2 &,\quad \xi_{18} &= \omega_5 - \omega_7
\end{aligned}
\tag{7.225}
$$

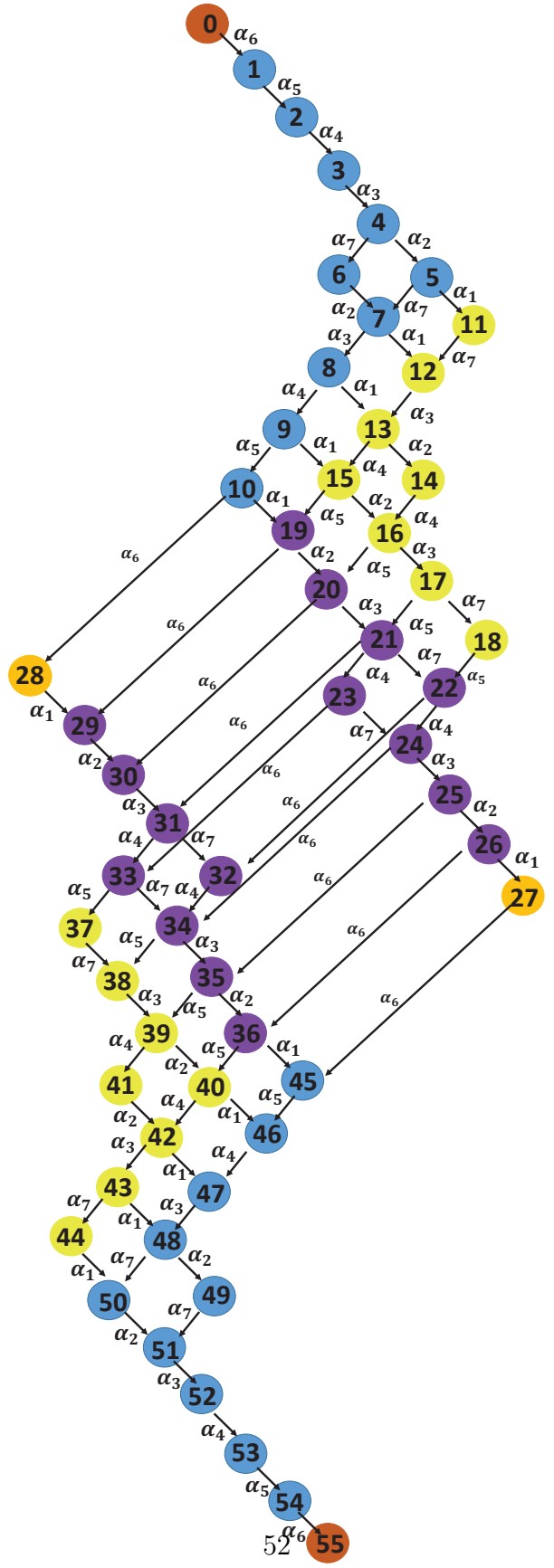

Figure 26: The decomposition of the **56** representation of $e_7$ in terms of representations of $e_6$. We have $\mathbf{56} = \mathbf{28}_+ \oplus \mathbf{28}_-$ where $\mathbf{28}_\pm$ are reducible like $\mathbf{1}_{\pm 3/2} \oplus \mathbf{27}_{\pm 1/2}$.

and

$$\begin{array}{rlrl}
\xi_{19} &= \omega_2 + \omega_6 - \omega_1 - \omega_5 & , & \quad \xi_{23} &= \omega_7 + \omega_6 - \omega_4 \\
\xi_{20} &= \omega_3 + \omega_6 - \omega_5 - \omega_2 & , & \quad \xi_{24} &= \omega_3 + \omega_6 - \omega_4 - \omega_7 \\
\xi_{21} &= \omega_7 + \omega_4 + \omega_6 - \omega_5 - \omega_3 & , & \quad \xi_{25} &= \omega_2 + \omega_6 - \omega_3 \\
\xi_{22} &= \omega_4 + \omega_6 - \omega_5 - \omega_7 & , & \quad \xi_{26} &= \omega_1 + \omega_6 - \omega_2
\end{array} \tag{7.226}$$

The last 27-th weight is equal to $\xi_{27} = \omega_6 - \omega_1$.

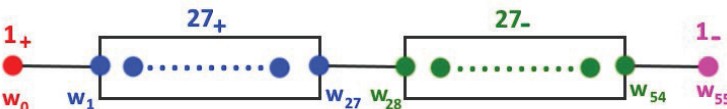

Figure 27: The decomposition of the **56** representation of E$_7$ in terms of representations of E$_6$. We have $\mathbf{56} = \mathbf{28}_+ \oplus \mathbf{28}_-$ with $\mathbf{28}_\pm$ reducible like $\mathbf{1}_{\pm 3/2} \oplus \mathbf{27}_{\pm 1/2}$.

## 7.2   Constructing the $\mathcal{L}^\mu_{e_7}$

Now, we consider the minuscule 't Hooft line embedded in the E$_7$ CS theory crossing a Wilson line $W^R_{e_7}$ with electric weight given by the representation **56**. To construct the L-operator $\mathcal{L}_{e_7}$ describing these topological lines' coupling, we follow the same approach adopted before for the study A-, D- and E$_6$ type theories.

### 7.2.1   Realising the generators of the $\mathbf{n}_{\pm 27}$ subalgebras

We begin by recalling that in the L-operator formula for the E$_7$ symmetry, namely $\mathcal{L}_{e_7} = e^X z^\mu e^Y$, the $\mu$ is the minuscule coweight given in (25) and X and Y are nilpotent matrices expanding as

$$X = \sum_{\beta=1}^{27} b^\beta X_\beta \quad , \quad Y = \sum_{\beta=1}^{27} c_\beta Y^\beta \tag{7.227}$$

Here, the twenty seven $b^\beta$ and twenty seven $c_\beta$ are the Darboux coordinates of the phase space of the E$_7$-type 't Hooft line. The realisation of the nilpotent generators $X_\beta$ and $Y^\beta$ can be first written using simple representation language like

$$\begin{array}{rl}
X_\beta &\equiv \ |\mathbf{1}_+\rangle \langle \mathbf{27}_+| + |\mathbf{27}_+\rangle \langle \mathbf{27}_-| + |\mathbf{27}_-\rangle \langle \mathbf{1}_-| \\
Y^\beta &\equiv \ |\mathbf{1}_-\rangle \langle \mathbf{27}_-| + |\mathbf{27}_-\rangle \langle \mathbf{27}_+| + |\mathbf{27}_+\rangle \langle \mathbf{1}_+|
\end{array} \tag{7.228}$$

where we dropped the charges from $\mathbf{1}_{\pm 3/2}$ and $\mathbf{27}_{\pm 1/2}$ for simplicity. The explicit form of these generators in terms of the weight states $|\xi_A\rangle$ and their duals $\langle \xi_A|$ is given by

$$\begin{array}{rl}
X_\beta &= \ \left|\xi_{0_+}\right\rangle \left\langle \xi_{\beta_+}\right| + \left|\xi_{\delta_+}\right\rangle \Gamma^{\delta_+ \gamma_-}_\beta \left\langle \xi_{\gamma_-}\right| + \left|\xi_{\beta_-}\right\rangle \left\langle \xi_{0_-}\right| \\
Y^\beta &= \ \left|\xi_{0_-}\right\rangle \left\langle \xi^{\beta_-}\right| + \left|\xi^{\gamma_-}\right\rangle \Gamma^\beta_{\gamma_- \delta_+} \left\langle \xi^{\delta_+}\right| + \left|\xi^{\beta_+}\right\rangle \left\langle \xi_{0_+}\right|
\end{array} \tag{7.229}$$

where $\Gamma^{\delta_+\gamma_-}_\beta$ and $\Gamma^\beta_{\gamma_-\delta_+}$ are couplings between states in the **27** representations of $E_6$; these tensors are allowed by the tensor product of $E_6$ representations [70]. The adjoint form of the minuscule coweight used is given by

$$\mu = \frac{3}{2}\varrho_{\mathbf{1}_+} + \frac{1}{2}\varrho_{\mathbf{27}_+} - \frac{1}{2}\varrho_{\mathbf{27}_-} - \frac{3}{2}\varrho_{\mathbf{1}_-} \tag{7.230}$$

where the four $\varrho_{\mathbf{R}_i}$'s are projectors on the $e_6$ representations $\mathbf{R}_i$ within the **56** of $e_7$, they read as follows

$$\varrho_{\mathbf{1}_q} = \left|\xi_{0_q}\right\rangle\left\langle\xi_{0_q}\right| \qquad , \qquad \varrho_{\mathbf{27}_q} = \left|\xi_{27_q}\right\rangle\left\langle\xi_{27_q}\right| \tag{7.231}$$

with $q = \pm$ and $\left\langle\xi_{0_q}|\xi_{0_q}\right\rangle = \left\langle\xi_{27_q}|\xi_{27_q}\right\rangle = 1$. They can also be written in formal notations as

$$\varrho_{\mathbf{1}_q} = \left|\mathbf{1}_q\right\rangle\left\langle\mathbf{1}_q\right| \qquad , \qquad \varrho_{\mathbf{27}_q} = \left|\mathbf{27}_q\right\rangle\left\langle\mathbf{27}_q\right| \tag{7.232}$$

Now, we need to compute the powers of the generators $X_\beta$ and $Y^\beta$ that will appear in the expansion of the L-operator. We find using the realisation (7.228-7.229) that the non vanishing monomials are

$$\begin{array}{llll}
X_\alpha X_\beta & \equiv & \left|\mathbf{1}_+\right\rangle\left\langle\mathbf{27}_-\right| + \left|\mathbf{27}_+\right\rangle\left\langle\mathbf{1}_-\right| & , \quad X_\alpha X_\beta X_\gamma \equiv \left|\mathbf{1}_+\right\rangle\left\langle\mathbf{1}_-\right| \\
Y^\alpha Y^\beta & \equiv & \left|\mathbf{1}_-\right\rangle\left\langle\mathbf{27}_+\right| + \left|\mathbf{27}_-\right\rangle\left\langle\mathbf{1}_+\right| & , \quad Y^\alpha Y^\beta Y^\gamma \equiv \left|\mathbf{1}_-\right\rangle\left\langle\mathbf{1}_+\right|
\end{array} \tag{7.233}$$

while the fourth order powers vanish identically. For the powers of the linear combinations $X = b^\beta X_\beta$ and $Y = c_\beta Y^\beta$, we find

$$\begin{array}{rcl}
X^2 & = & 2S^{\beta_-}\left|\xi_{0_+}\right\rangle\left\langle\xi_{\beta_-}\right| + 2S^{\beta_+}\left|\xi_{\beta_+}\right\rangle\left\langle\xi_{0_-}\right| \\
Y^2 & = & 2R_{\alpha_+}\left|\xi_{0_-}\right\rangle\left\langle\xi^{\alpha_+}\right| + 2R_{\alpha_-}\left|\xi^{\alpha_-}\right\rangle\left\langle\xi_{0_+}\right|
\end{array} \tag{7.234}$$

and

$$\begin{array}{rcl}
X^3 & = & 6\mathcal{E}\left|\xi_{0_+}\right\rangle\left\langle\xi_{0_-}\right| \\
Y^3 & = & 6\mathcal{F}\left|\xi_{0_-}\right\rangle\left\langle\xi_{0_+}\right|
\end{array} \tag{7.235}$$

and of course, $X^4 = Y^4 = 0$. The realisation (7.228-7.229) does also obey the commutation relations $[\mu, X_\beta] = X_\beta$ and $\left[\mu, Y^\beta\right] = -Y^\beta$ from which we deduce that

$$[\mu, X] = X \qquad , \qquad [\mu, Y] = -Y \tag{7.236}$$

as required by the Levi decomposition with respect to $\mu$.

### 7.2.2 The L-operator $\mathcal{L}^\mu_{e_7}$

Finally, to obtain the expression of $\mathcal{L}^\mu_{e_7}$ in terms of the 27+27 Darboux coordinates $b^\beta$ and $c_\beta$, we use the nilpotency properties mentioned above to write

$$\mathcal{L}^\mu_{e_7} = \left(I + X + \frac{1}{2}X^2 + \frac{1}{6}X^3\right)z^\mu\left(I + Y + \frac{1}{2}Y^2 + \frac{1}{6}Y^3\right) \tag{7.237}$$

and substitute with

$$z^\mu = z^{\frac{3}{2}}\varrho_{\mathbf{1}_+} + z^{\frac{1}{2}}\varrho_{\mathbf{27}_+} + z^{-\frac{1}{2}}\varrho_{\mathbf{27}_-} + z^{-\frac{3}{2}}\varrho_{\mathbf{1}_-} \tag{7.238}$$

We moreover need to take into account the special properties of the X and Y matrices, like for example $X\varrho_{1_+} = 0$ and $\varrho_{1_+}Y = 0$, to reduce the monomials of this L-operator down to 30 as given below

$$
\begin{aligned}
\mathcal{L}^{\mu}_{e_7} \;=\; & z^{\frac{3}{2}}\varrho_{1_+} + z^{\frac{1}{2}}\varrho_{27_+} + z^{-\frac{1}{2}}\varrho_{27_-} + z^{-\frac{3}{2}}\varrho_{1_-} \\
& z^{\frac{1}{2}}X\varrho_{27_+} + z^{-\frac{1}{2}}X\varrho_{27_-} + z^{-\frac{3}{2}}X\varrho_{1_-} + \\
& z^{\frac{1}{2}}\varrho_{27_+}Y + z^{-\frac{1}{2}}\varrho_{27_-}Y + z^{-\frac{3}{2}}\varrho_{1_-}Y + \\
& +\tfrac{1}{2}X^2 z^{-\frac{1}{2}}\varrho_{27_-} + \tfrac{1}{2}z^{-\frac{3}{2}}X^2\varrho_{1_-} + \tfrac{1}{6}z^{-\frac{3}{2}}X^3\varrho_{1_-} + \\
& \tfrac{1}{2}z^{-\frac{1}{2}}\varrho_{27_-}Y^2 + \tfrac{1}{2}z^{-\frac{3}{2}}\varrho_{1_-}Y^2 + \tfrac{1}{6}z^{-\frac{3}{2}}\varrho_{1_-}Y^3 + \\
& z^{\frac{1}{2}}X\varrho_{27_+}Y + z^{-\frac{1}{2}}X\varrho_{27_-}Y + z^{-\frac{3}{2}}X\varrho_{1_-}Y \\
& \tfrac{1}{2}z^{-\frac{1}{2}}X\varrho_{27_-}Y^2 + \tfrac{1}{2}z^{-\frac{3}{2}}X\varrho_{1_-}Y^2 + \\
& \tfrac{1}{2}z^{-\frac{1}{2}}X^2\varrho_{27_-}Y + \tfrac{1}{2}z^{-\frac{3}{2}}X^2\varrho_{1_-}Y + +\tfrac{1}{6}z^{-\frac{3}{2}}X\varrho_{1_-}Y^3 + \\
& \tfrac{1}{6}z^{-\frac{3}{2}}X^3\varrho_{1_-}Y + \tfrac{1}{12}z^{-\frac{3}{2}}X^2\varrho_{1_-}Y^3 + \tfrac{1}{12}z^{-\frac{3}{2}}X^3\varrho_{1_-}Y^2 + \\
& \tfrac{1}{4}z^{-\frac{1}{2}}X^2\varrho_{27_-}Y^2 + \tfrac{1}{4}z^{-\frac{3}{2}}X^2\varrho_{1_-}Y^2 + \tfrac{1}{36}z^{-\frac{3}{2}}X^3\varrho_{1_-}Y^3
\end{aligned}
\tag{7.239}
$$

The explicit form of $\mathcal{L}_{E_7}$ given in [59] is obtained by replacing $X = b^{\beta}X_{\beta}$, $Y = c_{\beta}Y^{\beta}$ and $\mu$ by their explicit realisations (7.229,7.230,7.234). This is clearly a cumbersome expression, that's why we use the quiver gauge description to exhibit the interesting information encoded in $\mathcal{L}^{\mu}_{e_7}$ and help visualize the key role of the Darboux coordinates.

## 7.3   Topological gauge quiver $\mathrm{Q}^{\mu}_{e_7}$

The shape of the gauge quiver $\mathrm{Q}^{\mu}_{e_7}$ associated to the $\mathcal{L}^{\mu}_{e_7}$ operator can be directly deduced from properties of the $e_7$ algebra by comparison with the previously built quivers $\mathrm{Q}^{\mu}_{sl_N}$, $\mathrm{Q}^{\mu}_{so_{2N}}$ and $\mathrm{Q}^{\mu}_{e_6}$. Firstly, we can say that the $\mathrm{Q}^{\mu}_{e_7}$ has four nodes $\mathcal{N}_i$ in 1:1 correspondence with the four projectors $\varrho_{1_\pm}$ and $\varrho_{27_\pm}$, and 12 links $\mathrm{L}_{ij}$ connecting the pairs $(\mathcal{N}_i, \mathcal{N}_j)$. Therefore, we can begin by visualizing this $\mathrm{Q}^{\mu}_{e_7}$ as given in the Figure **28** , and then move on to explicitly derive it and extract its features.

We represent the $\mathcal{L}^{\mu}_{e_7}$ in the projector basis using the $\varrho_{R_q}$ ordered like $\left(\varrho_{1_+}, \varrho_{27_+}, \varrho_{27_-}, \varrho_{1_-}\right)$

$$
\mathcal{L}^{\mu}_{e_7} = \begin{pmatrix}
\varrho_{1_+}\mathcal{L}\varrho_{1_+} & \varrho_{1_+}\mathcal{L}\varrho_{27_+} & \varrho_{1_+}\mathcal{L}\varrho_{27_-} & \varrho_{1_+}\mathcal{L}\varrho_{1_-} \\
\varrho_{27_+}\mathcal{L}\varrho_{1_+} & \varrho_{27_+}\mathcal{L}\varrho_{27_+} & \varrho_{27_+}\mathcal{L}\varrho_{27_-} & \varrho_{27_+}\mathcal{L}\varrho_{1_-} \\
\varrho_{27_-}\mathcal{L}\varrho_{1_+} & \varrho_{27_-}\mathcal{L}\varrho_{27_+} & \varrho_{27_-}\mathcal{L}\varrho_{27_-} & \varrho_{27_-}\mathcal{L}\varrho_{1_-} \\
\varrho_{1_-}\mathcal{L}\varrho_{1_+} & \varrho_{1_-}\mathcal{L}\varrho_{27_+} & \varrho_{1_-}\mathcal{L}\varrho_{27_-} & \varrho_{1_-}\mathcal{L}\varrho_{1_-}
\end{pmatrix}
\tag{7.240}
$$

The diagonal terms $\varrho_{R_i}\mathcal{L}\varrho_{R_i}$ are depicted by the four nodes $\mathcal{N}_{R_i}$ of $\mathrm{Q}^{\mu}_{e_7}$, while the off diagonal terms $\varrho_{R_i}\mathcal{L}\varrho_{R_j}$ with $i{\neq}j$ are associated to the twelve links $\mathrm{L}_{ij}$.

$$
\mathcal{N}_{R_i} \equiv \varrho_{R_i}\mathcal{L}\varrho_{R_i} \qquad , \qquad L_{ij} = \varrho_{R_i}\mathcal{L}\varrho_{R_j}
\tag{7.241}
$$

As the explicit calculation of these quantities is cumbersome, we decompose the matrix $\mathcal{L}^{\mu}_{e_7}$ (7.240) into four blocks $A^{\mu}_{e_7}$, $B^{\mu}_{e_7}$, $C^{\mu}_{e_7}$ and $D^{\mu}_{e_7}$ as follows

$$
\mathcal{L}^{\mu}_{e_7} = \begin{pmatrix} A^{\mu}_{e_7} & B^{\mu}_{e_7} \\ C^{\mu}_{e_7} & D^{\mu}_{e_7} \end{pmatrix}
\tag{7.242}
$$

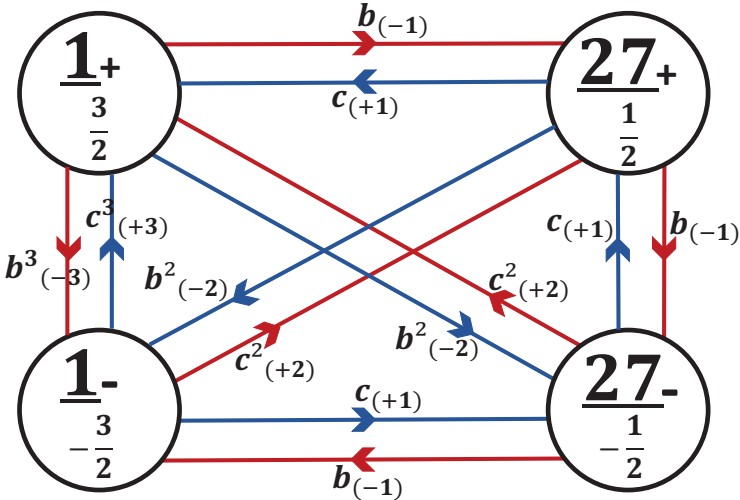

Figure 28: The topological quiver $Q_{E_7}$ representing $\mathcal{L}_{E_7}$. It has 4 nodes and 12 links. The nodes describe self-dual topological matter. The links describe bi-matter in $(R_i, \bar{R}_j)$ of $E_6$ charged under $SO(2)$ with charges $\pm 1, \pm 2, \pm 3$.

- *the block A* : concerns the sector $\mathbf{28}_+$ of $\mathbf{56}$ :

$$A^\mu_{e_7} = \begin{pmatrix} \varrho_{\mathbf{1}_+}\mathcal{L}\varrho_{\mathbf{1}_+} & \varrho_{\mathbf{1}_+}\mathcal{L}\varrho_{\mathbf{27}_+} \\ \varrho_{\mathbf{27}_+}\mathcal{L}\varrho_{\mathbf{1}_+} & \varrho_{\mathbf{27}_+}\mathcal{L}\varrho_{\mathbf{27}_+} \end{pmatrix} = \begin{pmatrix} A^I_I & A^{II}_I \\ A^I_{II} & A^{II}_{II} \end{pmatrix} \tag{7.243}$$

with

$$\begin{array}{rcl}
A^I_I &=& z^{\frac{3}{2}}\varrho_{\mathbf{1}_+} + z^{\frac{1}{2}}X\varrho_{\mathbf{27}_+}Y + \frac{1}{4}z^{-\frac{1}{2}}X^2\varrho_{\mathbf{27}_-}Y^2 + \frac{1}{36}z^{-\frac{3}{2}}X^3\varrho_{\mathbf{1}_-}Y^3 \\
A^{II}_I &=& z^{\frac{1}{2}}X\varrho_{\mathbf{27}_+} + \frac{1}{2}z^{-\frac{1}{2}}X^2\varrho_{\mathbf{27}_-}Y + \frac{1}{12}z^{-\frac{3}{2}}X^3\varrho_{\mathbf{1}_-}Y^2 \\
A^I_{II} &=& z^{\frac{1}{2}}\varrho_{\mathbf{27}_+}Y + \frac{1}{2}z^{-\frac{1}{2}}X\varrho_{\mathbf{27}_-}Y^2 + \frac{1}{12}z^{-\frac{3}{2}}X^2\varrho_{\mathbf{1}_-}Y^3 \\
A^{II}_{II} &=& z^{\frac{1}{2}}\varrho_{\mathbf{27}_+} + z^{-\frac{1}{2}}X\varrho_{\mathbf{27}_-}Y + \frac{1}{4}z^{-\frac{3}{2}}X^2\varrho_{\mathbf{1}_-}Y^2
\end{array} \tag{7.244}$$

The $A^I_I$ and $A^{II}_{II}$ are associated to the nodes $\mathcal{N}_{\mathbf{1}_{3/2}}$ and $\mathcal{N}_{\mathbf{27}_{1/2}}$, while the sub-blocks $A^{II}_I$ and $A^I_{II}$ describe links between these nodes.

- *the block D* : concerns the sector $\mathbf{28}_-$ of the representation $\mathbf{56}$ :

$$D^\mu_{e_7} = \begin{pmatrix} z^{-\frac{1}{2}}\varrho_{\mathbf{27}_-} + z^{-\frac{3}{2}}X\varrho_{\mathbf{1}_-}Y & z^{-\frac{3}{2}}X\varrho_{\mathbf{1}_-} \\ z^{-\frac{3}{2}}\varrho_{\mathbf{1}_-}Y & z^{-\frac{3}{2}}\varrho_{\mathbf{1}_-} \end{pmatrix} \tag{7.245}$$

Where $D^I_I$ and $D^{II}_{II}$ are associated to $\mathcal{N}_{\mathbf{27}_{-1/2}}$ and $\mathcal{N}_{\mathbf{1}_{-3/2}}$ and $D^{II}_I$ and $D^I_{II}$ are associated to links between them.

- *the blocks B and C:* Describe couplings between sectors $\mathbf{28}_+$ and $\mathbf{28}_-$ :

$$B^\mu_{e_7} = \begin{pmatrix} \frac{1}{2}X^2 z^{-\frac{1}{2}}\varrho_{\mathbf{27}_-} + \frac{1}{6}z^{-\frac{3}{2}}X^3\varrho_{\mathbf{1}_-}Y & \frac{1}{6}z^{-\frac{3}{2}}X^3\varrho_{\mathbf{1}_-} \\ z^{-\frac{1}{2}}X\varrho_{\mathbf{27}_-} + \frac{1}{2}z^{-\frac{3}{2}}X^2\varrho_{\mathbf{1}_-}Y & \frac{1}{2}z^{-\frac{3}{2}}X^2\varrho_{\mathbf{1}_-} \end{pmatrix} \tag{7.246}$$

$$C_{e_7}^{\mu} = \begin{pmatrix} \frac{1}{2} z^{-\frac{1}{2}} \varrho_{\mathbf{27}_-} Y^2 + \frac{1}{6} z^{-\frac{3}{2}} X \varrho_{\mathbf{1}_-} Y^3 & z^{-\frac{1}{2}} \varrho_{\mathbf{27}_-} Y + \frac{1}{2} z^{-\frac{3}{2}} X \varrho_{\mathbf{1}_-} Y^2 \\ \frac{1}{6} z^{-\frac{3}{2}} \varrho_{\mathbf{1}_-} Y^3 & \frac{1}{2} z^{-\frac{3}{2}} \varrho_{\mathbf{1}_-} Y^2 \end{pmatrix}$$

Entries of these matrices give 4+4 links between the nodes' pairs $\left( \mathcal{N}_{\mathbf{1}_{3/2}}, \mathcal{N}_{\mathbf{27}_{1/2}} \right)$ and $\left( \mathcal{N}_{\mathbf{27}_{-1/2}}, \mathcal{N}_{\mathbf{1}_{-3/2}} \right)$.

And so indeed, the topological gauge quiver $Q_{e_7}^{\mu}$ associated with $\mathcal{L}_{e_7}^{\mu}$ has four nodes $\mathcal{N}_i$ corresponding to the $e_6$ representations

$$\begin{array}{llll} \mathcal{N}_1 & : & \mathbf{1}_{+3/2} & , & \mathcal{N}_3 & : & \mathbf{27}_{-1/2} \\ \mathcal{N}_2 & : & \mathbf{27}_{+1/2} & , & \mathcal{N}_4 & : & \mathbf{1}_{-3/2} \end{array} \tag{7.247}$$

and describing self-dual topological gauge matter. It also has 12 links $L_{ij}$ describing topological bi-fundamental gauge matter $\langle R_i \bar{R}_j \rangle$ as collected in the following tables

| link | Repres | bi-matter | link | Repres | bi-matter |
|------|--------|-----------|------|--------|-----------|
| $L_{1\to2}$ | $\langle 1_{+3/2} \overline{27}_{-1/2} \rangle$ | $b^\alpha$ | $L_{2\to3}$ | $\langle 27_{+1/2} \overline{27}_{+1/2} \rangle$ | $b^\alpha$ |
| $L_{1\to3}$ | $\langle 1_{+3/2} \overline{27}_{+1/2} \rangle$ | $B^\alpha$ | $L_{2\to4}$ | $\langle 27_{+1/2} \bar{1}_{+3/2} \rangle$ | $B^\alpha$ |
| $L_{1\to4}$ | $\langle 1_{+3/2} \bar{1}_{+3/2} \rangle$ | $B_\alpha b^\alpha$ | $L_{3\to4}$ | $\langle 27_{-1/2} \bar{1}_{+3/2} \rangle$ | $b_\alpha$ |

(7.248)

and

| link | Repres | bi-matter | link | Repres | bi-matter |
|------|--------|-----------|------|--------|-----------|
| $L_{1\leftarrow2}$ | $\langle \bar{1}_{-3/2} 27_{+1/2} \rangle$ | $c_\alpha$ | $L_{2\leftarrow3}$ | $\langle \overline{27}_{-1/2} 27_{-1/2} \rangle$ | $c^\alpha$ |
| $L_{1\leftarrow3}$ | $\langle \bar{1}_{+3/2} 27_{+1/2} \rangle$ | $C_\alpha$ | $L_{2\leftarrow4}$ | $\langle \overline{27}_{-1/2} 1_{-3/2} \rangle$ | $C^\alpha$ |
| $L_{1\leftarrow4}$ | $\langle \bar{1}_{+3/2} 1_{+3/2} \rangle$ | $c_\alpha C^\alpha$ | $L_{3\leftarrow4}$ | $\langle \overline{27}_{+1/2} 1_{-3/2} \rangle$ | $c_\alpha$ |

(7.249)

In these tables, $B^\gamma$ stands for $b^\alpha \Gamma_{\alpha\beta}^\gamma b^\beta$ having charge $-2$, and $C_\gamma$ refers to $c_\alpha \bar{\Gamma}_\gamma^{\alpha\beta} c_\beta$ having charge $+2$. The composites $B_\alpha b^\alpha$ and $c_\alpha C^\alpha$ have charges -3 and +3 respectively.

# 8 Conclusion and comments

The results presented in this paper are based on the correspondence between two dimensional integrable models and four dimensional Chern-Simons gauge theory as formulated in [23]. In the $\boldsymbol{M}_4 = \mathbb{R}^2 \times \mathbb{CP}^1$ of the gauge theory, one can build an integrable lattice model by implementing a set of line defects looking like curves on $\mathbb{R}^2$ and points on $\mathbb{CP}^1$. In such construction, the integrability of the corresponding low-dimensional system constrained by the Yang Baxter or RLL equation is a direct result of the mixed topological-holomorphic nature of the line defects and the diffeomorphism invariance in four dimensions. The RLL equation for example, corresponds to the graphical equivalence of the intersections in different orders of two electric Wilson lines with one magnetic 't Hooft line, see Figure **5**. In this image, the explicit Feynman diagrams calculation for the intersection of two Wilson lines in

4D CS yields the first order expansion of the R-matrix acting on the two quantum spaces carried by the electrically charged lines [21]- [23]. The L-operator is realised as the intersection of an electric Wilson line with a magnetic 't Hooft line whose oscillator phase space acts as an auxiliary space [52].

This Wilson/'t Hooft coupling in the 4D CS theory is the particularly interesting ingredient of our current investigation, it allows to realise the Lax matrix as a building block of the transfer matrix generating conserved commuting quantities of the spin chain. This important quantity is calculated in the integrability literature using Yangian representations based techniques that can be cumbersome and inefficient in cases with complicated symmetries. Surprisingly, it was shown in [52] that the oscillator realisation of these L-operators for an XXX spin chain having the internal symmetry $g$ can be recovered from the analysis of solutions to the equations of motion of the 4D CS theory with gauge symmetry $G$, in the presence of interacting Wilson and 't Hooft lines. A general formula describing the coupling of a Wilson line with electric charge in a representation $\boldsymbol{R}$ of $G$ and a 't Hooft line with magnetic charge given by a minuscule coweight $\mu$ of $G$ reads as $\mathcal{L}_{\boldsymbol{R}}^{\mu} = e^{X_{\boldsymbol{R}}} z^{\mu} e^{Y_{\boldsymbol{R}}}$. This yields a matrix representation in terms of harmonic oscillators in $X_{\boldsymbol{R}}$ and $Y_{\boldsymbol{R}}$ with sub-blocks following from the Levi decomposition of $\boldsymbol{R}$ with respect to $\mu$.

The first part of our contribution concerned the exploitation of this formula to explicitly calculate this coupling for different types of 't Hooft and Wilson line defects in 4D Chern Simons theories with $SL_N$, $SO_{2N}$, E$_6$ and E$_7$ gauge symmetries. In particular, we investigated the splitting of various representations under the action of minuscule coweights as a first step towards the construction of L-operators in representations beyond the fundamental for ADE Lie algebras. Therefore, a better understanding of the effect of the Dirac-like singularity on the gauge field bundles behavior and the internal quantum states of a spin chain.

We remarked that the L-operators have unified intrinsic features that can be represented by topological quiver diagrams $Q_{\boldsymbol{R}}^{\mu}$ having a formal similarity with the well known graphs $Q_G^{susy}$ of supersymmetric quiver gauge theories embedded in type II strings. This formal link gives an interesting interpretation of the Darboux coordinates $(b^{\alpha}, c_{\beta})$ of the phase space of the L-operators in terms of topological bi-fundamental matter. In this regard, we gave several examples to ($i$) explain the strong aspects of this diagrammatic approach, and ($ii$) to show how it can be used to forecast the general form of the matrix representation of L-operators by indicating the action of its sub-blocks and their charges in terms of combinations of Darboux coordinates.

In particular, For the A-type Chern Simons theory, all fundamental coweights are minuscule, and therefore we give in Figure **29**, for a generic magnetic charge $\mu_k$ of $sl_N$, four quiver diagrams describing L-operators classified by representations $\boldsymbol{R}$ of the Wilson line.

In the case of D-type symetry, we have two types of minuscule 't Hooft lines associated to the vectorial and spinorial coweights of the $SO_{2N}$ gauge symmetry. In the figure **30**, we give quiver diagrams describing four possibilities of Wilson/'t Hooft couplings: a magnetic charge $\mu_1$ with electric $\boldsymbol{R = 2N}$ and with $\boldsymbol{R = adjso_{2N}}$, and magnetic $\mu_N \sim \mu_{N-1}$ with electric $\boldsymbol{R = 2^{N-1}}$ and with $\boldsymbol{R = adjso_{2N}}$.

The Figure **31** represents quiver gauge diagrams of exceptional type where we gave for each one of the E$_6$ and E$_7$ 4D CS theories the graphical descriptions for the coupling of the

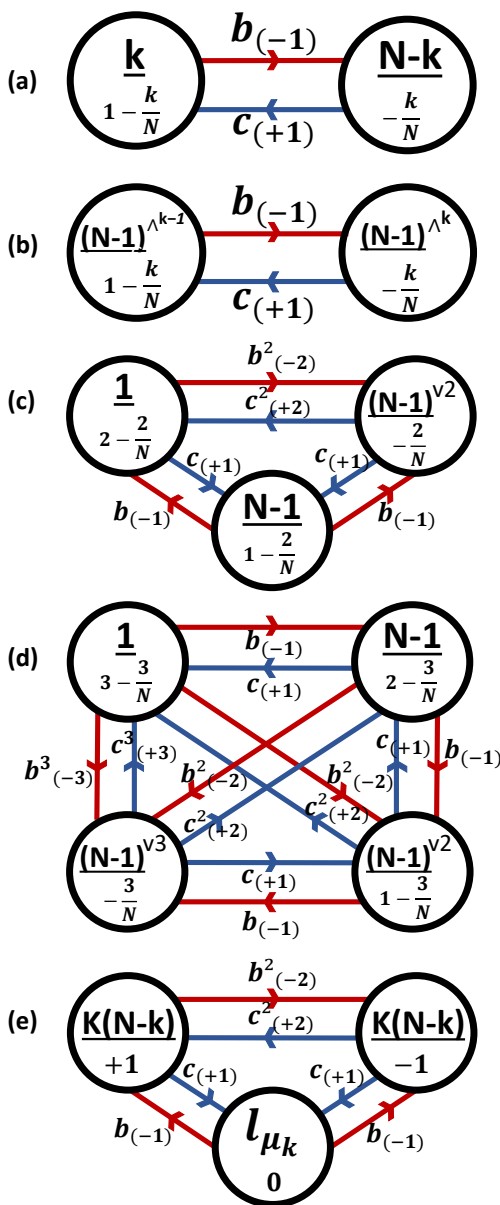

Figure 29: Leading elements of topological quiver digrams for the L-operators of A- type. These quivers are classified by the magnetic charge $\mu_k$ of the 't Hooft line and the representation $\boldsymbol{R}$. (a) Wilson line with charge $\boldsymbol{R} = \boldsymbol{N}$. (b) Wilson line with $\boldsymbol{R} = \boldsymbol{N}^{\wedge k}$. (c) Wilson line with $\boldsymbol{R} = \boldsymbol{N}^{\vee 2}$. (d) Wilson line with $\boldsymbol{R} = \boldsymbol{N}^{\vee 3}$. (e) Wilson line with charge $\boldsymbol{R} = \boldsymbol{adj} sl_N$.

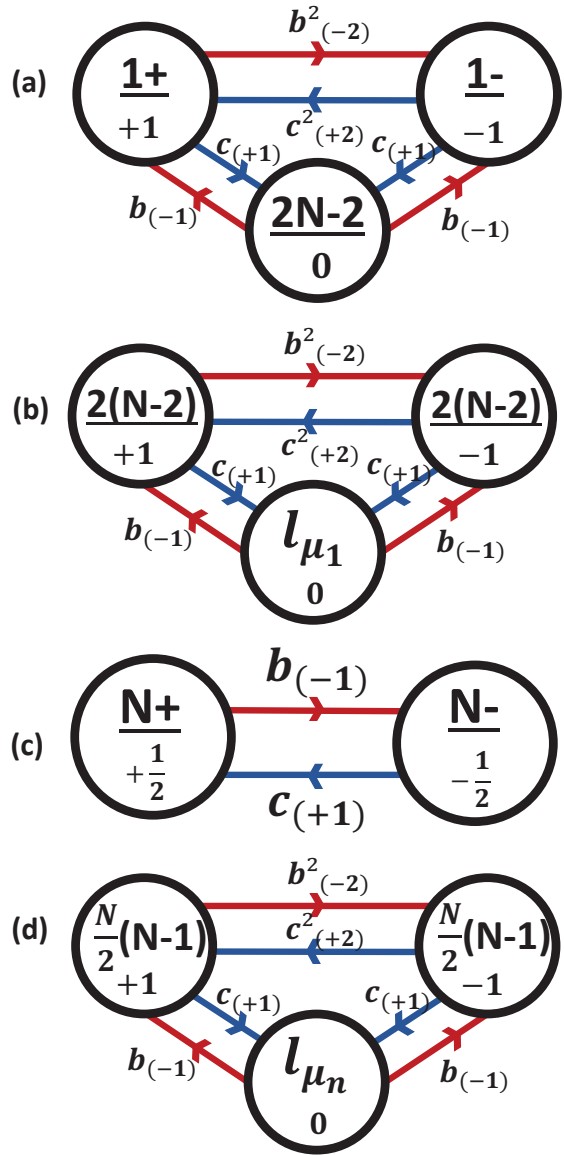

Figure 30: Leading elements of topological quiver digrams for the L-operators of D- type. The first two quivers correspond to the Levi decomposition with respect to the (vectorial) minuscule coweight $\mu_1$: (**a**) Wilson line with charge $R = 2N$. (**b**) Wilson line with $R = adjso_{2N}$. The other two quivers correspond to the Levi decomposition with respect to the (spinorial) minuscule coweight $\mu_N$: (c) Wilson line with $R = 2^{N-1}$. (d) Wilson line with $R = adjso_{2N}$.

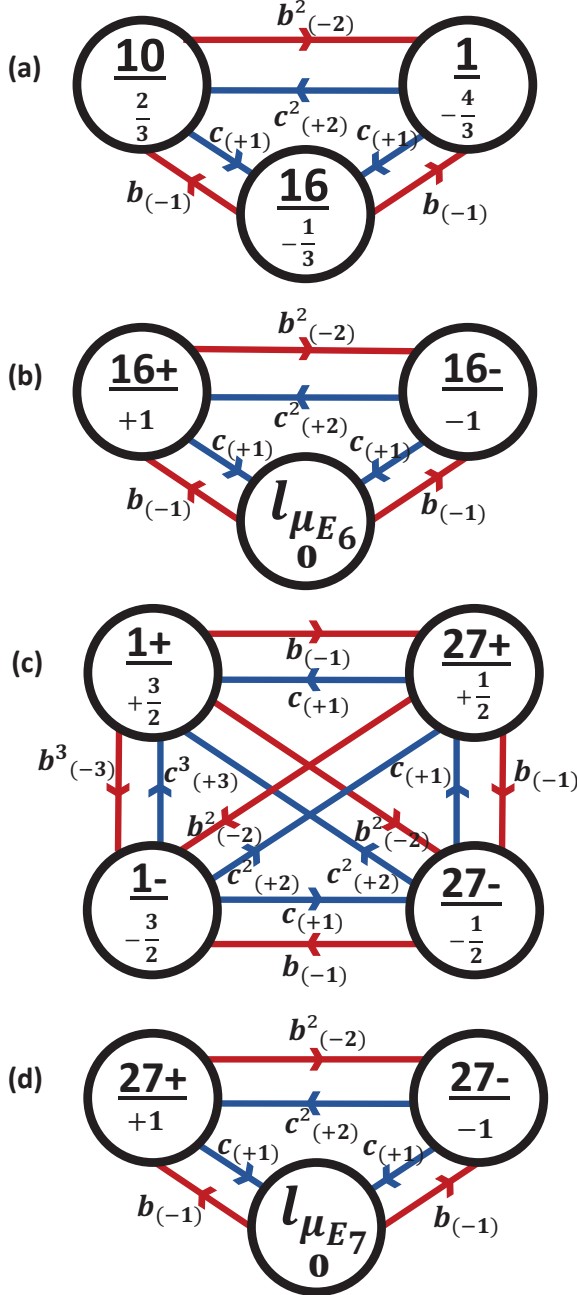

Figure 31: Leading elements of topological quiver digrams for the L-operators of E- type. The first two quivers for the $E_6$ gauge theory. (**a**) for the fundamental **27** of $E_6$; and (**b**) for the adjoint representation. The last two quivers regard the $E_7$ Chern-Simons theory. (**c**) for the fundamental **56** of $E_7$ and (**d**) for the adjoint representation.

minuscule 't Hooft line with Wilson lines in the fundamental and in the adjoint representations.

Notice however that this construction can be extended for the investigation of other L-operators that are still missing in the spin chain literature; and the interpretations associated to the components of the L-operator can also be used to link the diagrammatic description presented here to quiver diagrams associated to the realisation of 't Hooft line defects in supersymmetric quiver gauge theories.

Another exquisite property of this graphical quiver description in the 4D Chern-Simons topological theory is the natural appearance of a unified theory structure where the minuscule L-operators can be connected and classified in a larger $E_7$ 4D CS theory. In fact, the Lie algebras' decompositions with respect to minuscule coweights link the $E_7$ symmetry to the $E_6$ and then to the family of $D_N$ symmetries with $N \leq 5$ and/or the $A_N$ with $N \leq 4$. These chains of Levi decompositions lead to different possible paths for the $E_7$ symmetry breaking as described in Figure **32** [71]. To visualize this from the quiver descriptions of L-operators, we can focus on those corresponding to the fundamental representations and notice that the $Q_{fund}^{\mu(e_7)}$ has a node corresponding to the **27** of $E_6$; this node can be therefore imagined as including the $Q_{fund}^{\mu(e_6)}$ which in turn includes the $Q_{fund}^{\mu_{vect}(so_{10})}$ and so on. Finally, notice that the calculation of minuscule L-operators in 4D CS theories with $SO_{2N+1}$ and $SP_{2N}$ symmetries having each only one minuscule coweight, shows that the $\mathcal{L}_{so_{2N+1}}$ matrix is very similar to $\mathcal{L}_{so_{2N}}^{vect}$ while the $\mathcal{L}_{sp_{2N}}$ is similar to $\mathcal{L}_{so_{2N}}^{spin}$ [60]. This means that the corresponding quivers look like $Q_{so_{2N}}^{\mu_1}$ and $Q_{so_{2N}}^{\mu_N}$ which allows to include the B and C -type symmetries into this unified classification.

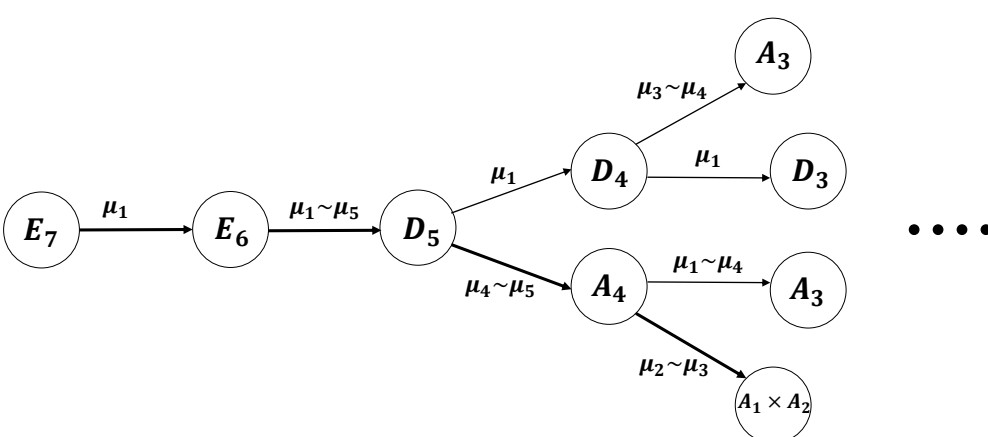

Figure 32: Breaking chains of $E_7$ symmetry as given by Levi decompositions with respect to minuscule coweights. The bold arrows describe the exceptional sequence leading to the Standard model-like group. The minuscule coweights $\mu$ correspond to the Lie algebra at which the arrow starts

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
