# Peer review of "'t Hooft lines of ADE-type and Topological Quivers"

_SciPost Physics_

## Round 1 · Referee Report · Anonymous · 2023-2-8

Report

Please see attached pdf.

Attachment

  • validity: -
  • significance: -
  • originality: -
  • clarity: -
  • formatting: -
  • grammar: -

Author:  Youssra Boujakhrout  on 2023-03-03  [id 3431]

(in reply to Report 1 on 2023-02-08)
Category:
answer to question

Please see the attached file for answers to the questions report 1.

Attachment:

Y-Reply-to-referee1-scipost_202212_00073v1.pdf

Anonymous on 2023-03-06  [id 3442]

(in reply to Youssra Boujakhrout on 2023-03-03 [id 3431])

I'm happy to say that from the reply it seems my questions have been addressed, though I don't seem to be able to see the updated draft. Has it been uploaded?

Anonymous on 2023-03-19  [id 3496]

(in reply to Anonymous Comment on 2023-03-06 [id 3442])

Thank you. The submission page says I can only update the response to the report; and should not upload the revised version of the paper.

---

## Round 1 · Referee Report · Anonymous · 2023-3-20

# Referee Report on Boujakhrout et. al.

## immediate

## March 20, 2023

The subject of the paper " 't Hooft lines of ADE-type and Topological Quivers" by Boujakhrout et. al. lies in the general area of theoretical physics that explores the relationship between integrable systems in two dimensions, and gauge theories of a mixed topological-holomorphic nature in four dimensions. The particular quantum field theory that is the origin of many of the cornerstone notions in two-dimensional integrability in a four-dimensional version of Chern-Simons theory.

The current paper focuses and builds upon the following aspect of the integrability/4dCS relationship. A particularly important ingredient in integrable spin chains is Baxter's Q-operator, which by using certain algebraic relations allows one to write down the Bethe equations of the spin chain. A convenient construction of the Q-operator for the XXX spin chain is given by taking the trace of a product of L-operators (solutions to the RLL integrability equation) valued in a certain Fock representation. The four-dimensional Chern-Simons origin of the Q-operator was shown to be 't Hooft line defects in Costello, Gaiotto, and Yamazaki's 2021 paper. They proceeded by reproducing these Fock-valued L-operators for the gauge groups $SL(n)$ and $SO(n)$ by an analysis of the phase space of four-dimensional Chern-Simons theory in the presence of 't Hooft lines carrying minuscule charge. In the present paper, Boujakhrout et. al. inspired by this construction of the L-operator show how various aspects of it can be conveniently cast into certain graphs which they term as "topological gauge quivers." They work out these quivers for the simple Lie algebras of type $A$ and $D$ along with $E_6$ and $E_7$.

I believe the manuscript meets the criteria for publication after the following suggested modifications are made.

1. Fixing of many typographical errors. A sample of these errors include the second paragraph of page 2: "inersecting" instead of "intersecting", second paragraph of page 3, "interprete" instead of "interpret", and "Topololgical" instead of "Topological" in the heading of section 3.3. There are many more but I only list a sample.

2. In the second paragraph of page 2, the reference [31] seems to be the wrong one (it cites a paper by Fukushima et. al. where as the correct one I believe is [21]).

3. In the same paragraph, it seems inaccurate to state that the Wilson lines are "intersecting", as they are usually taken to be separated in the holomorphic plane. "Crossing" is more accurate.

4. The reference [26] that is credited with writing the 4d Chern-Simons action in the first paragraph of 2.1 again seems inaccurate (it refers to a 2019 paper by Vicedo).

5. In the same paragraph above it is worth mentioning that we consider $\mathrm{SL}_N$ over the complex numbers $\mathbb{C}$, and so all gauge-fields are complex-valued.

6. Again in the same paragraph, I find the phrasing when mentioning the "missing" component confusing, since it comes directly after equation (2.3) is written. Instead it should be discussed as an alternate viewpoint.

7. The equations of motion written in (2.4) are not quite accurate since they also imply $F_{z\bar{z}} = 0$, and the 4d CS EOM don't have such an equation.

8. Directly below that, on the first paragraph of page 5, the "topological nature" of the theory is mentioned. It is important to note that it is only a mixture of topological and holomorphic.

9. In the second-to-last paragraph on page 5, the adjoint representation is referred to as $N \times \overline{N}$. This does not seem accurate to me.

10. The caption under Figure 2 has a paranthesis with "(x=cte)", which does not make much sense. It seems like a typographical error.

11. In the discussion of the Wilson lines, the "naive" viewpoint is mentioned. In order to not mislead the reader a more accurate viewpoint should also be remarked upon (in particular it should mention the relevance and importance of the Yangian $Y(\mathfrak{g})$). Similar remarks should be included for 't Hooft operators (namely it should be clarified that the discussion is only being carried out at the semi-classical level).

12. The left hand side of equation (2.7) is a one-form whereas the right hand side is a function. Is it meant to refer to a specific component (the $y$-direction)?

13. In the same paragraph the notation $A^{[\mu]}$ and $\mathcal{A}^{[\mu]}$ are inconsistently used.

14. In (2.8) the $z$-dependence in the right-hand side should be made more explicit.

---

## Editorial Decision

resubmitted